# IN-CONTEXT LEARNING LEARNS LABEL RELATION-SHIPS BUT IS NOT CONVENTIONAL LEARNING

**Jannik Kossen**[1▽]        **Yarin Gal**[1△]        **Tom Rainforth**[2△]

[1] OATML, Department of Computer Science, University of Oxford
[2] Department of Statistics, University of Oxford

## ABSTRACT

The predictions of Large Language Models (LLMs) on downstream tasks often improve significantly when including examples of the input–label relationship in the context. However, there is currently no consensus about *how* this in-context learning (ICL) ability of LLMs works. For example, while Xie et al. (2022) liken ICL to a general-purpose learning algorithm, Min et al. (2022b) argue ICL does not even learn label relationships from in-context examples. In this paper, we provide novel insights into how ICL leverages label information, revealing both capabilities and limitations. To ensure we obtain a comprehensive picture of ICL behavior, we study probabilistic aspects of ICL predictions and thoroughly examine the dynamics of ICL as more examples are provided. Our experiments show that ICL predictions almost always depend on in-context labels and that ICL can learn truly novel tasks in-context. However, we also find that ICL struggles to fully overcome prediction preferences acquired from pre-training data and, further, that ICL does not consider all in-context information equally.

## 1 INTRODUCTION

Brown et al. (2020) have shown that Large Language Models (LLMs) (Radford et al., 2019; Chowdhery et al., 2022; Hoffmann et al., 2022; Zhang et al., 2022a) can perform so-called *in-context learning* (ICL) of supervised tasks. In contrast to standard in-weights learning, e.g. gradient-based finetuning of model parameters, ICL requires no parameter updates. Instead, examples of the input–label relationship of the downstream task are simply prepended to the query for which the LLM predicts. This is sometimes also referred to as *few-shot ICL* to differentiate from other ICL variants that do not use example demonstrations (Liu et al., 2023). Few-shot ICL is widely used, e.g. in all LLM publications cited above, to improve predictions across a variety of established NLP tasks, such as sentiment or document classification, question answering, or natural language inference.

However, there is currently no consensus on *why* ICL improves predictions, with prior work presenting a large variety of often contradictory perspectives. For example, Brown et al. (2020) highlight similarities between the behavior of ICL and finetuning of LLMs, such as improvements with model size and number of examples. Since then, some have argued that ICL works because it implements general-purpose learning algorithms such as Bayesian inference or gradient descent (Xie et al., 2022; Huszár, 2023; Hahn & Goyal, 2023; Jiang, 2023; Zhang et al., 2023; Von Oswald et al., 2023; Akyürek et al., 2023; Han et al., 2023). In contrast, others have highlighted practical shortcomings of ICL, suggesting ICL does not really 'learn' from examples in the way one expects (Liu et al., 2022; Lu et al., 2022; Zhao et al., 2021; Chen et al., 2022; Agrawal et al., 2023; Chang & Jia, 2022; Razeghi et al., 2022; Li & Qiu, 2023; Wei et al., 2023). In particular, Min et al. (2022b) claim their 'findings suggest that [LLMs] do not learn new tasks at test time' in the sense of 'capturing the input-label correspondence'. Clearly, these claims, if true, are not compatible with the behavior we would expect from a general-purpose learning algorithm.

In this paper, we address the pressing need for an improved understanding of how information given in-context affects ICL predictions. Concretely, we formulate a set of questions that encode our beliefs about how an idealized *conventional learning algorithm* should incorporate label information

---

[▽]Correspondence to jannik.kossen@cs.ox.ac.uk.  [△] Equal advising.

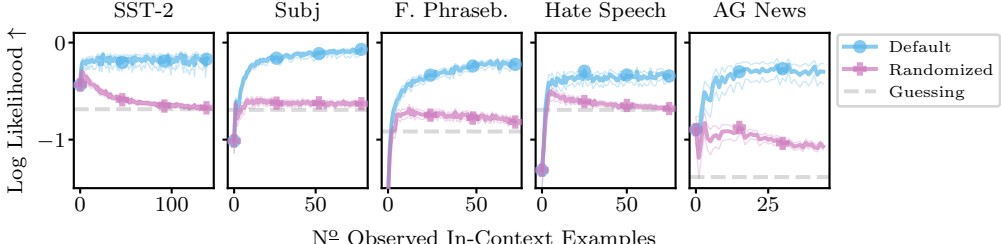

Figure 1: ICL predictions generally depend on the conditional label distribution of in-context examples: when in-context labels are randomized, average log likelihoods of label predictions decrease compared to ICL with default labels for LLaMa-2-70B across a variety of tasks. Results averaged over 500 in-context datasets and thin lines are 99 % confidence intervals. See §5 for details.

and then study ICL behavior in relation to this concept of a conventional learner. (1) Does ICL take the input–label relationship of the in-context examples into account when predicting for test queries? (2) Is ICL powerful enough to overcome prediction preferences originating from pre-training? (3) Does ICL treat all information provided in-context equally? To study these questions rigorously, we rephrase them as null hypotheses that we study empirically. Our results yield an improved understanding of the similarities and differences between ICL and idealized conventional learners.

Unlike prior work, we study in detail how ICL predictions evolve as an increasing number of examples are provided, from no examples at all up to the maximum possible, across a range of LLMs and tasks. We further show that using probabilistic metrics better highlights the resulting ICL dynamics, often revealing large changes in the confidence of ICL predictions, even when accuracy metrics barely change at all. These measures ensure we obtain a comprehensive picture of ICL behavior.

In our experiments, we first examine if ICL predictions depend on the labels of in-context examples by studying how probabilistic metrics react to randomized in-context labels (§5, Fig. 1). Further, we study ICL on a truly novel task the LLM *cannot* know from pre-training (§6). Both experiments show that ICL typically considers in-context label relations. We then investigate if ICL is powerful enough to overcome prediction preferences learned from pre-training data (§7). In our experiments, we find this is typically not the case as ICL performance plateaus if label relations oppose pre-training preference. Further, while additional prompting can improve ICL here, we ultimately do not find prompts that lead to the desired behavior. Finally, we study if ICL treats all information provided in-context equally (§8). This is important when the context contains multiple different label relationships. By modifying label relations during ICL, we find it does not treat all in-context information equally, and, instead, ICL preferentially makes use of information closer to the query.

In summary, our results suggest a new middle ground regarding the capabilities and limitations of ICL. While ICL can learn from label information, it does so differently than an idealized learner. Our findings thus contribute to a better understanding of information processing in ICL, which, in turn, is crucial to our ability to deploy LLMs safely and effectively. For example, Bai et al. (2022b) suggest to use ICL for alignment, which relies on ICL being able to sufficiently adjust LLM behavior.

## 2 BACKGROUND

A large and growing body of prior work studies ICL. We here highlight those studies that are most relevant to our motivation, and discuss other related work later in §10.

Since its introduction by Brown et al. (2020), few-shot ICL has become an integral part of LLM evaluations: e.g. many recent publications rely on few-shot ICL tasks, such as the popular HELM benchmark (Liang et al., 2022), to evaluate their LLMs (Chowdhery et al., 2022; Hoffmann et al., 2022).

In the wake of ICL's success, follow up work has speculated if ICL implements a general purpose learning algorithm such as gradient descent (Von Oswald et al., 2023) or Bayesian inference (Xie et al., 2022). This line of work implies that ICL captures not just how much LLMs have learned during pre-training but, rather, how much LLMs have *learned how to learn* novel supervised tasks in-context. However, so far arguments have been largely theoretical, lacking solid experimental evidence in actual LLMs (Zhang et al., 2023; Huszár, 2023; Wies et al., 2023; Jiang, 2023).

Conversely, a variety of studies have highlighted unexpected shortcomings of ICL. For example, ICL can be sensitive to the formatting (Min et al., 2022b) or order (Lu et al., 2022) of the in-context learning examples. Further, LLMs prefer to predict labels that are common in the pre-training data (Zhao et al., 2021), they can predict drastically different for similar prompts (Chen et al., 2022), and they rely on task formulations similar to those observed in the pre-training data (Wu et al., 2023).

In particular, Min et al. (2022b) claim that ICL does not learn label relationships from in-context examples and that ICL 'performance drops only marginally when labels in the demonstrations are replaced by random labels'. Further, they suggest that, instead of learning input–label relationships, ICL only works because the model learns about the general label space, the formatting of the examples, and their input distribution. They assert that ICL does 'not learn new tasks at test time' and that 'ground truth demonstrations are in fact not required' in many common scenarios. In the first part of our evaluation, we will revisit and ultimately disagree with these claims.

## 3 NULL HYPOTHESES ON HOW ICL INCORPORATES LABEL INFORMATION

We wish to obtain a better understanding of how ICL uses information about the input–label relationship provided in-context. We therefore study to what extent ICL behavior matches our expectations of how a machine learning algorithm *should* behave in an idealized world. To this end, we introduce the concept of a 'conventional learning algorithm' as an algorithm that conforms to our intuitions of how an idealized learner will make predictions given data. In particular, we focus on the following intuitions of an idealized learner: (1) it makes use of the labels for learning, (2) it allows the true label relationship to be learned when provided with sufficient data, and (3) it considers all information in the data equally. Note here that some existing approaches may not always conform to some or all of these intuitions, e.g. due to imperfections in training schemes, but our concept of the conventional learning algorithm reflects how we would expect them to behave if our training worked perfectly.

To allow these institutions to be tested for ICL, we now introduce a series of null hypotheses that we will subsequently try to falsify. Our first hypothesis follows from the notion that conventional learners make use of the conditional distribution of labels given inputs.

***Null Hypothesis 1 (NH1)**: ICL predictions are independent of the conditional label distribution of the examples given in-context.*

We will investigate NH1 in multiple ways. In §5, we revisit and refine the randomized in-context label experiment of Min et al. (2022b): in addition to revising their results for point predictions, we propose the use of *probabilistic metrics* to study if label randomization really does not affect ICL predictive beliefs. Then, in §6, we study ICL on a novel task that we create and which ICL can *only* solve by learning a novel label relationship that it cannot know from pre-training.

Next, we study *how* ICL incorporates label information. The pre-trained model already contains information towards many NLP tasks: even without any ICL, predictions are significantly better than random guessing. This raises the question of how information given in-context interacts with this *pre-training preference*. In typical applications of conventional learners, we would expect that predictions eventually follow the label relation of the training examples when provided with sufficient data. A learner that does not conform to this is inherently limited in what it can learn. With NH2, we study if ICL conforms to these intuitions and can overcome the initial pre-training preference.

***Null Hypothesis 2 (NH2)**: ICL can overcome the zero-shot prediction preferences of the pre-trained model.*

In §7, we modify in-context label relationships and study how this changes ICL predictions. If NH2 is true, ICL should eventually predict according to any label relation given in-context.

Our last hypothesis relies on the following typical property of conventional learning algorithms: they will consider all information in the examples equally. If a dataset contains multiple sources of information about a label relationship, e.g. the dataset itself is a union of multiple datasets, then all information is considered equally by the learner. NH3 investigates if this is holds true for ICL as well.

***Null Hypothesis 3 (NH3)**: ICL considers all information given in-context equally.*

In §8, we study non-stationary input distributions for which label relations *change* during ICL. If NH3 is true, ICL predictions should not depend on the order in which we present label relations.

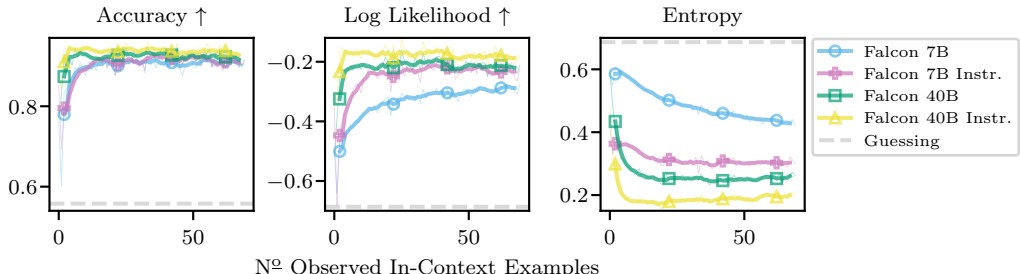

Figure 2: Few-shot ICL **training dynamics** in a default label scenario on SST-2. Accuracy (↑) and log likelihood (↑) improve with in-context dataset size, and entropies decrease appropriately. Averages over 500 random subsets, thick lines with moving average (window size 5) for clarity.

## 4  EXPERIMENTAL SETUP & ICL TRAINING DYNAMICS

We here detail our experimental setup for the subsequent evaluation of few-shot ICL behavior.

**Models & Tasks.** We employ LLMs from the LLaMa-2 (Touvron et al., 2023b), LLaMa (Touvron et al., 2023a), and Falcon (TII, 2023) families due to their strong performance and open source nature. We evaluate on SST-2, Subjective (Subj.), Financial Phrasebank (FP), Hate Speech (HS), AG News (AGN), MQP, MRPC, RTE, and WNLI. We provide citations for all tasks in §D.

**Context Size.** We always report few-shot ICL performance across *all possible* numbers of in-context demonstrations, i.e. from zero-shot performance up to the maximum number of examples within the LLMs' input token limit. This is in contrast to prior work, which often evaluates few-shot ICL at only a few context set sizes, and allows us to obtain a comprehensive picture of ICL *'training dynamics'*.

**Computationally Cheap ICL Evaluations.** We propose a novel evaluation strategy that obtains ICL predictions at all possible numbers of in-context demonstrations without incurring any additional cost. Concretely, we exploit the fact that each forward pass through the model gives not just a prediction for the next token, but rather, the predicted probabilities for *each* input token (given all preceding tokens). By extracting those token predictions that correspond to labels of in-context examples, we obtain few-shot ICL predictions at all in-context dataset sizes with each forward pass. We refer to §B for a formalization of few-shot ICL and further description of our evaluation strategy.

**Evaluation Metrics.** We evaluate few-shot ICL performance in terms of accuracy (↑) and (average) log likelihood (↑) of label predictions. We also report entropy, which, while not a performance metric, is useful for understanding how much predicted probabilities are spread over classes. We average metrics over sets of in-context examples drawn randomly and without replacement from the training set, and we compare to a guessing baseline that predicts with probabilities equal to class frequencies.

**Default Training Dynamics.** Before modifying label relationships in the following sections, Fig. 2 shows standard few-shot ICL training dynamics for Falcon models on SST-2. We observe reasonable behavior for all models: as more in-context examples are observed, accuracies and log likelihoods increase, while entropies decrease. Notably, log likelihoods and entropies show in-context *learning* more clearly: predicted probabilities continue to improve at larger context sizes, whereas accuracies saturate quickly. Differences between models are more noticeable for probabilistic metrics, too: entropies reveal that larger or instruction-tuned Falcon models predict with higher certainty on SST-2. Similar findings also hold for LLaMa and LLaMA-2 models, for which we provide results in Fig. F.1.

## 5  DO ICL PREDICTIONS DEPEND ON IN-CONTEXT LABELS?

We now study the null hypotheses (NH) formulated in §3, starting with NH1, which states that ICL predictions are independent of the conditional label distribution of the in-context examples. To this end, we first revisit the experiments of Min et al. (2022b), replacing all labels of in-context examples with labels drawn randomly from the training set of the task. If NH1 is true, then accuracy, log-likelihood, and entropy should be identical for the randomized and standard label scenario. We note that, while we believe the results of this experiment are already sufficient to reject NH1, the experiments in §6–§8 will provide additional strong evidence for this conclusion.

**Observations & Discussion.** We evaluate NH1 across all our models, tasks, and metrics, computing full ICL training curves as introduced in §4. Figure 1 shows log likelihoods for LLaMa-2-70B for

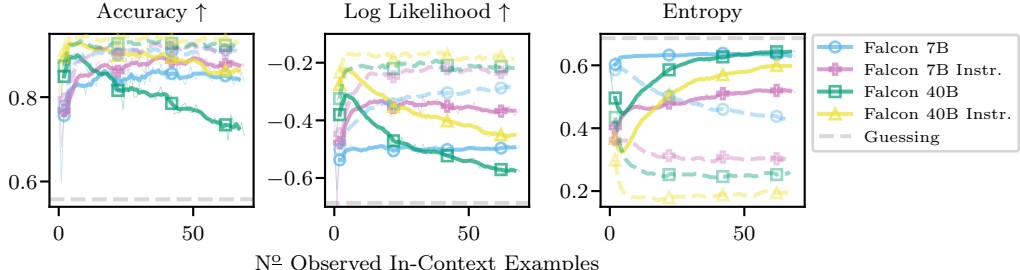

Figure 3: Few-shot ICL with **randomized labels** for SST-2: Compared to default ICL behavior (dashed lines), log likelihoods and entropies of the Falcon models degrade when in-context labels are randomized. Accuracies show differences less clearly than probabilistic log likelihood and entropy. Averages over 500 repetitions, thick lines with moving average (window size 5) for clarity.

Table 1: Average differences between ICL log likelihoods for default and randomized labels. Bold entries indicate differences are statistically significant. We can disregard lightgray entries: for them, default ICL performance is not significantly better than a random guessing baseline. Whenever default ICL outperforms the baseline, ICL almost always performs significantly worse (positive differences) for random labels. Averages over 500 runs at max. context size, standard errors in Table F.1.

| Δ Log Likelihood | SST-2 | Subj | FP | HS | AGN | MQP | MRPC | RTE | WNLI |
|---|---|---|---|---|---|---|---|---|---|
| LLaMa-2 7B | **0.42** | **0.39** | **0.57** | **0.18** | **0.53** | 0.03 | 0.02 | **0.03** | 0.02 |
| LLaMa-2 13B | **0.41** | **0.62** | **0.49** | **0.24** | **0.81** | **0.04** | 0.01 | **0.06** | **0.02** |
| LLaMa-2 70B | **0.51** | **0.53** | **0.57** | **0.34** | **0.80** | **0.29** | **0.04** | **0.22** | **0.18** |
| Falcon 7B | **0.20** | **0.19** | **0.25** | **0.06** | **0.31** | 0.01 | 0.01 | −0.01 | 0.01 |
| Falcon 7B Instr. | **0.13** | **0.08** | **0.11** | **0.03** | **0.15** | 0.03 | 0.02 | −0.00 | 0.00 |
| Falcon 40B | **0.34** | **0.35** | **0.31** | **0.18** | **0.90** | **0.06** | 0.01 | 0.01 | 0.02 |
| Falcon 40B Instr. | **0.25** | **0.37** | **0.27** | 0.02 | **0.77** | **0.06** | 0.02 | 0.02 | 0.04 |

a selection of tasks, and Fig. 3 shows all metrics for all Falcon models on SST-2. We observe significant differences in ICL behavior for the default and randomized label scenario. As the context size grows, likelihoods eventually degrade significantly for randomized labels. In Fig. 3, we can further see entropies increase when randomizing labels. This is reasonable from a probabilistic learning perspective: as noisy labels are observed, estimates of uncertainty will typically increase. While differences are large for probabilistic log likelihood and entropy, they can be harder to spot for accuracy. In Fig. 2, only Falcon-40B experiences a sizeable accuracy decrease. (For LLaMa(-2) models, accuracy decreases more frequently and can approach guessing level, cf. Figs. F.3 to F.8.)

We provide results for label randomization across all our models, tasks, and metrics: we invite the reader to view the full set of training curves in §F and provide a summary of the results in Table 1. It shows the average difference in log likelihoods between the default and randomized labels at the maximum number of demonstrations for each task and model. We gray out entries where ICL on default labels does not outperform the guessing baseline as we are only interested in studying label randomization when ICL works in the first place. When default label performance is better than random (black entries), differences are almost always significantly positive (bold entries), indicating ICL performs worse for randomized labels. ***Based on these results, we reject NH1 that ICL predictions do not depend on the conditional label distribution of in-context examples.***

Notably, Table 1 shows that LLaMa-2-70B, our largest and most capable model, always performs worse under label randomization. This suggests the importance of labels in ICL will increase as models become more powerful in the future. However, performance often degrades significantly even for smaller models, although they struggle to reach better than random performance on the entailment tasks MQP, MRPC, RTE, and WNLI. Occasionally, likelihoods improve despite random labels, e.g. for the small Falcon models in Fig. 3. Following Pan et al. (2023); Min et al. (2022b), we attribute this to ICL 'recognizing', rather than learning, the task from the random label demonstrations. However, we note that, even here, there are significant performance gaps to the default scenario. We conclude that label randomization adversely affecting ICL predictions is the rule not the exception.

**Discussion of Min et al. (2022b).** Lastly, we discuss possible reasons for why Min et al. (2022b) arrive at the conclusion that label randomization only 'barely hurts' ICL performance: (1) They do

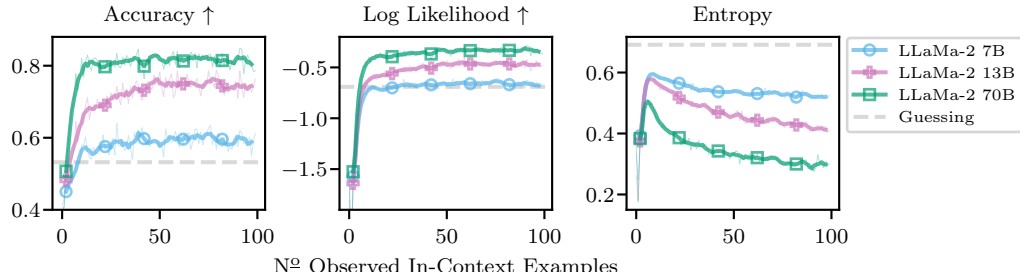

Figure 4: Few-shot ICL achieves accuracies significantly better than random guessing on our **novel author identification** task. Thus, LLMs can learn novel label relationships entirely in-context. Averages over 500 runs, thick lines with additional moving average (window size 5) for clarity.

not study probabilistic metrics, which are more sensitive to randomization. (2) They use a fixed ICL dataset size of 16, but effects of random labels increase with growing contexts. (3) Only one model they study has more than 20B parameters (GPT-3), but we observe that larger models react more to randomization. (Pan et al. (2023) also observe this, cf. §10.) (4) On some tasks, performance for Min et al. (2022b) could be close to random guessing, where label randomization has less of an effect.

## 6 CAN ICL LEARN TRULY NOVEL LABEL RELATIONSHIPS?

The results of §5 show that ICL predictions *do* depend on the label relationship of in-context examples. Here, we explore the *extent* to which ICL can extract label information from the context. Concretely, we study if LLMs can learn *truly novel* label relationships in-context. To do this, we create a task that is guaranteed to not appear in the pre-training data. The task needs to be distinct from established NLP tasks, for which the pre-training data could be contaminated and for which, often, strong zero-shot performance shows the model has learned the task, perhaps implicitly, during pre-training.

Specifically, we create an authorship identification (Stamatatos, 2009) dataset from private messages between two authors of this paper. The task is to identify the author corresponding to a given message. As messages stem from private communication, they are guaranteed to not be part of the pre-training corpus. For ICL to succeed here, it needs to learn the novel input–label relationship provided in-context: while the LLM could have some general notion of authorship identification tasks, the specific input–label relationship is definitely novel, as the authors' private writing styles cannot be known to the LLM. We give further details on the task in §C.

**Observations & Discussion.** Figure 4 shows that ICL with LLaMa-2 succeeds at learning the author identification task. Accuracies and log likelihoods increase, agreeing with expectations about conventional learning. Performance improves with model size, but all models perform better than random. We show results for more LLMs in Fig. F.13: all models, except Falcon-7B-Instruct, achieve better than random performance. We conclude that ***LLMs can learn truly novel tasks in-context***, correctly inferring the label relation from examples. These results also strongly support our previous rejection of NH1 as, clearly, ICL predictions must depend on labels to learn the novel task.

## 7 CAN ICL OVERCOME PRE-TRAINING PREFERENCE?

With NH2, we explore how in-context label information trades off against the LLM's *pre-training preference*, i.e. its zero-shot predictions based on label relationships inferred from pre-training and stored in the model parameters. Often, pre-training preference and in-context label relationships agree: e.g. in Fig. 3, performance is high zero-shot and then improves with ICL. To test NH2 if ICL can overcome pre-training preference, we create scenarios where pre-training preference and in-context observations are not aligned. We then study if ICL behavior is compatible with fully overcoming pre-training preference as we would expect from a conventional learner.

Concretely, we use replacement label relationships when constructing the in-context examples. (1) We flip the default labels, e.g. (negative, positive) get mapped to (positive, negative) for SST-2. (2) We study arbitrary labels, e.g. (negative, positive) become (A, B) or (B, A)—we deliberately choose *arbitrary* labels here such that the LLM should not have a significant preference for assigning them to positive or negative. We then evaluate ICL performance for predicting the *replacement*

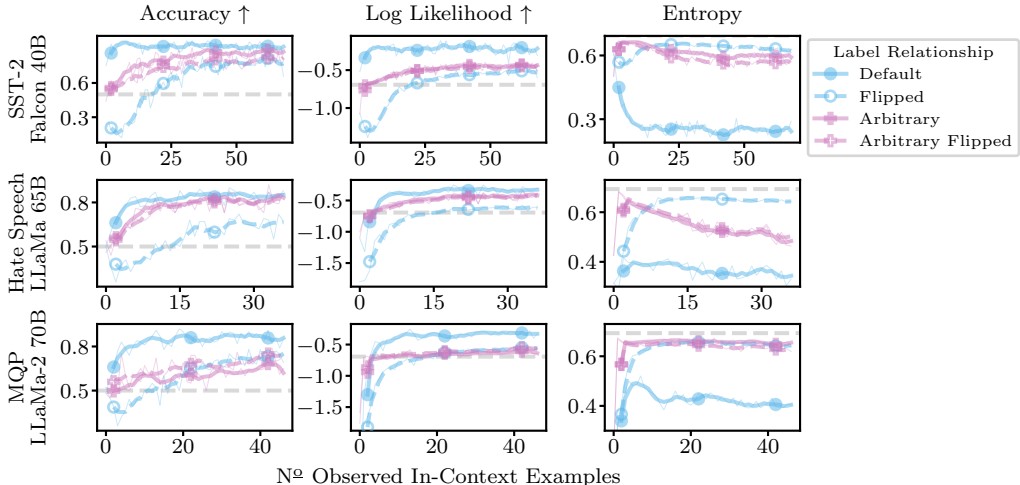

Figure 5: Few-shot ICL with **replacement labels** for Falcon-40B on SST-2, LLaMa-65B on Hate Speech, and LLaMa-2-70B on MQP. Table 2 and §F contain results for all other models and tasks. ICL achieves better than guessing performance for all label relations and models. However, predictions for flipped labels (dashed blue) plateau at a higher entropies and lower likelihoods than those for the default label relation (solid blue). For arbitrary labels (pink), the model performs similarly for both label directions. Averages over 100 runs and thick lines with moving average (window size 5).

Table 2: Average differences between ICL entropies for default and flipped labels. Bold entries indicate differences are statistically significant. Again, we disregard entries for which default ICL performance is not significantly better than the guessing baseline (lightgray entries). When default ICL outperforms the baseline, ICL entropies are almost always significantly different between scenarios. We average 100 runs, report results at maximum context size, and show standard errors in Table F.2.

| Δ Entropy | SST-2 | Subj | FP | HS | AGN | MQP | MRPC | RTE | WNLI |
|---|---|---|---|---|---|---|---|---|---|
| LLaMa-2 7B | **−0.52** | 0.01 | **−0.40** | **−0.10** | **−0.75** | −0.00 | 0.05 | **−0.03** | −0.01 |
| LLaMa-2 13B | **−0.48** | **−0.06** | **−0.47** | **−0.19** | **−0.95** | **−0.03** | −0.07 | **−0.11** | −0.07 |
| LLaMa-2 70B | **−0.17** | **−0.10** | **−0.40** | **−0.26** | **−1.00** | **−0.24** | **−0.15** | **−0.26** | **−0.21** |
| Falcon 7B | **−0.28** | **−0.12** | **−0.02** | **−0.06** | **−0.52** | 0.00 | 0.00 | −0.00 | −0.01 |
| Falcon 7B Instr. | **−0.37** | **−0.07** | **−0.33** | **−0.05** | **−0.22** | −0.02 | 0.13 | 0.01 | 0.00 |
| Falcon 40B | **−0.39** | **−0.23** | **−0.42** | **−0.19** | **−0.90** | −0.00 | −0.10 | **−0.02** | −0.00 |
| Falcon 40B Instr. | **−0.48** | **−0.16** | **−0.43** | **−0.31** | **−0.92** | **−0.10** | −0.02 | **−0.06** | **−0.01** |

*label relationship*, e.g. the flipped labels in (1). Note that, we rely on the flipped label experiments to evaluate NH2; results on arbitrary labels serve to complete the picture, and we discuss them later.

**Observations & Discussion.** We evaluate NH2 across all our models, tasks, and metrics. Figure 5 shows results for a selection of large models and datasets. Evidently, the LLMs can, to some extent, learn to predict the flipped label relationships against pre-training preference. Accuracies on flipped labels reach levels significantly better than random guessing. However, in particular for entropies, there is a consistent gap between the default and flipped label scenarios: ICL predictions on flipped labels are much less certain. Importantly, given the plateauing behavior, this gap is unlikely to disappear with additional in-context observations. (Practically speaking, we cannot actually add any additional examples as input size is maximal already and will deteriorate when exceeding the LLMs' input token limit; this is itself a limitation of ICL compared to conventional learning.) It seems that label relationships inferred from pre-training have a permanent effect that cannot be overcome through in-context observations. This does not agree with conventional learning: predictions on flipped labels should continue to improve as observations continue to contradict pre-training preference.

Crucially, we observe this behavior across models and tasks. We encourage the reader to view the full set of training curves in §F. Table 2 provides a summary of the results, showing differences in entropy between default and flipped label scenarios at maximum context size, highlighting statistical significance in bold and graying out entries where ICL fails on default labels. Across the board, we again observe that predictions on flipped labels plateau: a significant gap between predictions on default and flipped labels remains, even at maximum input size. For the models we study, *we reject*

***NH2 that ICL can overcome prediction preferences from pre-training.*** Again, the results here strongly support our previous rejection of NH1, as clearly, predictions change for replacement labels.

Figure 5 also shows that for replacement labels (A, B) and (B, A) both directions are similarly easy for ICL to learn. This suggests the LLM has not learned a preference for them from pre-training, agreeing with our intuition. Further, learning arbitrary labels is slower than learning default labels but faster than learning flipped labels, which agrees with intuitions about inductive biases.

**Can Prompting Help?** In §A, we further study if specific prompts, i.e. instructions that inform the LLMs of the flipped labels, can improve ICL predictions. We find that, while prompts initially can help the model predict on flipped labels, eventually, prompts no longer improve predictions.

## 8 How Does ICL Aggregate In-Context Information?

With NH3, we study if ICL considers all in-context information equally. We have just seen that ICL does not treat pre-training preference equivalently to in-context label information. However, it is similarly important to understand how ICL treats different sources of purely in-context information.

To test NH3, we change the label relationship *during* in-context learning in three different scenarios. (D → F): after $N$ observations of the **d**efault label relation we **f**lip the label relation for all following observations, e.g. from (negative, positive) to (positive, negative) for SST-2. (F → D): we now start with $N$ flipped label observations and then expose the model to default labels. (Alternate F ↔ D): we alternate between the default and flipped labels after *each* observation. For all three setups, after $2N$ observations, the model has observed the same number of the flipped and default label examples. If NH3 is true, ICL should treat all observed label relations equally, no matter their position in the input. This means, predictions should be the same for all three scenarios after $2N$ observations.

**Observations & Discussion.** Figure 6 shows accuracies for a selection of models and tasks, and we report full probabilistic metrics for all tasks and model combinations for which ICL was able to learn the flipped relationships well in Figs. F.32 to F.36. We observe that, across almost all tasks and model combinations, predictions are significantly different between the three setups after observing the same number of examples of both label relationships (after $2N$ total observations, red dashed line in the figure). ***We thus reject NH3 that ICL treats all information provided in-context equivalently.***

After the changepoint $N$, predictions immediately begin to adjust to the new label relationship. In particular, after $2N$ observations, the (F → D) setup has a bias for predicting according to the default label relationship, while the (D → F) setup has a bias for predicting the flipped label relationship.

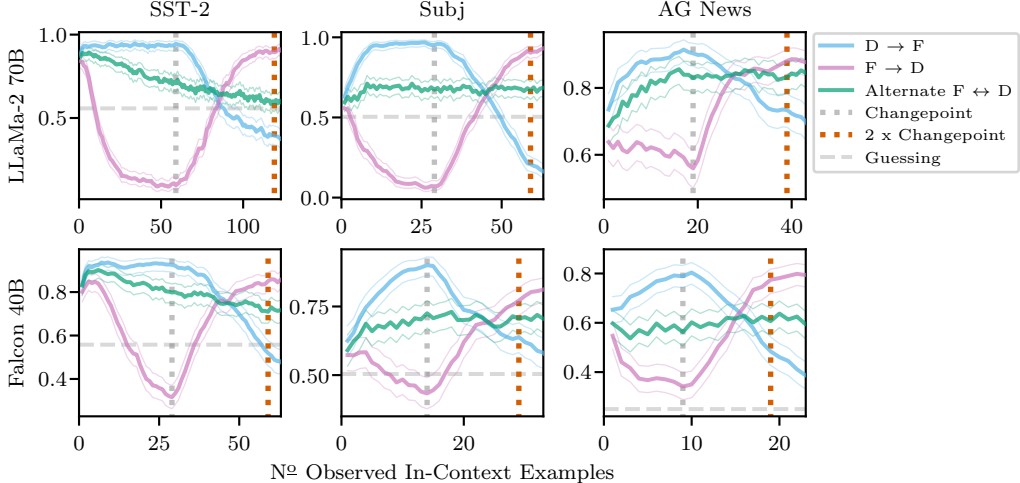

Figure 6: Few-shot ICL accuracies when the **label relationship changes throughout ICL**. For (D → F), we start with **d**efault labels and change to **f**lipped labels at the changepoint, for (F → D) we change from flipped to the default labels at the changepoint, and for (Alternating F ↔ D) we alternate between the two label relationships after every observation. For all setups, at '2 x Changepoint', the LLMs have observed the same number of examples for both label relations. If, according to NH3, ICL treats all in-context information equally, predictions should be equal at that point—but they are not. Bootstrapped $99\%$ confidence intervals, moving averages (size 3), and $500$ repetitions.

In other words, LLMs prefer to use information that is *closer* to the query, instead of considering all available information equally. Our finding is distinct from Zhao et al. (2021), who observe that ICL preferentially predicts labels that appear frequently near the query for a single *fixed* label relation. Lastly, we note once more that the results here also strongly support our previous rejection of NH1.

## 9 Discussion & Limitations

**Alignment.** For alignment of LLMs, it is crucial to understand how pre-training preference and inputs trade-off, as well as how different parts of the input, such as a context string and user input, interact and influence predictions. Our results suggest prompt-based alignment (Bai et al., 2022b) may struggle to overwrite pre-training preference and could itself easily be overcome by future user input.

**Do Labels Always Matter?** It is plausible that labels matter less for other NLP tasks such as question answering, where in-context examples may provide limited information towards the answer of the query question. However, for the randomized label experiment, capable LLMs might still identify that the provided in-context answers are random and imitate this in their predictions.

**Limitations.** We focus on few-shot ICL tasks where evaluation is based on logits and not free-form generation. We do this mostly to avoid complications around evaluating free-form generation tasks and believe our results should transfer to this setting. Further, our experiments do not cover RLHF-finetuned LLMs (Christiano et al., 2017; Ziegler et al., 2019; Ouyang et al., 2022).

## 10 Related Work

Some recent work has studied the effect of labels in ICL. Yoo et al. (2022) also revisit label randomization and find significant variance across tasks and models. Pan et al. (2023) further separate ICL into label-independent and -dependent learning, which they study by replacing labels with arbitrary tokens. Wei et al. (2023) find that smaller or instruction-tuned models are less capable when performing ICL with replacement labels. Similar to Min et al. (2022b), the above studies do not consider probabilistic metrics or full ICL training curves, and thus can underestimate changes in ICL predictions for modified labels. For example, Pan et al. (2023) find that the gap between random and default labels is 'insignificant' for small models, which our results, in particular for probabilistic metrics, contradict, cf. §5. Further, Wei et al. (2023) claim that 'large models can override prior knowledge from pretraining [. . . ] in-context' and 'small models do not change their predictions when seeing flipped labels', which is not supported by our results in §7. Lastly, Gao et al. (2021) observe that replacement labels can also degrade performance when *finetuning* language models. More generally, ICL has been the subject of many recent studies. For example, Min et al. (2022a) fine-tune language models to improve ICL, Si et al. (2023) measure the inductive bias of ICL predictions, Chan et al. (2022b); Dasgupta et al. (2022) study differences between in-weights and in-context generalization, and Chang & Jia (2022); Liu et al. (2022); Zhang et al. (2022b) observe that the selection of examples affect ICL predictions significantly. In this paper, we emphasize a probabilistic treatment of ICL predictions. Uncertainty in LLMs has previously been studied, e.g. by Kadavath et al. (2022); Lin et al. (2023); Bai et al. (2022a); Gonen et al. (2022). On non-language tasks, Kirsch et al. (2022); Chan et al. (2022a) study properties that lead to the emergence of ICL. Also related are Garnelo et al. (2018); Kossen et al. (2021); Gordon et al. (2019), who propose deep non-parametric models on non-language tasks. Unlike ICL, they can guarantee invariance to example-order or closely approximate Bayesian predictive distributions (Müller et al., 2022).

## 11 Conclusions

In this paper, we have investigated how the conditional label distribution of in-context examples affects ICL predictions. To ensure our conclusions represent ICL behavior well, we have studied ICL across all possible in-context dataset sizes and considered probabilistic aspects of ICL predictions. In some sense, we have shown that ICL is both better and worse than expected. On the one hand, our results demonstrate that, against expectations set by prior work, ICL does incorporate in-context label information and can even learn truly novel tasks in-context. On the other hand, we have shown that analogies between ICL and conventional learning algorithms fall short in a variety of ways. In particular, label relationships inferred from pre-training have a lasting effect that cannot be surmounted by in-context observations. Additional prompting can improve but likely not overcome this deficiency. Further, ICL does not treat all information provided in-context equally and preferentially makes use of label information that appears closer to the query.

REPRODUCIBILITY STATEMENT

We discuss the details of our experimental evaluation in §E. We provide the code to reproduce our results at the following repository: github.com/jlko/in_context_learning.

ACKNOWLEDGMENTS

Tom Rainforth is supported by the UK EPSRC grant EP/Y037200/1. Jannik Kossen acknowledges funding from the New College Yeotown Scholarship.

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

## A    CAN PROMPTS HELP ICL LEARN FLIPPED LABEL RELATIONSHIPS?

In this section, we explore if *prompts* can be used to overcome the plateauing ICL performance when default labels are flipped. Before the in-context examples, we insert prompt strings that inform the LLM of the flipped label relationship in some fashion, and should thus help the LLM adjust to it during ICL. These prompts could fundamentally change few-shot ICL behavior. In fact, one can think of the in-context learner as the union of LLM and prompt, where so far the prompt was simply left empty. There could exist prompts that help ICL learn the flipped label relationship better than without them, or as well as in the default scenario. Note that, regardless of the outcome here, NH2 remains rejected as ICL should not need to rely on a prompt to correctly consider in-context observations. Nevertheless, for the 'prompted few-shot ICL' setup, NH2 should be reconsidered.

We explore the following three prompts: 'In the following ...', (Instruct Prompt) '...negative means positive and positive means negative' (here for SST-2 and adapted to other tasks), (Ignore Prompt) '...ignore all prior knowledge', and (Invert Prompt) '...flip the meaning for all answers'.

**Observations & Discussions.** Figure A.1 gives results for the prompted few-shot ICL setup for LLaMa-65B and Falcon-40B on SST-2. Figures F.38 to F.46 give results for our largest models across all tasks. However, prompting is most successful for the scenarios in Fig. A.1, making this the most interesting result to study NH2. In particular, prompting has a surprisingly weak effect for LLaMa-2-70B. In Fig. A.1 we observe that prompts, in particular the instruct and invert prompts, can help improve ICL performance. However, it seems that the positive impact from prompting is restricted to an initial boost at small in-context datasets sizes. We then sometimes observe a 'dip' in performance, which could indicate ICL forgetting about the prompt. At large context sizes, none of our prompts have any advantage, and flipped label performance again plateaus short of performance for the default label setup. ***Therefore, we reject NH2 for the prompted ICL variations that we study.***

It is possible that there exist prompts—that we have not found—for which we cannot reject NH2. However, we are sceptical these prompts exist for the models we study, as their behavior at large context sizes is strikingly similar across all prompts we investigate. For more capable LLMs, we suspect it may be possible to obtain a zero-shot performance on the flipped scenario that is equal to the zero-shot of the default scenario, i.e. the prompt leads to the LLM perfectly flipping all its zero-shot predictions. However, we are unsure if, in addition to flipping zero-shot predictions, such prompts would also improve *ICL* on flipped label observations to be as good as in the default scenario.

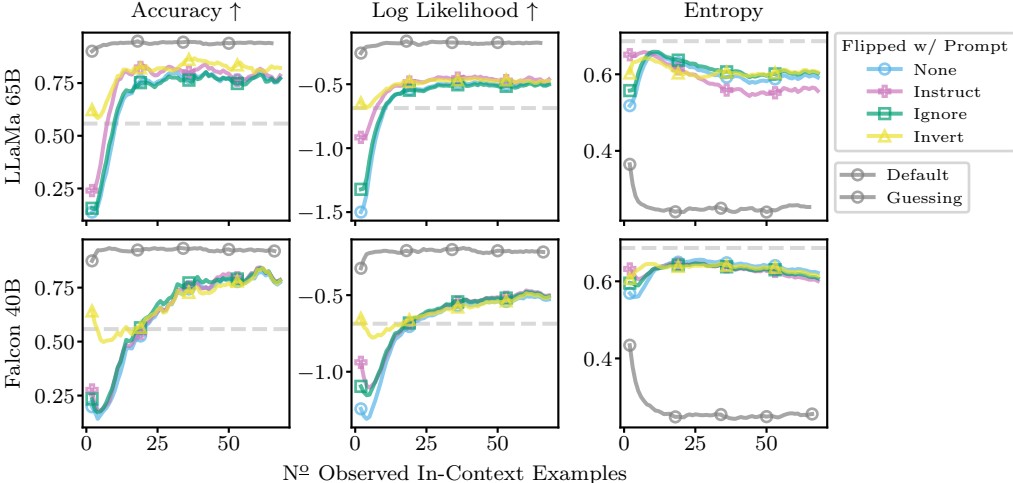

Figure A.1: **Prompted** few-shot ICL with **flipped labels** on **SST-2**. Some prompts are able to improve ICL on flipped labels compared to not using a prompt as before (label none in the figure). However, improvements have no lasting effect: performance at larger context sizes does not improve and still plateaus short of the default scenario (solid grey line). We average over 100 random subsets and then additionally apply moving averages (window size 5) for clarity.

## B  Evaluation Approach for Cheap In-Context Learning Dynamics

In this section, we suggest a—to the best of our knowledge—novel way of evaluating ICL that gives performance metrics at all in-context dataset sizes in a single forward pass without incurring additional cost. We start by introducing the notation necessary to formalize few-shot ICL in LLMs.

**Dataset to Input String.**  The few-shot task is defined by a dataset $\mathcal{D} = \{(S_i, Y_i)\}_{i=1}^N$, where $S_i \in \mathcal{T}^{d_{S_i}}$ are input sentences from which to predict the associated labels $Y_i \in \mathcal{T}^{d_{y_i}}$, and $\mathcal{T}^v$ are text strings of length $v$. A *verbalizer* $V(S, Y)$ takes a sentence–label pair and maps it to an *example*, e.g. the sequence 'I am happy' and label 'positive' are verbalized as 'Sentence: I am happy\n Label: positive\n'. We also define a *query verbalizer* $V_q(S)$ that maps a test query $S$ to a query example, e.g. 'I am sad' is mapped to 'Sentence: I am sad\n Label:', such that the next-token prediction of an LLM will be encouraged to predict the label for the query. We apply the verbalizer to the entire dataset set and concatenate its output to obtain the *context* $\mathcal{C} = \oplus_{i=1}^N V(S_i, Y_i)$. Finally, we concatenate context $\mathcal{C}$ and verbalized query $V_q(S)$, where $S$ is a sentence drawn from a separate test set, to obtain the input to the language model, $I = \mathcal{C} \oplus V_q(S) \in \mathcal{T}^{d_I}$.

**Input String to Tokens.**  The input $I$ is *tokenized* before it can be processed by the language model. The tokenizer, $T(I) = (X_1, \ldots, X_M)$, maps an input sequence $I$ to a sequence of integers, or tokens, $X_i \in (1, \ldots, D)$, where $D$ is the vocabulary size, i.e. the number of unique tokens. We keep track of which token positions correspond to labels, $\mathcal{L} = (l_1, \ldots, l_N)$, e.g. the indices of the tokens immediately following the string 'Label:' in the above example.

**Tokens to Predictions.**  In the following, we use capital letters to denote random variables and lower-case letters for their realizations. Here, we describe the behavior of *decoder-only* language models (Liu et al., 2018; Radford et al., 2018), a popular architecture choice for LLMs. Given the observed sequence of input tokens $(X_1 = x_1, \ldots, X_M = x_M)$, a single forward pass through the language model gives an estimate of the *joint probability*,

$$p(X_1 = x_1) \cdot p(X_2 = x_2 \mid X_1 = x_1) \cdot \ldots \cdot p(X_M = x_M \mid X_1 = x_1, \ldots, X_{M-1} = x_{M-1}). \quad (1)$$

We highglight that Eq. (1) gives the joint probability *at the observed outcomes*: we obtain $M$ 'one-step ahead' predictions, each conditioned only on observed outcomes and not on model predictions. Equation (1) is a common objective in LLM training, where 'the joint probability the model assigns to the observations' is sometimes referred to as *teacher forcing* (Williams & Zipser, 1989).

At test time, LLMs are usually iteratively conditioned on their own *predictions*, generating novel outputs via multiple forward passes, i.e. one first samples $\hat{x}_M \sim p(X_M | \ldots)$, and then $\hat{x}_{M+1} \sim p(X_{M+1} | \ldots, X_M = \hat{x}_M)$, and so on. We here use '$\ldots$' to stand in for any additional tokens also conditioned on, e.g. $(x_1, \ldots, x_{M-1})$. One usually ignores all other terms of the joint here—the predictions for $(X_1, \ldots, X_{M-1})$ that are generated in each forward pass—as only the last term $p(X_M | \ldots)$ is needed to sample the next token, i.e. the label in standard few-shot ICL applications.

**Single-Forward Pass ICL Training Dynamics.**  We now explain our approach for efficient evaluation of ICL training dynamics. Given input tokens $(X_1, \ldots, X_M)$ for the few-shot ICL setup described above, we first select those terms from Eq. (1) that correspond to label token predictions,

$$\prod_{i=1}^N p(X_{l_i} = x_{l_i} \mid X_1 = x_1, \ldots, X_{m<l_i} = x_{m<l_i}). \quad (2)$$

For each term, the model predicts a distribution over the entire token vocabulary, i.e. $p(X_{l_i} | \ldots)$ is a categorical distribution, $p = (p_1, \ldots, p_D)$, which is then evaluated at the observed tokens in Eq. (2). We can transform this into a prediction over only the few-shot task label $Y$ by selecting the indices of the categorical distribution which correspond to the tokenized labels and then renormalizing, $p(Y) = (p_{t_1}, \ldots, p_{t_C} | \ldots)$, where $C$ is the number of unique labels which are encoded to tokens $(t_1, \ldots, t_C)$. With this, we can rewrite Eq. (2) as the joint probability the model assigns to the sequence of labels given input sentences

$$p(Y_1 = y_1 \mid S_1 = s_1) \cdot p(Y_2 = y_2 \mid S_1 = s_1, Y_1 = y_1, S_2 = s_2) \cdot \ldots$$
$$\cdot p(Y_N = y_N \mid S_1 = s_1, Y_1 = y_1, \ldots, S_{N-1} = s_{N-1}, Y_{N-1} = y_{N-1}, S_N = s_N). \quad (3)$$

Note how, because the joint is evaluated at the observations, its individual terms are always conditioned on the true labels and not previous predictions. This allow us to cheaply compute the *training*

*dynamics* of ICL as a function of increasing in-context dataset size. With each forward pass, we obtain the individual terms of Eq. (3), which are the few-shot ICL predictions at all possible context dataset sizes, $i = (1, \ldots, N)$. In contrast, in standard few-shot ICL evaluations, each forward pass only yields predictions for a single test query, neglecting the information the joint contains about the first $N - 1$ label predictions. There may be interesting applications of Eq. (3) to model selection, as the quantity has links to both Bayesian evidence (Murphy, 2022) and cross-validation (Fong & Holmes, 2020), although we do not explore this any further in this paper.

**Multi-Token Labels.** So far, we have assumed that each label string is encoded as a single token. However, our approach can also be applied if some or all labels are encoded as multiple tokens. In essence, we continue to measure only the probability the model assigns to the first token of each label, making the (fairly harmless) assumption that the first (or only) token that each label is encoded to is unique among labels. We believe this is justified, as, given the first token for a label, the model should near-deterministically predict the remaining tokens, i.e. all the predictive information is contained in the first token the model predicts for a label. For example, for the Subjectivity dataset, the label 'objective' is encoded by the LLaMa tokenizer as a single token but the label 'subjective' is encoded as two tokens, [subject, ive]. We only use the probability assigned to [subject] to assign probabilities to 'subjective', and ignore any predictions for [ive]. However, if the model successfully accommodates the pattern of the in-context example labels, we would expect probabilities for [ive] following [subject] to be close to 1 always.

For LLaMa-7B on Subjectivity, we have investigated the above assumption empirically. After the first observation of the 'subjectivity' label in-context, the probability of predicting [ive] after observing [subject] are $0.9998 \pm 0.0003$ for the following 12 instances of the 'subjectivity' label, with probabilities normalized over *all* tokens of the vocabulary here. In other words, we can safely evaluate the performance of the LLaMa model from its predictions of only the [subject] token, even though the full label is split over two tokens [subject, ive].

## C  AUTHORSHIP IDENTIFICATION TASK

We here give details on our novel authorship identification task.

**Data Collection & Processing.** We extract the last 151 messages sent between two authors of this paper on the Slack messaging platform. If multiple messages are sent in a row by one author, these count as multiple inputs. We filter out 42 messages that were of low quality: URLs, article summaries, missed call notifications, and meeting notes. This leaves us with 58 and 51 messages per author. We set the maximum message length to be 200 and truncate any messages longer than that. Before truncation, the longest message was 579 characters long. The median message length is 68 before and after truncation, mean and standard deviation shrink from $100 \pm 98$ to $88 \pm 65$. For use in ICL, we treat this dataset as we would any other and present messages in random order.

**Data Release.** For now, we have decided to not make this dataset public for two reasons: (1) It contains genuinely private communication, and (2) releasing the data would mean that future LLMs might be trained on it, so we could no longer use it to test their ability to learn truly novel label relationships in-context. However, below, we give 6 random examples from the dataset:

Author 1  Would 10.30am on Tuesday work?
Author 1  Sounds good. When are you back again?
Author 1  Yeah, we might have to find somewhere else depending on whether my office mates are in, but its out of term so should be plenty of free meeting rooms if needed
Author 2  No problem!
Author 2  I'll be in [redacted] next week, so do you think we can meet in person?
Author 2  The vacation is 16 days!

## D  DATASET CITATIONS

We evaluate on SST-2 (Socher et al., 2013), Subjective (Wang & Manning, 2012), Financial Phrasebank (Malo et al., 2014), Hate Speech (de Gibert et al., 2018), AG News (Zhang et al., 2015), Medical Questions Pairs (MQP) (McCreery et al., 2020), as well as Microsoft Research Paraphrase Corpus (MRPC) (Dolan & Brockett, 2005), Recognizing Textual Entailment (RTE) (Dagan et al., 2005), and Winograd Schema Challenge (WNLI) (Levesque et al., 2012) from GLUE (Wang et al., 2019).

# E  EXPERIMENT DETAILS

Below we give additional details on our experimental evaluation.

**Guessing Baseline.** In our experiments, we frequently display a 'guessing based on class frequencies' baseline as a grey-dashed line. This baseline presents an *informed guess* that relies only on knowing the class frequencies of the task and makes the exact same prediction for each input datapoint. We here explain how we compute this baseline for accuracy, entropy, and log likelihood. We are given a classification task with $C$ classes which appear with frequencies $p = [p_1, \ldots, p_C]$ in the training set. The baseline always predicts $p = [p_1, \ldots, p_C]$, i.e. it predicts the class frequencies. For accuracy, it thus always predicts the majority class $c^* = \arg\max_k p_k$, which leads to accuracy $p_{c^*}$. Further, the baseline prediction leads to a log likelihood of $\sum_k p_k \log p_k$ and an entropy of $-\sum_k p_k \log p_k$ under the training data distribution.

**Class Flipping.** While most of our tasks are *binary* classification, Financial Phrasebank and AG News are not. For these datasets, when 'flipping' labels in §7, §A, and §8, we actually rotate labels instead, i.e. we reassign labels $y$ as $y \leftarrow (y+1) \mod C$, where $C$ is the number of classes. For AG News, ['world', 'sports', 'business', 'science and technology'] get mapped to ['sports', 'business', 'science and technology', 'world']. For Financial Phrasebank, ['negative', 'neutral', 'positive'] get mapped to ['neutral', 'positive', 'negative']. Note that, for Financial Phrasebank, rotating the labels is harder than naively inverting the label order, as rotating does not leave the meaning of the 'neutral' label unchanged.

**In-Context Example Formatting.** We use the following simple templates to format the in-context examples. For SST-2, Subjectivity, Financial Phrasebank, Hate Speech, and our author identification task, we use the following line of Python code to format each input example:
`f"Sentence: '{sentence}'\nAnswer: {label}\n\n"`.
For MRPC, WNLI, and RTE, we format instances with
`f"Sentence 1: '{sentence1}'\nSentence 2: '{sentence2}'\nAnswer: {label}\n\n"`.
For MQP, we use
`f"Question 1: '{sentence1}'\nQuestion 2: '{sentence2}'\nAnswer: {label}\n\n"`.

**Implementation.** We rely on the Hugging Face Python library (Wolf et al., 2020) and PyTorch (Paszke et al., 2019) to implement the experiments of this paper. We use half-precision floating-point numbers for LLaMa-65B and LLaMa-2-70B, and we use 8 bit-quantization for all other models, which we have found to not affect performance notably. In Fig. E.1, we illustrate this by showing the difference between 8 bit quantization and full 32 bit precision for default ICL and ICL with label randomization for LLaMa-2-7B on the Subjectivity dataset: there is no significant loss of precision or change in behavior from 8 bit quantization.

**Datasets.** We use Hugging Face Spaces to access all tasks considered in this paper. For Hate Speech, we select the first 1000 examples with labels 0 and 1, skipping datapoints with labels 2 and 3. We do not use custom processing for any other dataset.

**Whitespace Tokenization.** To evaluate few-shot ICL performance as introduced in §4, we need to identify the tokens that individual task labels are encoded to. We here detail how to achieve this at the example of the SST-2 label 'positive'. In particular, we highlight the, perhaps unexpected, effects of whitespaces on input tokenization. These details are important and, if not considered correctly, can degrade performance significantly.

For Falcon models, the tokenizer encodes 'Answer:' as [20309, 37], 'Answer:-' as [20309, 37, 204], and 'Answer:-positive' as [20309, 37, 3508]. We here use dashes '-' instead of whitespaces for improved legibility. Clearly, the relevant token for the label 'positive' is [3508]. Note how the token [204] for the trailing whitespace disappears again after appending the label. Further, just encoding 'positive' without a preceding whitespace gives [28265]. However, this token does not appear in the input, where the label is preceded by a whitespace—we should thus use the token [3508] to measure ICL performance.

The LLaMa and LLaMa-2 tokenizer encodes 'Answer:' as [673, 29901], 'Answer:-' as [673, 29901, 29871], and 'Answer:-positive' as [673, 29901, 6374]. Thus,

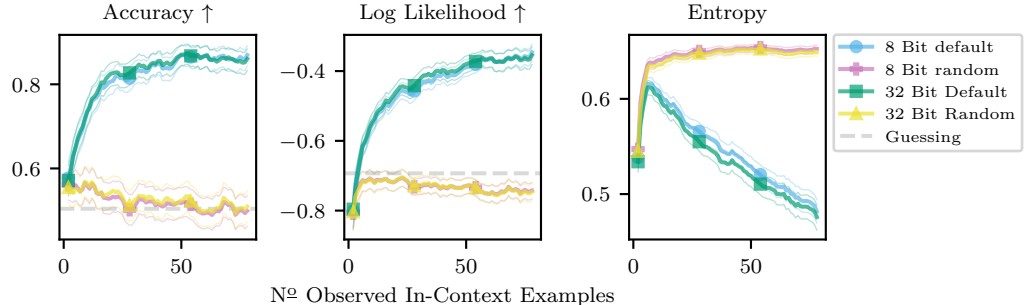

Figure E.1: Few-shot ICL at 8 bit and 32 bit precision with default and randomized labels for LLaMa-2-7B on Subjectivity. There is no significant performance degradation from 8 bit quantization. We average over 500 random in-context datasets, thin lines are bootstrapped 99 % confidence intervals, and we apply moving averages (window size 5) for clarity.

Table E.1: Maximum number of in-context examples we consider for each model-task combination. Below, we shorten AG News as AGN, Hate Speech as HS, and Financial Phrasebank as FP.

|          | SST-2 | Subj | FP | HS | AGN | MQP | MRPC | RTE | WNLI |
|----------|-------|------|----|----|-----|-----|------|-----|------|
| LLaMa-2  | 140   | 79   | 73 | 76 | 45  | 47  | 40   | 28  | 57   |
| LLaMa    | 66    | 37   | 33 | 26 | 21  | 21  | 20   | 13  | 27   |
| Falcon   | 67    | 39   | 37 | 28 | 25  | 23  | 21   | 15  | 30   |

the relevant token for the label 'positive' is `[6374]`. In contrast to the Falcon tokenizer, just encoding `positive` without a preceding whitespace also gives `[6374]`.

Lastly, similar caveats apply to the classic evaluation procedure for few-shot ICL, where we only evaluate the prediction for a single test query at the end of the input. Here, it is crucially important that we do not end inputs with a trailing whitespace. As we have seen above, for both LLaMa and Falcon tokenizers, the trailing whitespace leads to the generation of an extra token that is not present when encoding complete in-context examples, as the whitespace would usually be included in the label prediction itself. This change in tokenization between in-context examples and test query can adversely affect ICL performance.

**Statistical Significance.** In Tables 1, 2, F.1, and F.2 we bold differences if they are statistically significant at a 95 % level. Concretely, we compute if the absolute average differences are larger than 1.96 times the standard error. Similarly, when deciding if default label performance is significantly better than random guessing performance, we check if mean performance plus 1.645 times the standard error is larger than the guessing baseline across accuracy and log likelihood.

**Maximum Context Dataset Size.** For each task, we create in-context datasets by sub-sampling from the training set of the task. Falcon and LLaMa support input sizes up to 2048 tokens, and LLaMa-2 supports up to 4096 input tokens. For all models, performance will degrade if the input size exceeds this limit. This caps the number of in-context examples we can include for each task. For tasks where the individual input sentences are longer, we will be able to include fewer examples in-context. Across all in-context datasets that we sample for a task, we compute the minimum number of in-context examples needed to exceed the token limit of 2048 or 4096. This is the maximum in-context dataset size up to which we report results for that task. We list these numbers in Table E.1

**Calibration.** We do not calibrate predicted probabilities by first dividing them by a 'prior' probability and then renormalizing as suggested by Zhao et al. (2021). We have found this rarely improves, and sometimes degrades predictions, cf. Fig. E.2. We observe this happening in particular for tasks where labels are encoded as multiple tokens.

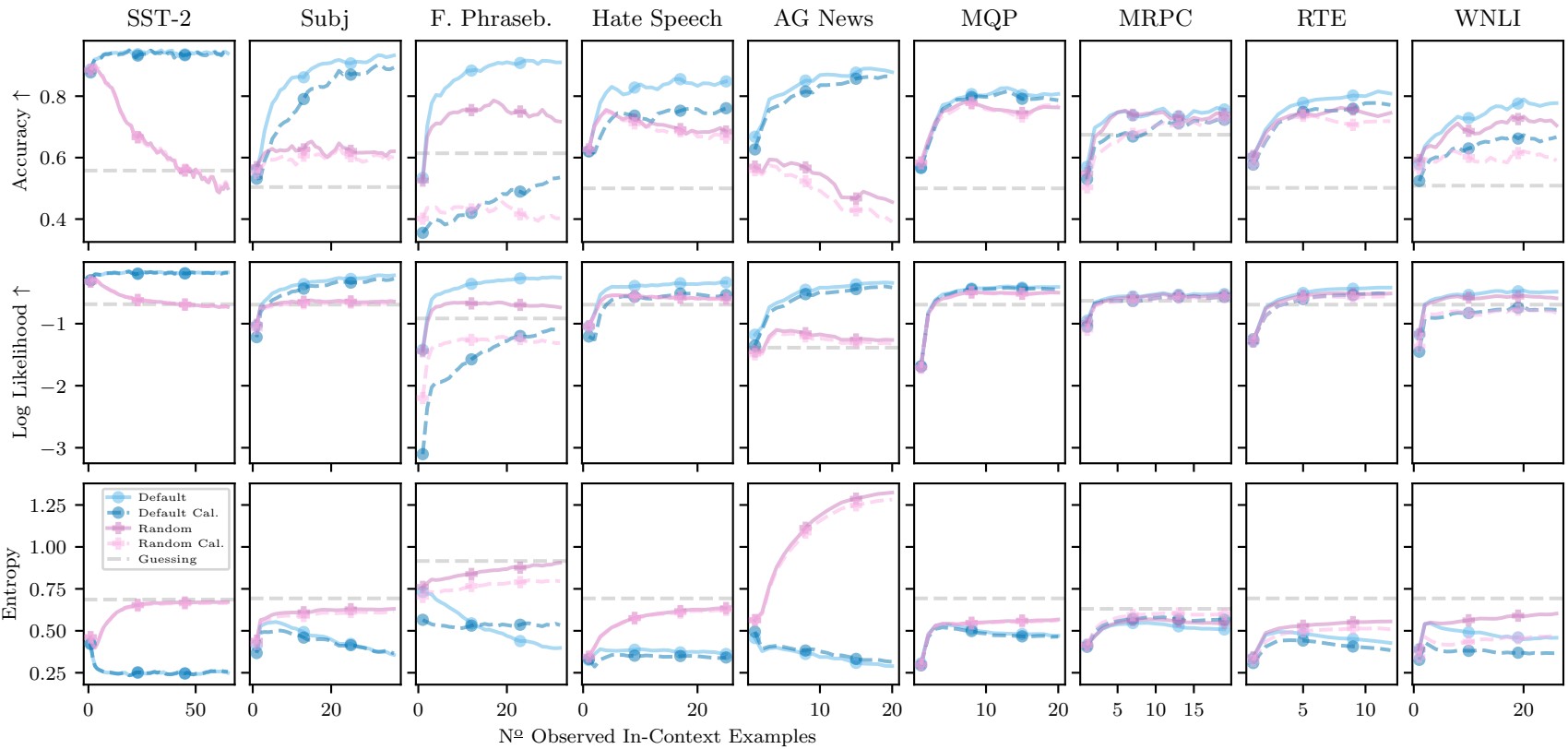

Figure E.2: Few-shot ICL with randomized labels for **LLaMa-65B**. We additionally display performance for random and default labels when 'calibrating' the predicted probabilities as suggested by Zhao et al. (2021). We do not find calibration helpful to improve ICL performance. We average over 500 random in-context datasets and thin lines are bootstrapped 99 % confidence intervals and we apply a moving average (window size 3) for clarity.

## F EXTENDED RESULTS

**Section 4 – Training Dynamics**: Figure F.1 shows few-shot ICL training dynamics on SST-2 for a selection of models at different parameter counts.

**Section 5 – Label Randomization**: Figure F.2 gives results for randomized labels for all models on SST2. Figures F.3 to F.12 give results for all tasks and models comparing ICL with randomized labels to the default label setup. Table F.1 gives full summary statistics across all models, tasks, and metrics for the label randomization experiment.

**Section 6 – Author ID Task**: Figure F.13 gives few-shot ICL results for all models on our novel authorship identification task.

**Section 7 – Flipped Labels**: Figures F.14 to F.31 give results for all tasks and models for the modified label relationship experiments. We also report performance for additional replacement labels across task and models. Table F.2 gives full summary statistics across all models, tasks, and metrics for the difference between default and flipped label performance.

**Section 8 – Dynamic Label Flipping**: In Figs. F.32 to F.36, we give results for the experiments investigating NH3 for all large models on tasks where label flipping gave strong performance in §7. For LLaMa-2-70B on Hate Speech in Fig. F.35, metrics appear very similar initially. However, when exploring additional changepoints in Fig. F.37, we do find significant differences in predictions.

**Appendix A – Prompting with Flipped Labels**: Figures F.38 to F.46 give results for all tasks and models for the prompted few-shot ICL setup.

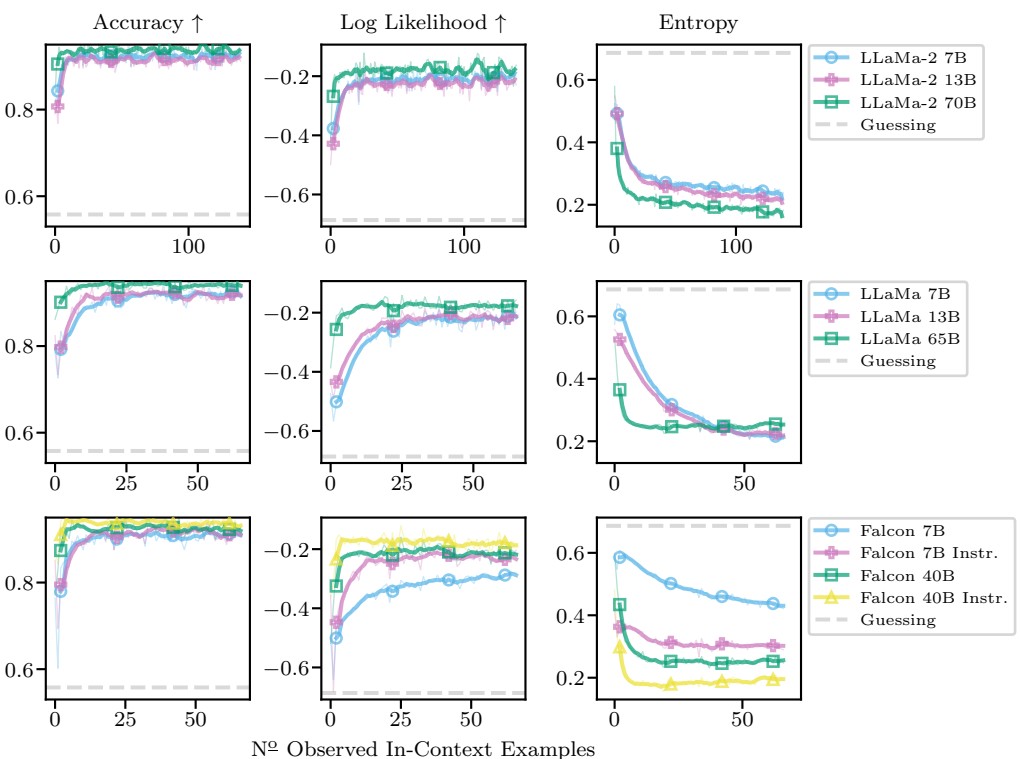

Figure F.1: Few-shot ICL **training dynamics** in a standard scenario on **SST-2**. Accuracy (↑) and log likelihood (↑) improve with in-context dataset size, and entropies decrease appropriately. Averages over 500 random subsets of the SST-2 training set (thin lines), additionally applying a moving average with window size 5 for clarity (thick lines).

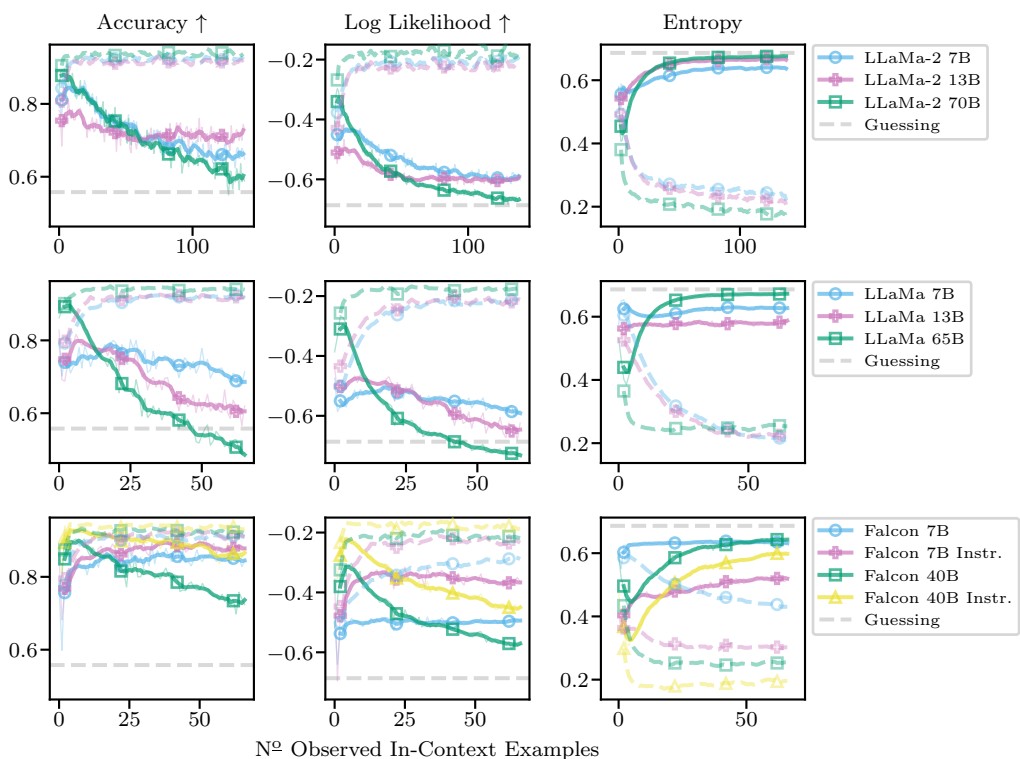

Figure F.2: Few-shot ICL with **randomized labels** for **SST-2**: Compared to default ICL behavior (dashed lines, cf. Fig. F.1), log likelihoods and entropies of the models degrade when in-context example labels are randomized. Thin lines are averages over 500 repetitions, thick lines with moving average (window size 5), guessing baseline based on class frequencies.

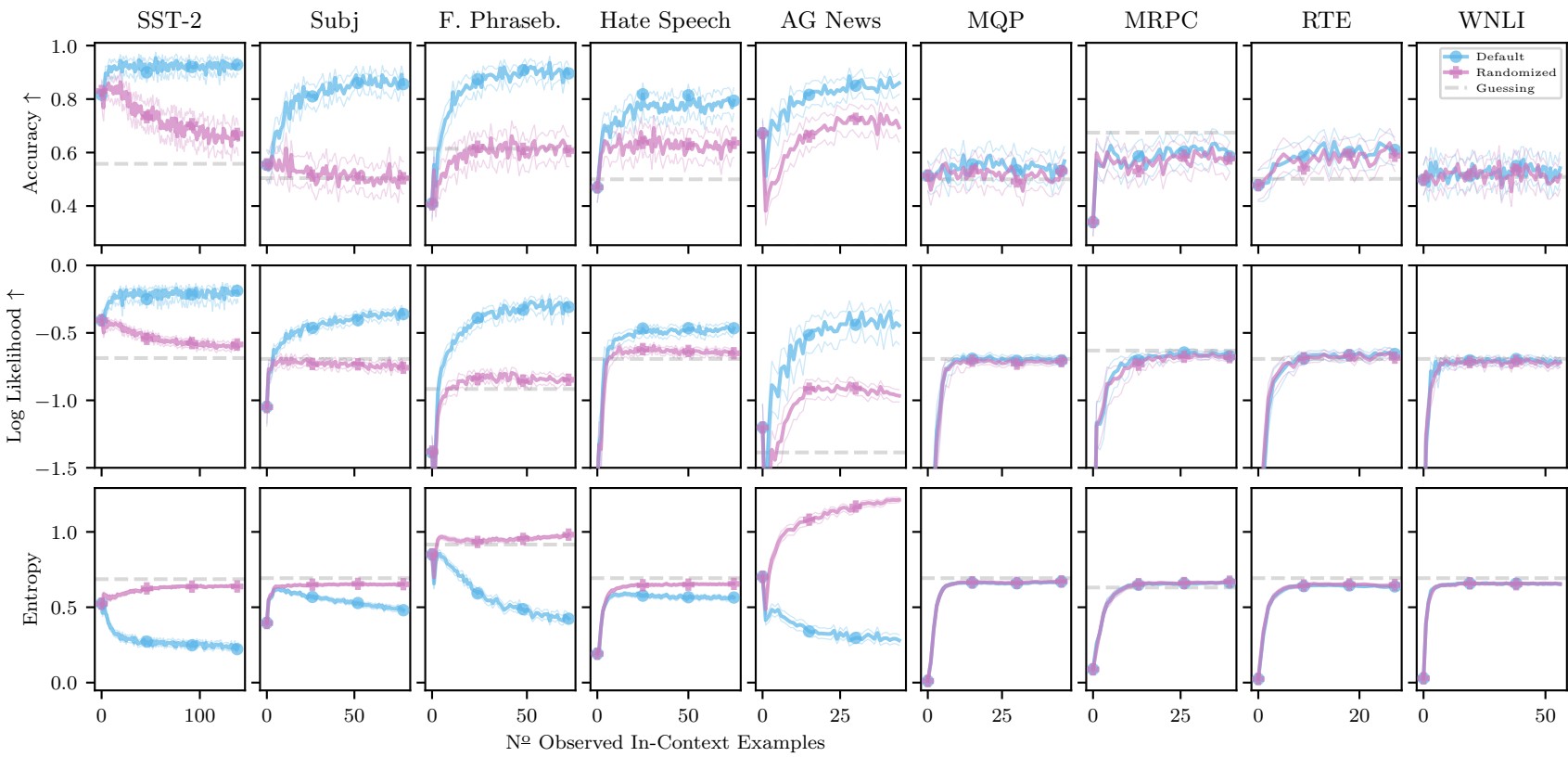

Figure F.3: Few-shot ICL with **randomized labels** for **LLaMa-2-7B**: accuracies, entropies, and log likelihoods behave differently for randomized and default labels when performance is above randomly guessing class frequencies. While accuracy can be noisy, differences are clearly visible for probabilistic entropy and log likelihood. We average over 500 random in-context datasets, thin lines are bootstrapped 99 % confidence intervals.

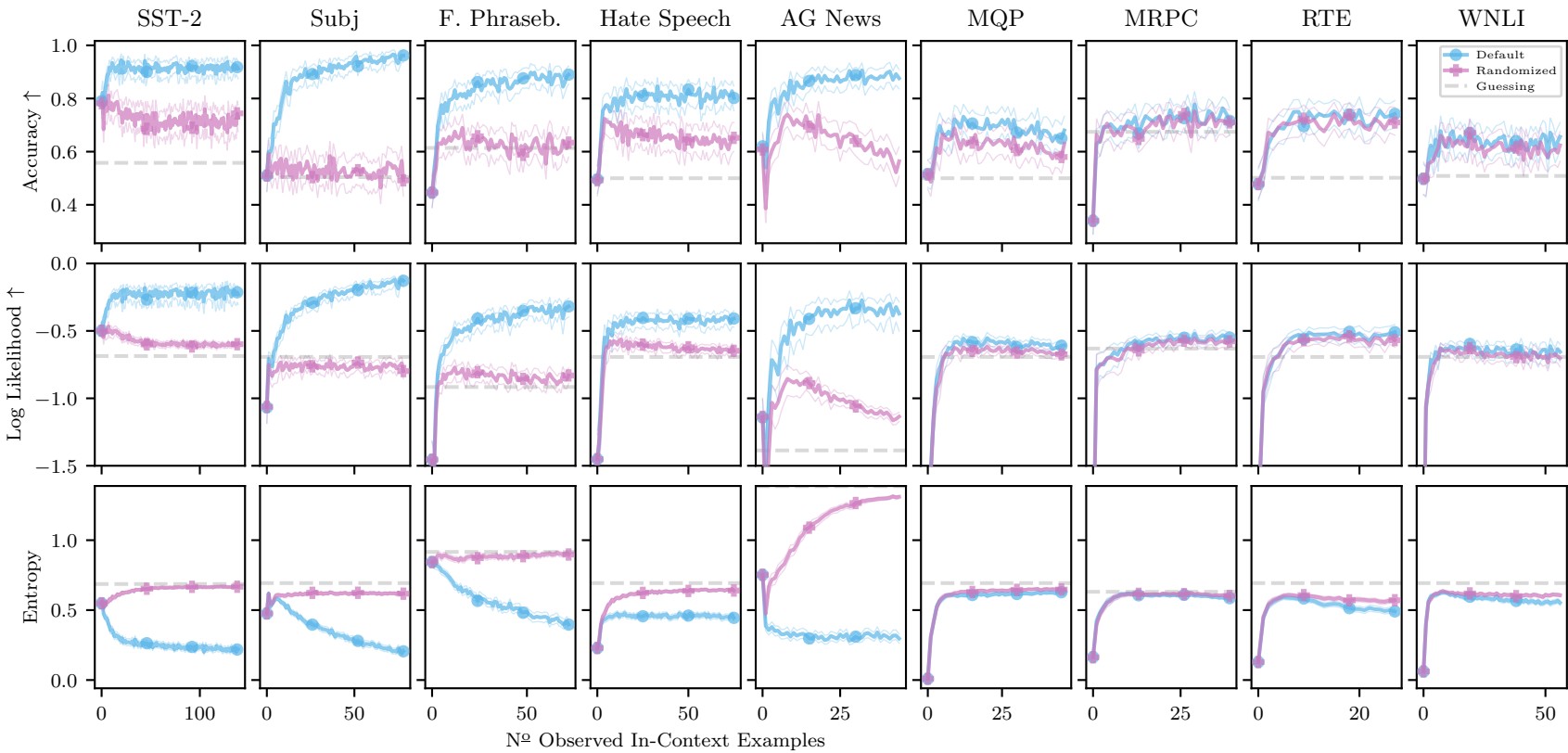

Figure F.4: Few-shot ICL with **randomized labels** for **LLaMa-2-13B**: accuracies, entropies, and log likelihoods behave differently for randomized and default labels when performance is above randomly guessing class frequencies. While accuracy can be noisy, differences are clearly visible for probabilistic entropy and log likelihood. We average over 500 random in-context datasets, thin lines are bootstrapped 99 % confidence intervals.

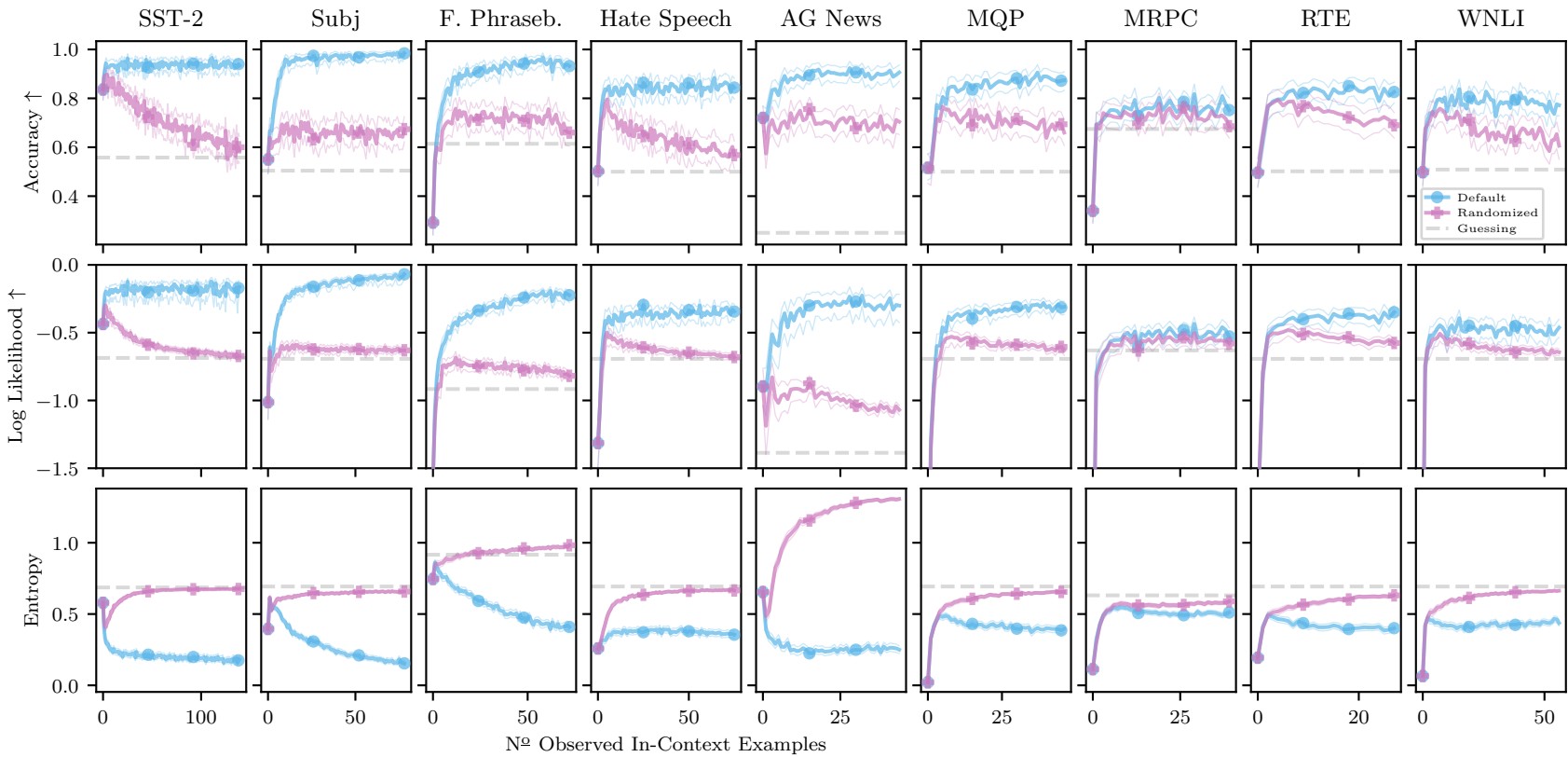

Figure F.5: Few-shot ICL with **randomized labels** for **LLaMa-2-70B**: accuracies, entropies, and log likelihoods behave differently for randomized and default labels when performance is above randomly guessing class frequencies. While accuracy can be noisy, differences are clearly visible for probabilistic entropy and log likelihood. We average over 500 random in-context datasets, thin lines are bootstrapped 99 % confidence intervals.

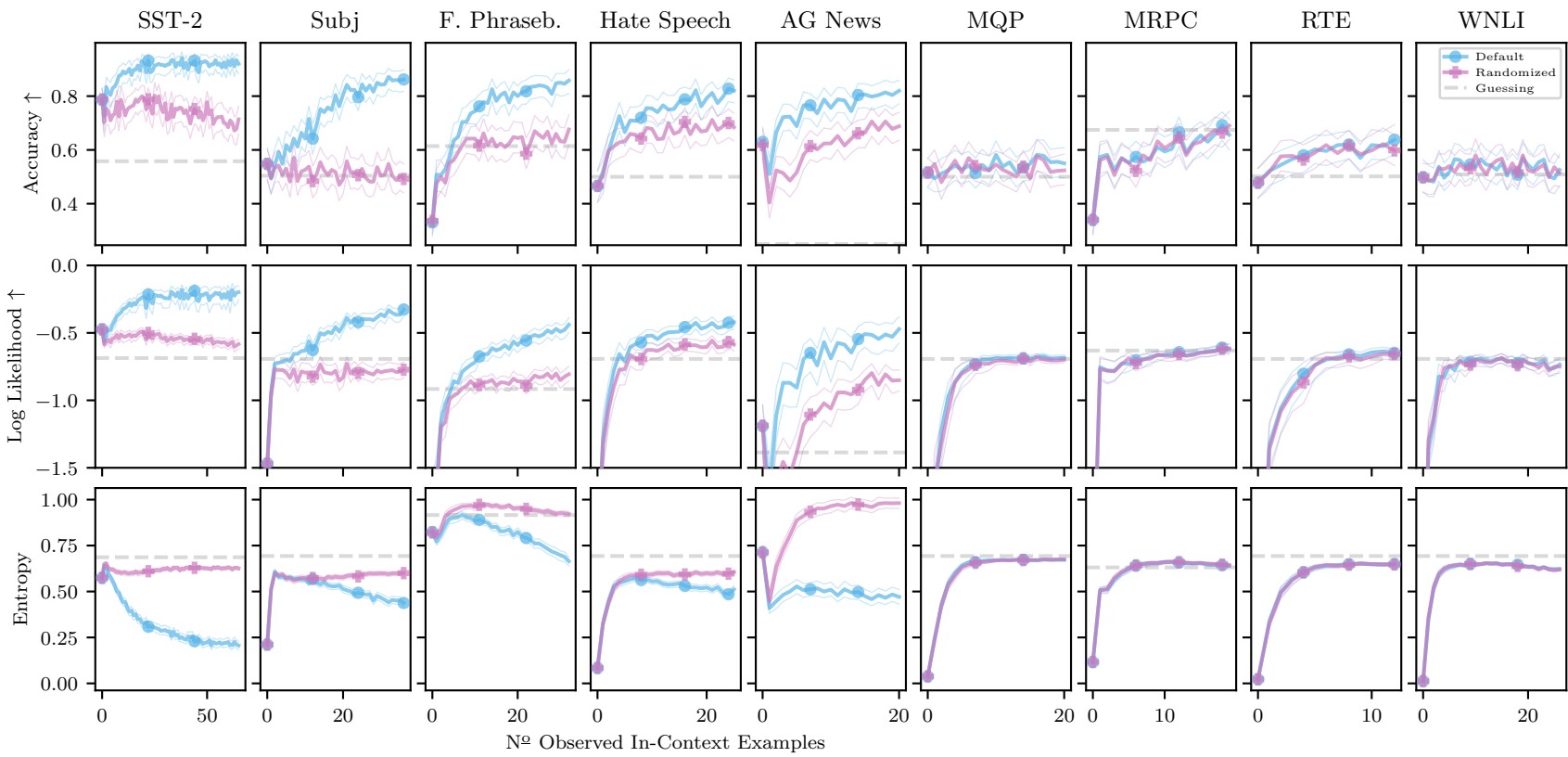

Figure F.6: Few-shot ICL with **randomized labels** for **LLaMa-7B**: accuracies, entropies, and log likelihoods behave differently for randomized and default labels when performance is above randomly guessing class frequencies. While accuracy can be noisy, differences are clearly visible for probabilistic entropy and log likelihood. We average over 500 random in-context datasets, thin lines are bootstrapped 99 % confidence intervals.

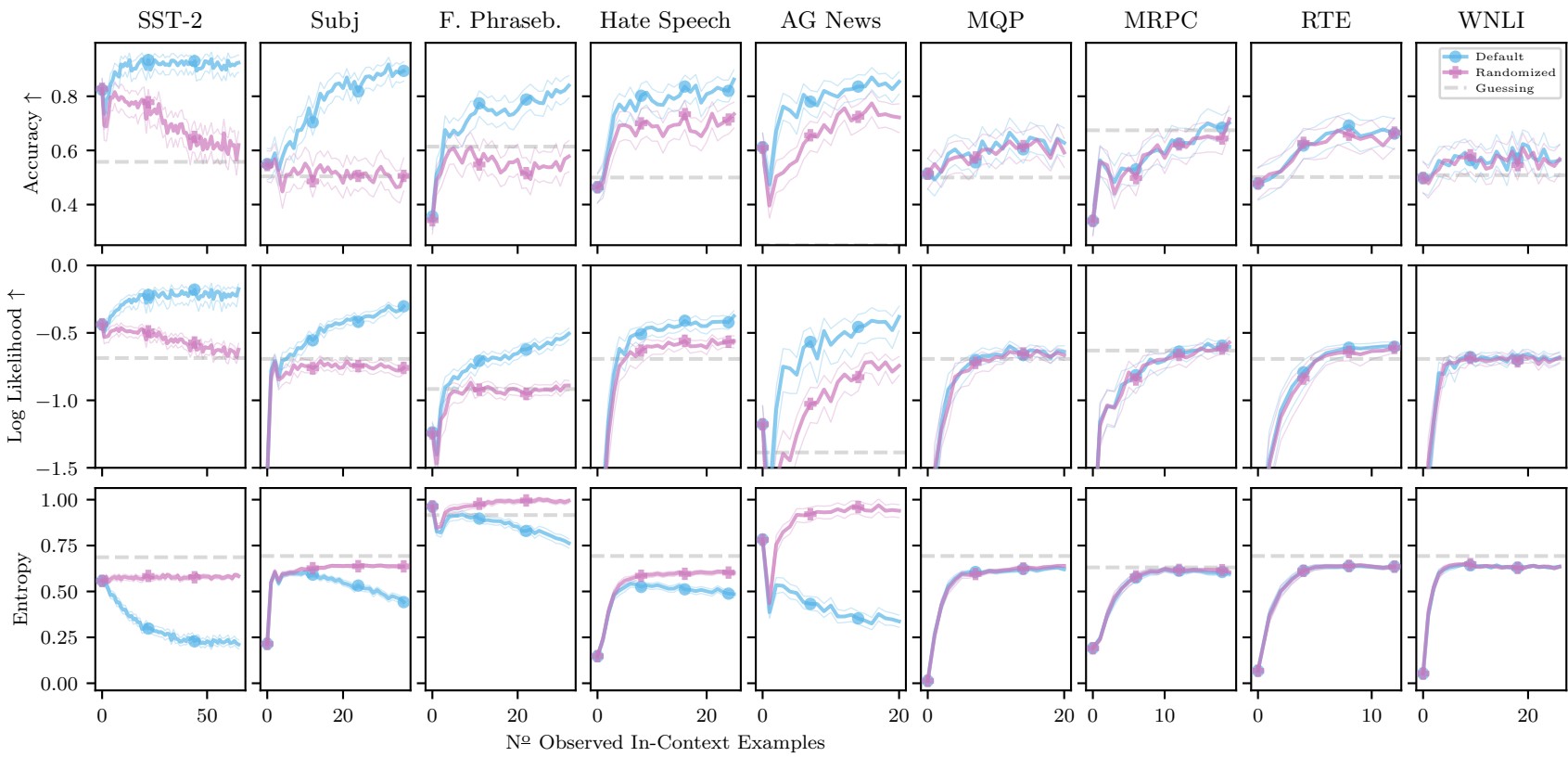

Figure F.7: Few-shot ICL with **randomized labels** for **LLaMa-13B**: accuracies, entropies, and log likelihoods behave differently for randomized and default labels when performance is above randomly guessing class frequencies. While accuracy can be noisy, differences are clearly visible for probabilistic entropy and log likelihood. We average over 500 random in-context datasets, thin lines are bootstrapped 99 % confidence intervals.

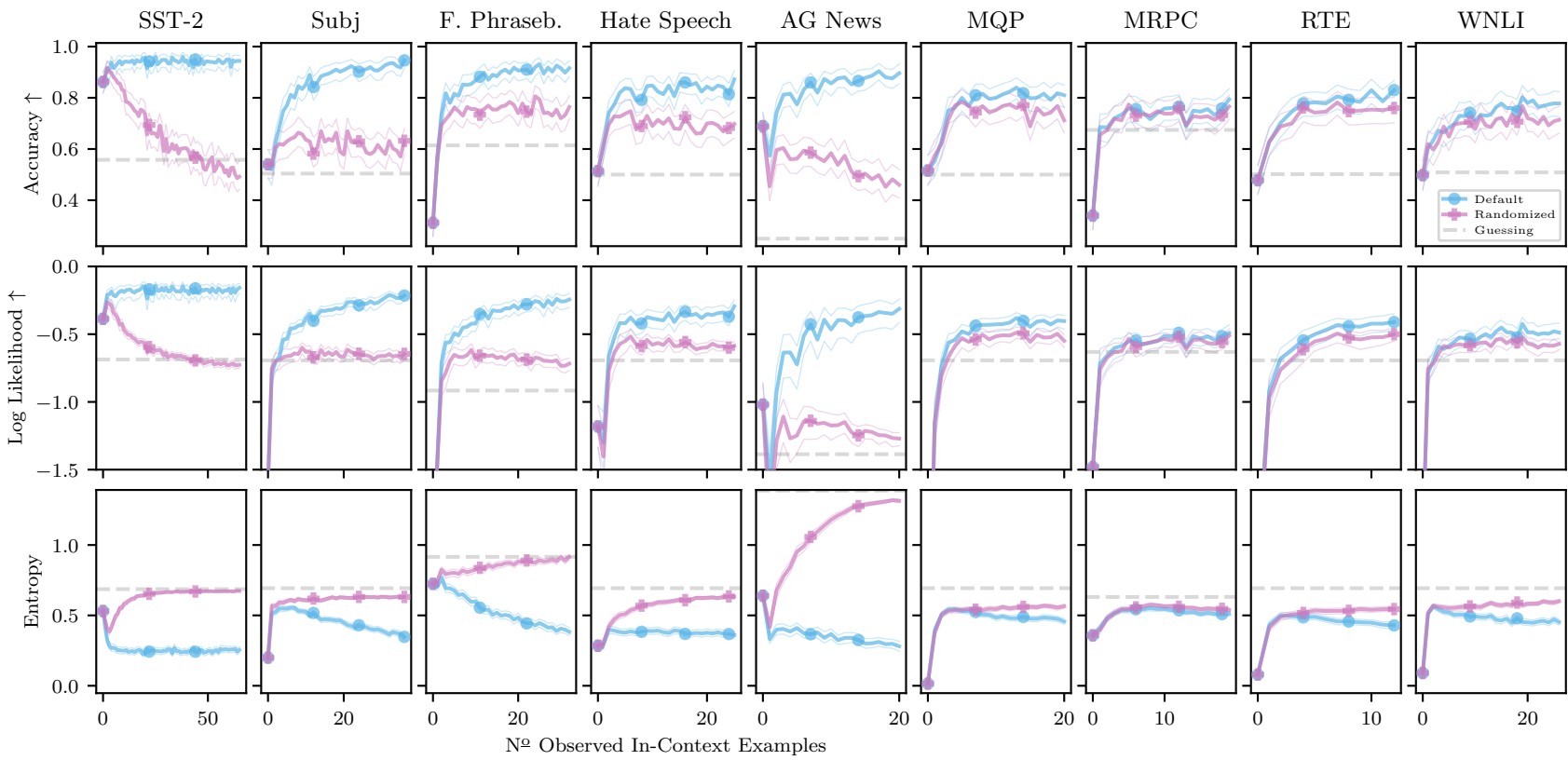

Figure F.8: Few-shot ICL with **randomized labels** for **LLaMa-65B**: accuracies, entropies, and log likelihoods behave differently for randomized and default labels when performance is above randomly guessing class frequencies. While accuracy can be noisy, differences are clearly visible for probabilistic entropy and log likelihood. We average over 500 random in-context datasets, thin lines are bootstrapped 99 % confidence intervals.

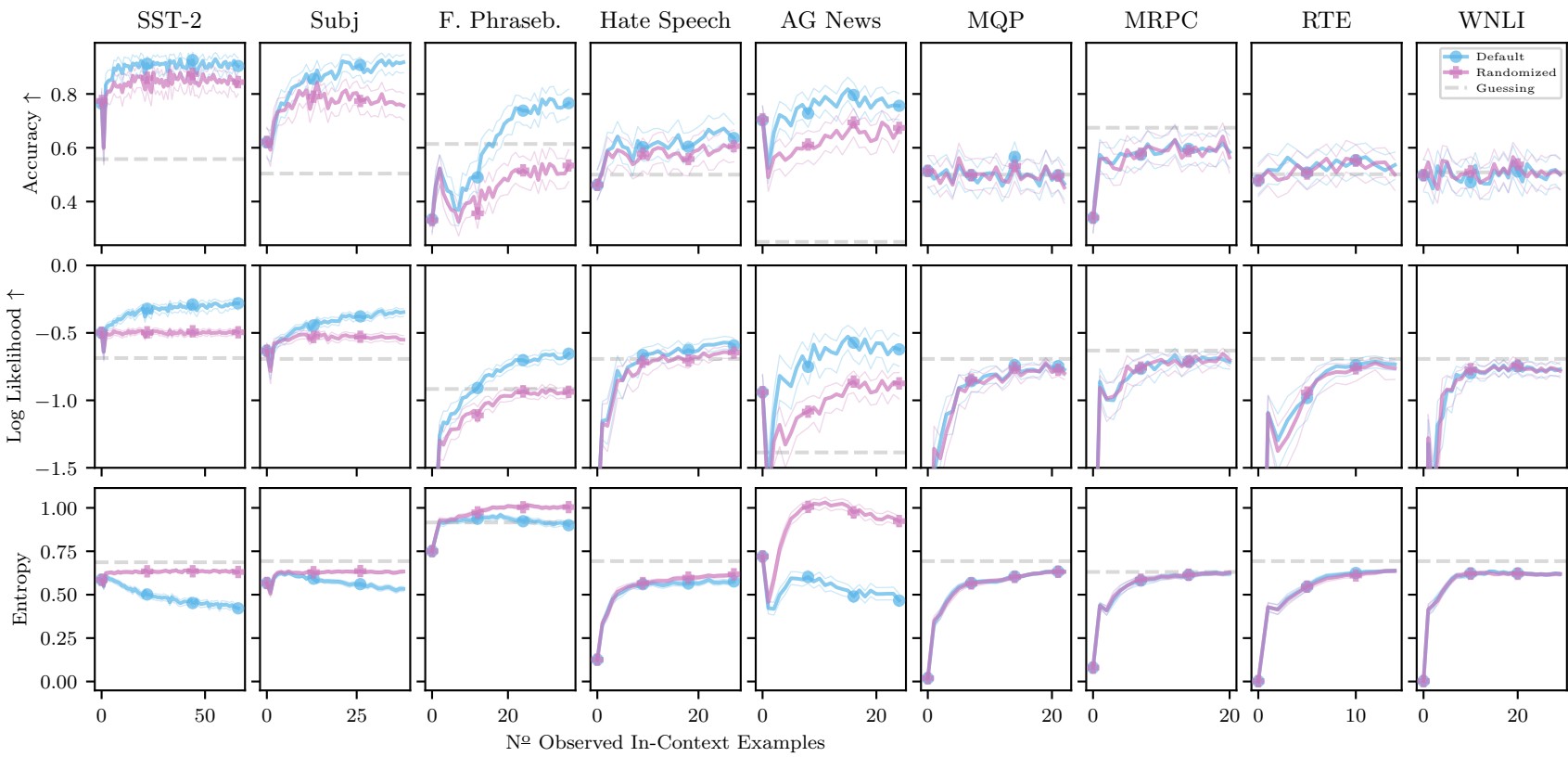

Figure F.9: Few-shot ICL with **randomized labels** for **Falcon-7B**: accuracies, entropies, and log likelihoods behave differently for randomized and default labels when performance is above randomly guessing class frequencies. While accuracy can be noisy, differences are clearly visible for probabilistic entropy and log likelihood. We average over 500 random in-context datasets, thin lines are bootstrapped 99 % confidence intervals.

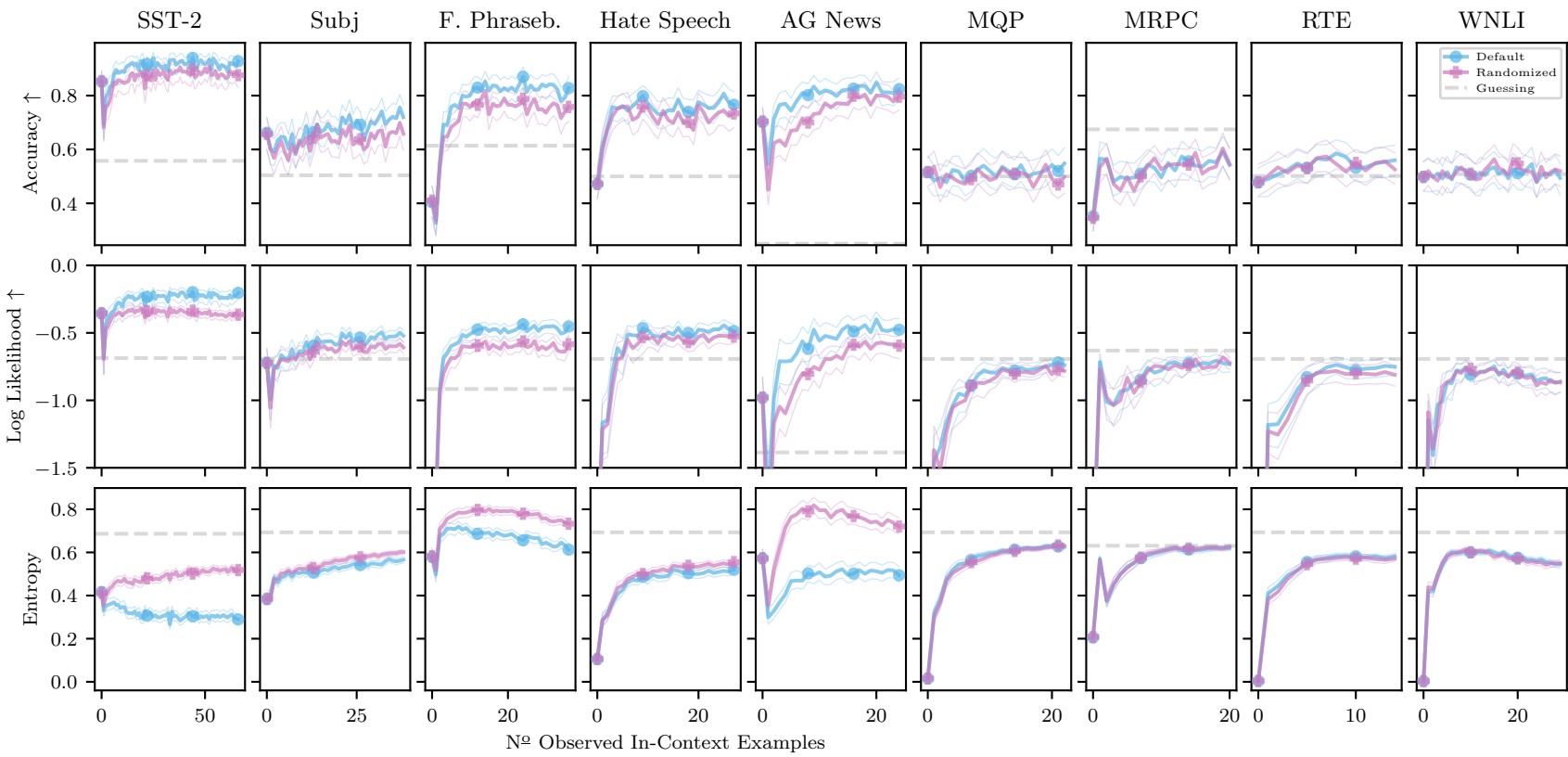

Figure F.10: Few-shot ICL with **randomized labels** for **Falcon-7B-Instruct**: accuracies, entropies, and log likelihoods behave differently for randomized and default labels when performance is above randomly guessing class frequencies. While accuracy can be noisy, differences are clearly visible for probabilistic entropy and log likelihood. We average over 500 random in-context datasets, thin lines are bootstrapped 99 % confidence intervals.

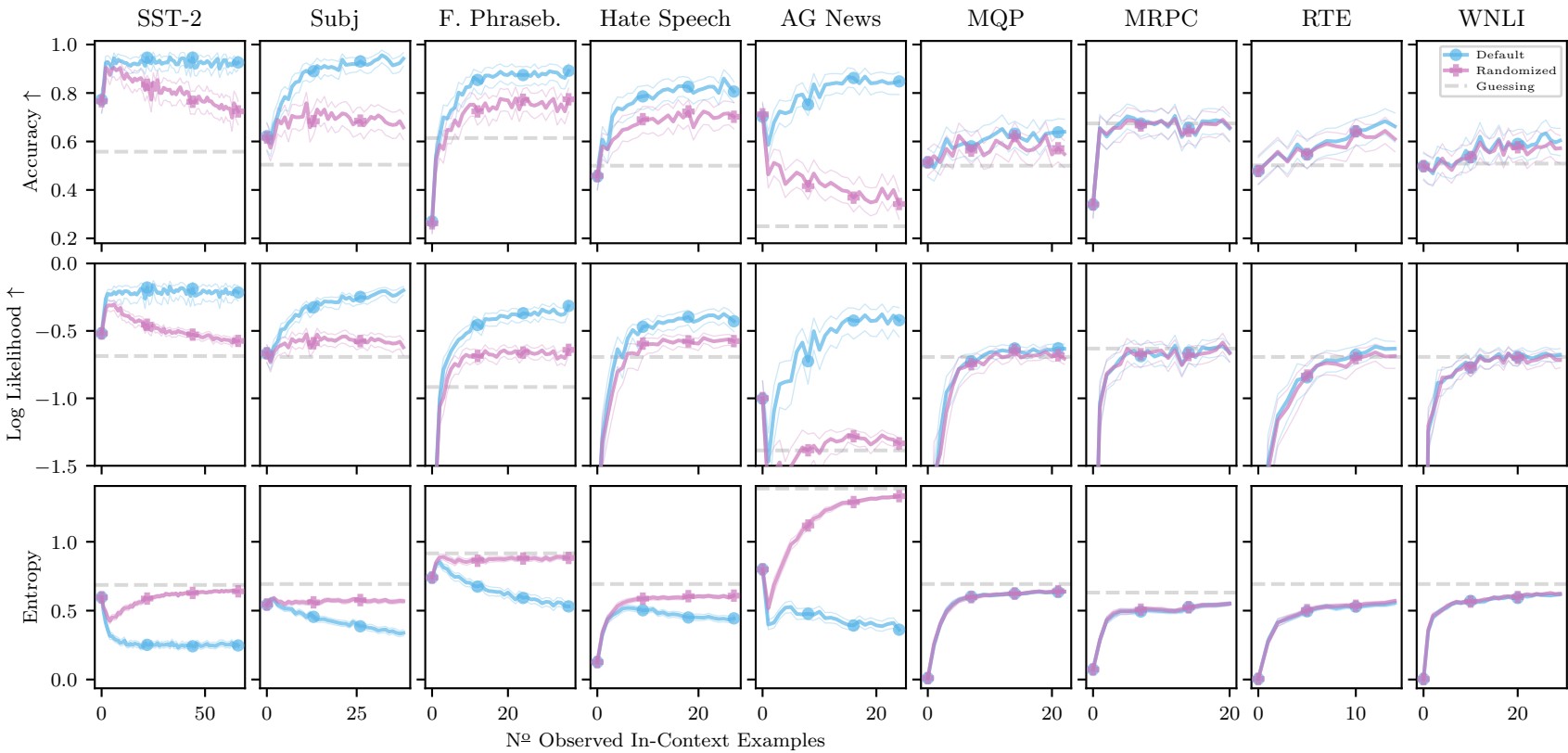

Figure F.11: Few-shot ICL with **randomized labels** for **Falcon-40B**: accuracies, entropies, and log likelihoods behave differently for randomized and default labels when performance is above randomly guessing class frequencies. While accuracy can be noisy, differences are clearly visible for probabilistic entropy and log likelihood. We average over 500 random in-context datasets, thin lines are bootstrapped 99 % confidence intervals.

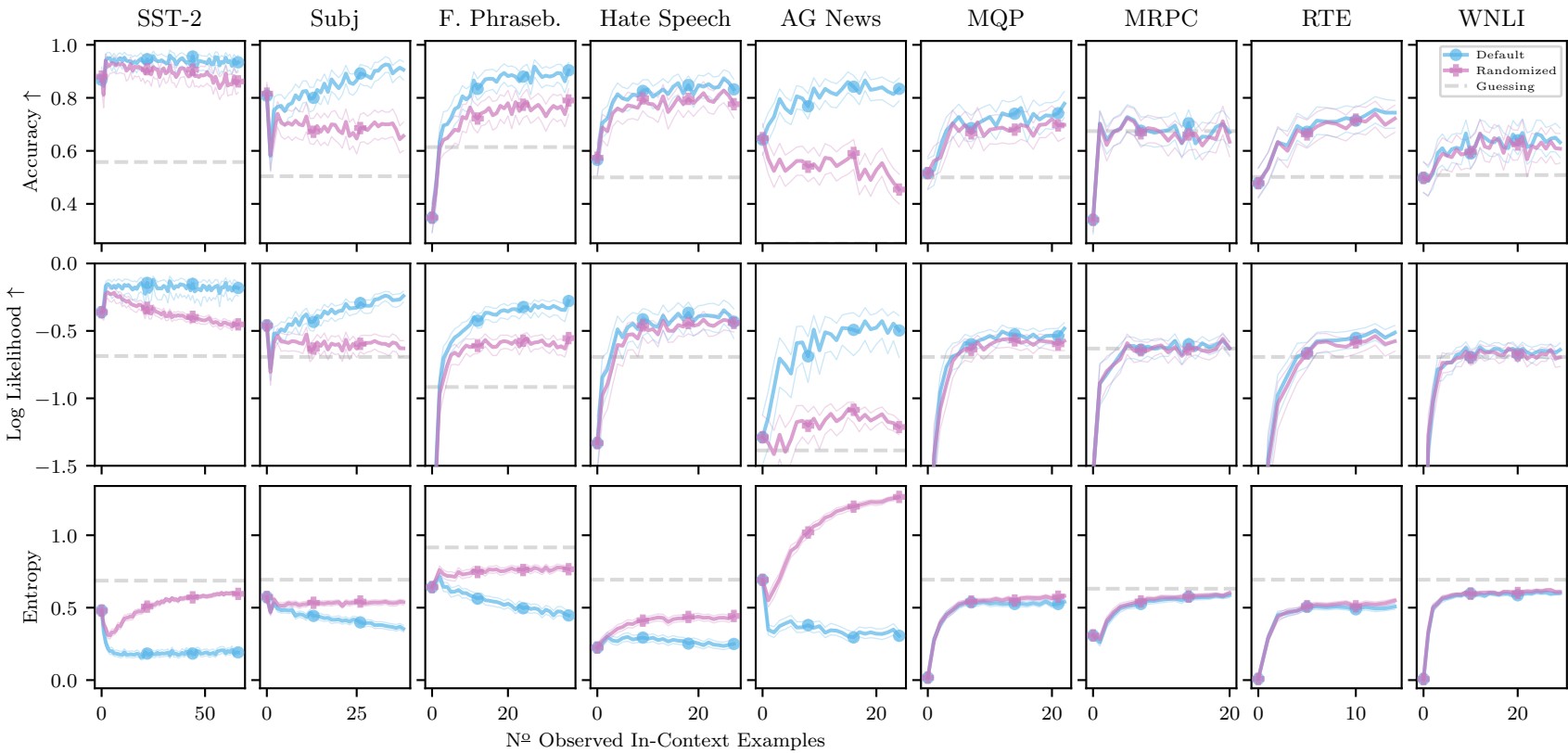

Figure F.12: Few-shot ICL with **randomized labels** for **Falcon-40B-Instruct**: accuracies, entropies, and log likelihoods behave differently for randomized and default labels when performance is above randomly guessing class frequencies. While accuracy can be noisy, differences are clearly visible for probabilistic entropy and log likelihood. We average over 500 random in-context datasets, thin lines are bootstrapped 99 % confidence intervals.

Table F.1: Full summary statistics for the randomization experiment of §5. We strongly encourage readers to also view the full training curves across all possible context sizes in §F. Across all metrics, here show the average difference between the default and randomized label scenario. We compute 'Metric(Default) - Metric(Random)', such that positive accuracy/log likelihood differences indicate that ICL performs worse with randomized labels. We compute differences at the maximum context size for each task-model combination (cf.§E and Table E.1). We display experiments where ICL accuracies or log likelihoods on the default labels do not significantly exceed random guessing performance in lightgray. Across models, and metrics, model performance is usually significantly worse when labels are randomized and default performance exceeds random guessing. We display mean differences and standard errors over 500 runs. We bold entries for which mean differences are statistically significant.

| Δ Log Lik. | SST-2 | Subj | F. Phraseb. | Hate Speech | AG News | MQP | MRPC | RTE | WNLI |
|---|---|---|---|---|---|---|---|---|---|
| LLaMa-2 7B | $0.42 \pm 0.02$ | $0.39 \pm 0.02$ | $0.57 \pm 0.02$ | $0.18 \pm 0.01$ | $0.53 \pm 0.04$ | $0.03 \pm 0.01$ | $0.02 \pm 0.01$ | $0.03 \pm 0.01$ | $0.02 \pm 0.01$ |
| LLaMa-2 13B | $0.41 \pm 0.02$ | $0.62 \pm 0.03$ | $0.49 \pm 0.03$ | $0.24 \pm 0.02$ | $0.81 \pm 0.04$ | $0.04 \pm 0.01$ | $0.01 \pm 0.01$ | $0.06 \pm 0.02$ | $0.02 \pm 0.03$ |
| LLaMa-2 70B | $0.51 \pm 0.03$ | $0.53 \pm 0.02$ | $0.57 \pm 0.02$ | $0.34 \pm 0.02$ | $0.80 \pm 0.03$ | $0.29 \pm 0.02$ | $0.04 \pm 0.02$ | $0.22 \pm 0.02$ | $0.18 \pm 0.02$ |
| LLaMa 7B | $0.39 \pm 0.03$ | $0.42 \pm 0.03$ | $0.36 \pm 0.02$ | $0.15 \pm 0.02$ | $0.30 \pm 0.03$ | $0.03 \pm 0.01$ | $0.00 \pm 0.01$ | $0.03 \pm 0.02$ | $0.01 \pm 0.02$ |
| LLaMa 13B | $0.44 \pm 0.03$ | $0.45 \pm 0.02$ | $0.37 \pm 0.02$ | $0.16 \pm 0.02$ | $0.32 \pm 0.03$ | $0.04 \pm 0.02$ | $-0.01 \pm 0.02$ | $0.05 \pm 0.02$ | $0.02 \pm 0.02$ |
| LLaMa 65B | $0.55 \pm 0.02$ | $0.45 \pm 0.02$ | $0.49 \pm 0.02$ | $0.23 \pm 0.02$ | $0.88 \pm 0.04$ | $0.14 \pm 0.02$ | $0.01 \pm 0.01$ | $0.12 \pm 0.02$ | $0.08 \pm 0.02$ |
| Falcon 7B | $0.20 \pm 0.01$ | $0.19 \pm 0.01$ | $0.25 \pm 0.01$ | $0.06 \pm 0.01$ | $0.31 \pm 0.03$ | $0.01 \pm 0.02$ | $0.01 \pm 0.02$ | $-0.01 \pm 0.02$ | $0.01 \pm 0.02$ |
| Falcon 7B Instr. | $0.13 \pm 0.01$ | $0.08 \pm 0.01$ | $0.11 \pm 0.01$ | $0.03 \pm 0.02$ | $0.15 \pm 0.02$ | $0.03 \pm 0.02$ | $0.02 \pm 0.03$ | $-0.00 \pm 0.02$ | $0.00 \pm 0.02$ |
| Falcon 40B | $0.34 \pm 0.02$ | $0.35 \pm 0.02$ | $0.31 \pm 0.02$ | $0.18 \pm 0.02$ | $0.90 \pm 0.04$ | $0.06 \pm 0.02$ | $0.01 \pm 0.02$ | $0.01 \pm 0.02$ | $0.02 \pm 0.02$ |
| Falcon 40B Instr. | $0.25 \pm 0.02$ | $0.37 \pm 0.03$ | $0.27 \pm 0.02$ | $0.02 \pm 0.03$ | $0.77 \pm 0.04$ | $0.06 \pm 0.02$ | $0.02 \pm 0.02$ | $0.02 \pm 0.02$ | $0.04 \pm 0.02$ |

| Δ Accuracy | SST-2 | Subj | F. Phraseb. | Hate Speech | AG News | MQP | MRPC | RTE | WNLI |
|---|---|---|---|---|---|---|---|---|---|
| LLaMa-2 7B | $28.4 \pm 2.1$ | $39.8 \pm 2.4$ | $30.4 \pm 2.4$ | $16.2 \pm 2.3$ | $13.4 \pm 2.2$ | $3.8 \pm 2.2$ | $5.2 \pm 2.8$ | $4.0 \pm 2.4$ | $3.0 \pm 2.5$ |
| LLaMa-2 13B | $19.8 \pm 2.0$ | $46.0 \pm 2.4$ | $24.4 \pm 2.4$ | $19.0 \pm 2.3$ | $34.6 \pm 2.3$ | $3.2 \pm 2.5$ | $2.8 \pm 1.4$ | $1.6 \pm 1.9$ | $3.6 \pm 2.3$ |
| LLaMa-2 70B | $34.8 \pm 2.3$ | $29.8 \pm 2.1$ | $27.0 \pm 2.2$ | $28.6 \pm 2.5$ | $24.4 \pm 2.1$ | $19.6 \pm 2.5$ | $3.6 \pm 1.7$ | $14.0 \pm 2.0$ | $13.6 \pm 2.5$ |
| LLaMa 7B | $24.6 \pm 2.2$ | $36.4 \pm 2.5$ | $19.8 \pm 2.1$ | $10.8 \pm 2.2$ | $11.2 \pm 2.0$ | $4.6 \pm 2.8$ | $-0.2 \pm 1.9$ | $2.8 \pm 2.6$ | $1.0 \pm 2.0$ |
| LLaMa 13B | $31.4 \pm 2.2$ | $42.4 \pm 2.5$ | $26.4 \pm 2.7$ | $13.8 \pm 2.0$ | $9.2 \pm 1.8$ | $4.0 \pm 2.6$ | $-2.6 \pm 2.4$ | $2.0 \pm 2.4$ | $3.0 \pm 2.5$ |
| LLaMa 65B | $47.4 \pm 2.5$ | $34.4 \pm 2.3$ | $18.4 \pm 2.1$ | $13.6 \pm 2.2$ | $39.2 \pm 2.7$ | $8.4 \pm 2.0$ | $2.0 \pm 1.4$ | $7.6 \pm 1.8$ | $4.6 \pm 2.1$ |
| Falcon 7B | $6.8 \pm 1.6$ | $14.4 \pm 2.0$ | $24.0 \pm 2.4$ | $7.6 \pm 1.8$ | $13.2 \pm 2.0$ | $3.0 \pm 2.2$ | $2.6 \pm 2.4$ | $-1.4 \pm 2.9$ | $0.0 \pm 2.1$ |
| Falcon 7B Instr. | $3.6 \pm 1.4$ | $7.6 \pm 1.8$ | $5.2 \pm 1.5$ | $2.0 \pm 2.0$ | $8.2 \pm 1.9$ | $1.6 \pm 2.6$ | $3.0 \pm 2.9$ | $-1.0 \pm 2.3$ | $2.8 \pm 1.8$ |
| Falcon 40B | $20.2 \pm 2.1$ | $22.2 \pm 2.1$ | $10.0 \pm 1.8$ | $10.6 \pm 2.2$ | $49.4 \pm 2.4$ | $11.2 \pm 2.4$ | $-0.4 \pm 1.1$ | $0.4 \pm 1.7$ | $1.2 \pm 2.2$ |
| Falcon 40B Instr. | $6.8 \pm 1.4$ | $21.8 \pm 2.2$ | $10.4 \pm 1.8$ | $0.8 \pm 1.6$ | $40.4 \pm 2.4$ | $4.2 \pm 2.0$ | $3.4 \pm 2.3$ | $3.2 \pm 1.9$ | $3.6 \pm 2.2$ |

| Δ Entropy | SST-2 | Subj | F. Phraseb. | Hate Speech | AG News | MQP | MRPC | RTE | WNLI |
|---|---|---|---|---|---|---|---|---|---|
| LLaMa-2 7B | $-0.465 \pm 0.007$ | $-0.177 \pm 0.008$ | $-0.529 \pm 0.012$ | $-0.099 \pm 0.006$ | $-0.910 \pm 0.014$ | $-0.004 \pm 0.002$ | $-0.005 \pm 0.002$ | $-0.014 \pm 0.004$ | $0.004 \pm 0.003$ |
| LLaMa-2 13B | $-0.499 \pm 0.008$ | $-0.429 \pm 0.009$ | $-0.477 \pm 0.013$ | $-0.204 \pm 0.008$ | $-1.009 \pm 0.013$ | $-0.018 \pm 0.004$ | $-0.021 \pm 0.004$ | $-0.079 \pm 0.006$ | $-0.058 \pm 0.006$ |
| LLaMa-2 70B | $-0.564 \pm 0.007$ | $-0.500 \pm 0.007$ | $-0.563 \pm 0.010$ | $-0.313 \pm 0.010$ | $-1.046 \pm 0.011$ | $-0.242 \pm 0.009$ | $-0.074 \pm 0.006$ | $-0.246 \pm 0.009$ | $-0.220 \pm 0.008$ |
| LLaMa 7B | $-0.426 \pm 0.009$ | $-0.153 \pm 0.009$ | $-0.225 \pm 0.011$ | $-0.103 \pm 0.007$ | $-0.497 \pm 0.013$ | $-0.002 \pm 0.001$ | $-0.001 \pm 0.003$ | $-0.002 \pm 0.004$ | $-0.001 \pm 0.004$ |
| LLaMa 13B | $-0.376 \pm 0.009$ | $-0.193 \pm 0.009$ | $-0.218 \pm 0.009$ | $-0.112 \pm 0.007$ | $-0.567 \pm 0.015$ | $-0.008 \pm 0.004$ | $-0.014 \pm 0.005$ | $-0.012 \pm 0.004$ | $0.001 \pm 0.004$ |
| LLaMa 65B | $-0.422 \pm 0.009$ | $-0.283 \pm 0.008$ | $-0.502 \pm 0.011$ | $-0.273 \pm 0.009$ | $-1.032 \pm 0.013$ | $-0.100 \pm 0.007$ | $-0.040 \pm 0.005$ | $-0.128 \pm 0.008$ | $-0.153 \pm 0.008$ |
| Falcon 7B | $-0.196 \pm 0.007$ | $-0.109 \pm 0.006$ | $-0.099 \pm 0.006$ | $-0.046 \pm 0.005$ | $-0.428 \pm 0.015$ | $-0.000 \pm 0.004$ | $0.003 \pm 0.005$ | $-0.002 \pm 0.004$ | $-0.002 \pm 0.004$ |
| Falcon 7B Instr. | $-0.207 \pm 0.007$ | $-0.038 \pm 0.004$ | $-0.115 \pm 0.007$ | $-0.041 \pm 0.006$ | $-0.200 \pm 0.011$ | $0.000 \pm 0.004$ | $0.007 \pm 0.006$ | $0.003 \pm 0.006$ | $-0.003 \pm 0.006$ |
| Falcon 40B | $-0.382 \pm 0.009$ | $-0.248 \pm 0.009$ | $-0.358 \pm 0.010$ | $-0.175 \pm 0.008$ | $-0.923 \pm 0.015$ | $-0.002 \pm 0.004$ | $0.001 \pm 0.006$ | $-0.008 \pm 0.005$ | $-0.001 \pm 0.004$ |
| Falcon 40B Instr. | $-0.396 \pm 0.008$ | $-0.177 \pm 0.009$ | $-0.313 \pm 0.010$ | $-0.202 \pm 0.009$ | $-0.922 \pm 0.015$ | $-0.039 \pm 0.006$ | $-0.013 \pm 0.007$ | $-0.033 \pm 0.006$ | $-0.012 \pm 0.005$ |

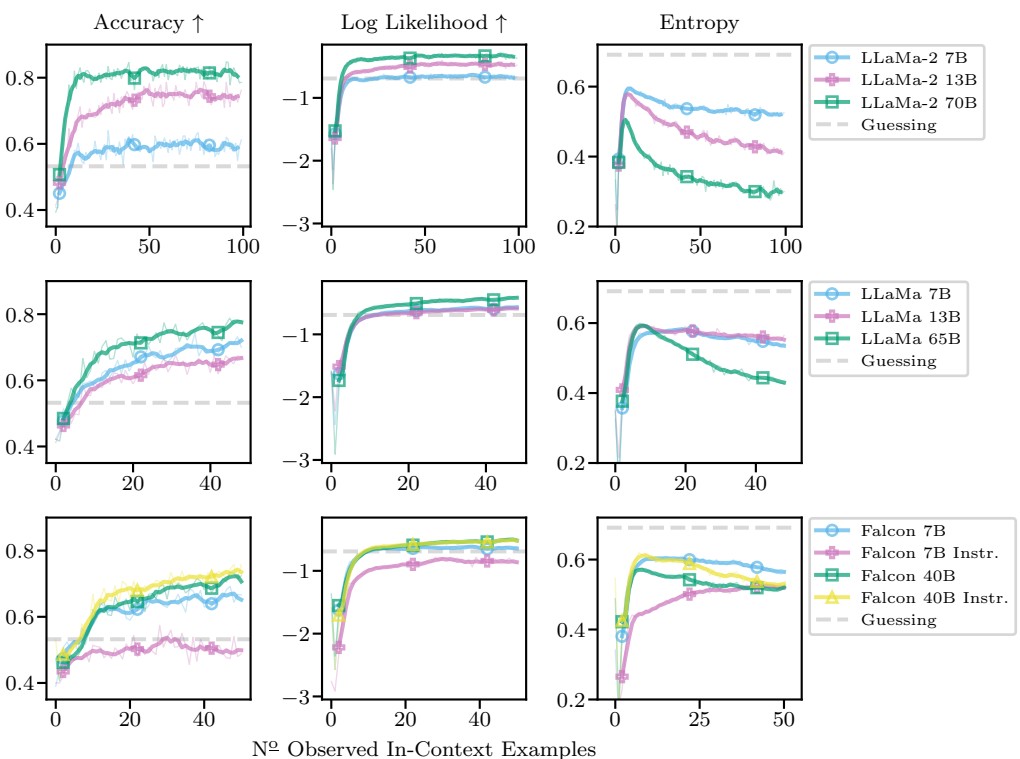

Figure F.13: Few-shot ICL on **novel author identification** task for all models. Models achieves accuracies significantly better than random guessing on our **novel author identification** task. Thus, ICL predictions must depend on the label relationship provided in-context. Thin lines are averages over 500 runs, thick lines moving average (window size 5).

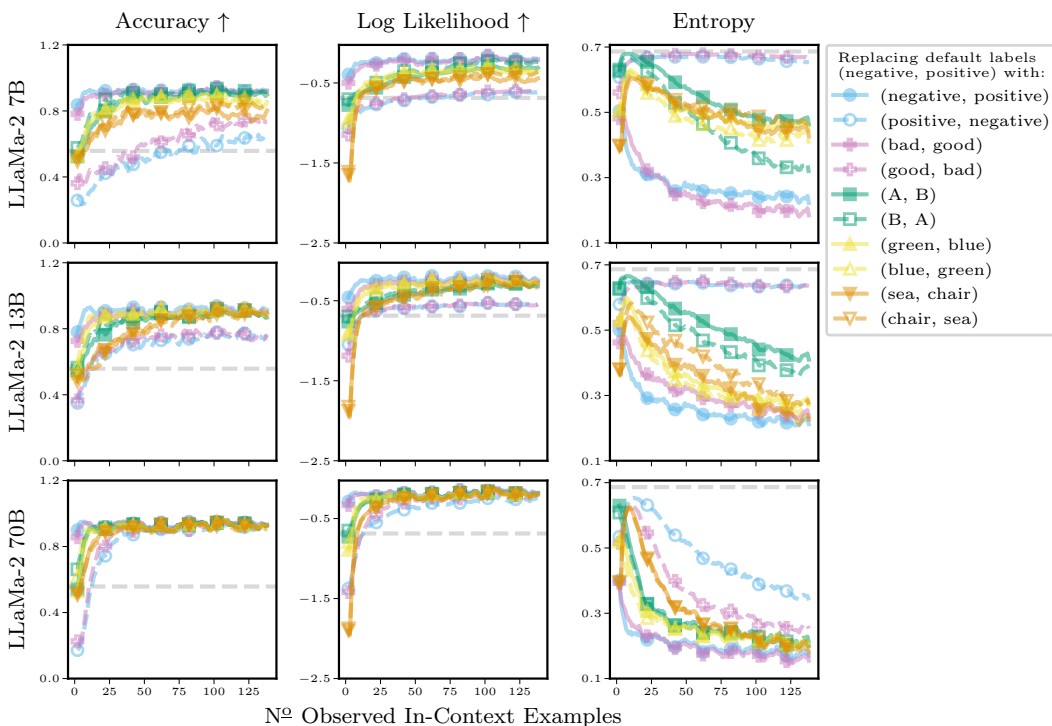

Figure F.14: Few-shot ICL with **replacement labels** on **SST-2** for LLaMa-2 models. In addition to the default labels, we study a variety of replacement labels (see legend). We average over 100 random subsets and then additionally apply moving averages (window size 5) for clarity.

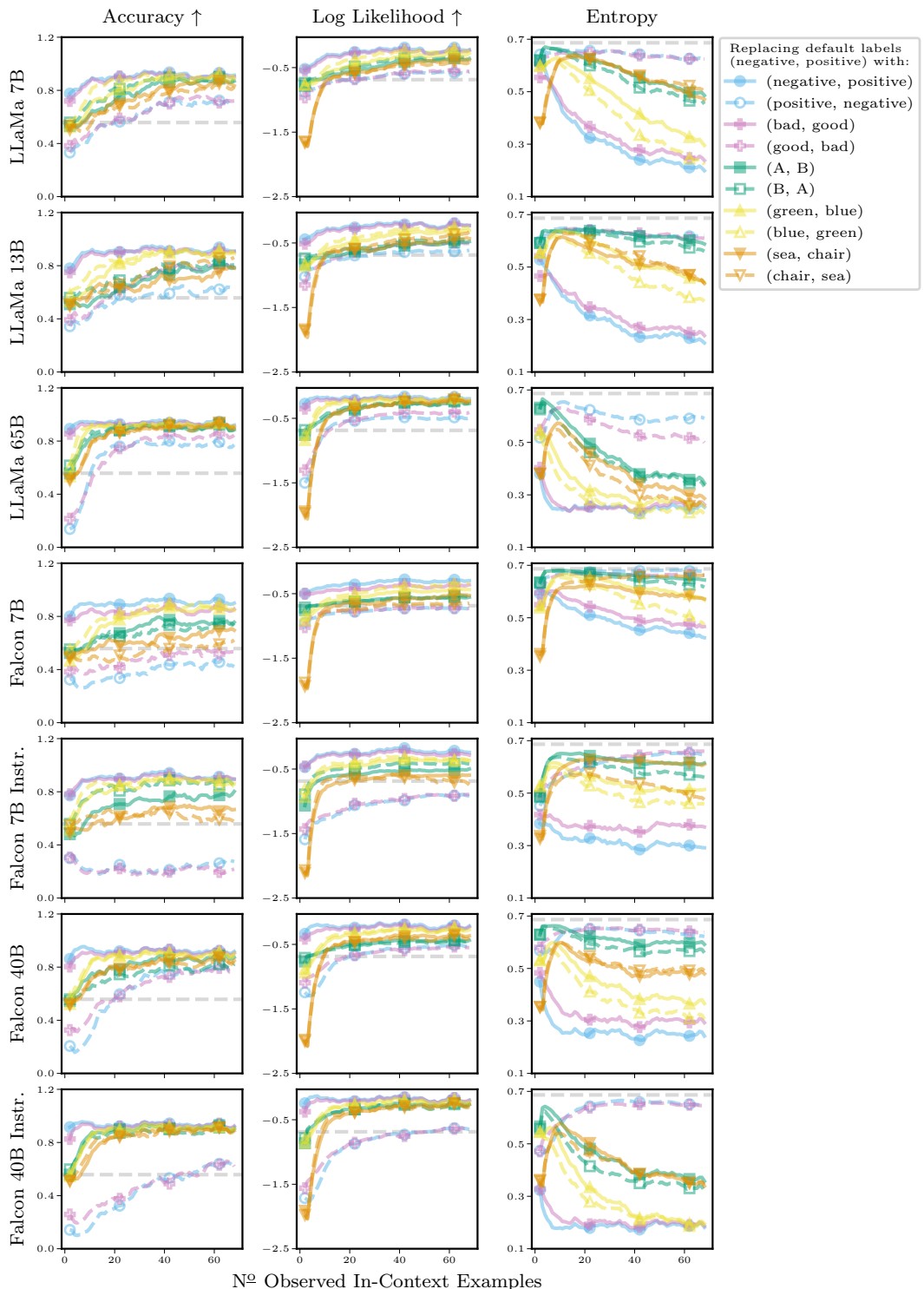

Figure F.15: Few-shot ICL with **replacement labels** on **SST-2** for LLaMa and Falcon models. In addition to the default labels, we study a variety of replacement labels (see legend). We average over 100 random subsets and then additionally apply moving averages (window size 5) for clarity.

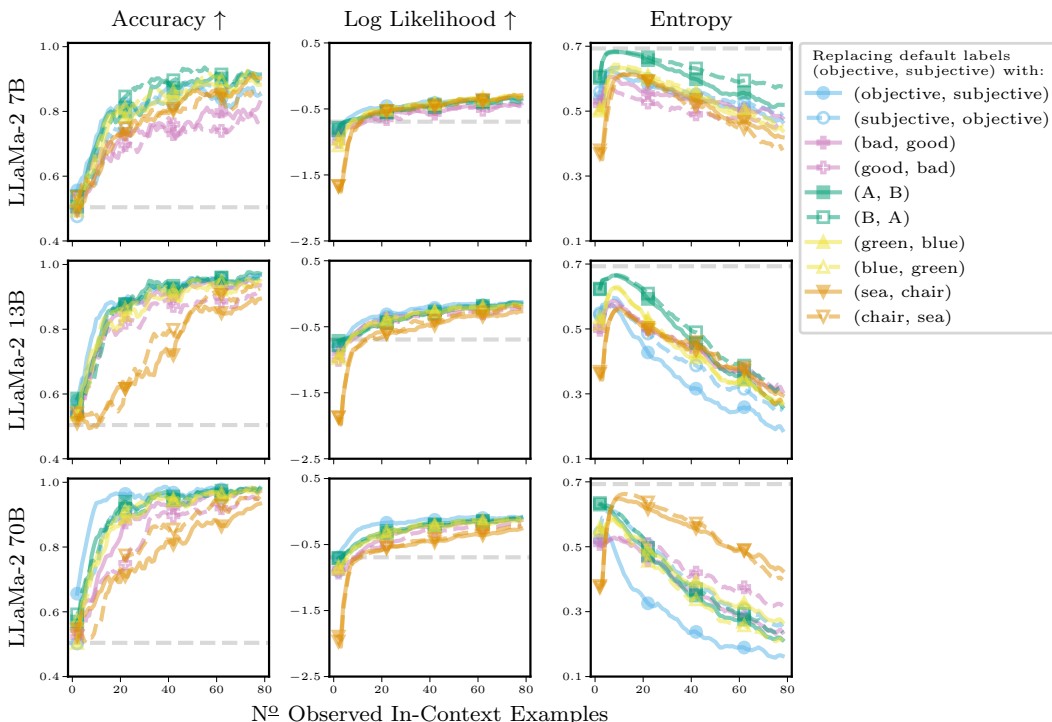

Figure F.16: Few-shot ICL with **replacement labels** on **Subjectivity** for LLaMa-2 models. In addition to the default labels, we study a variety of replacement labels (see legend). We average over 100 random subsets and then additionally apply moving averages (window size 5) for clarity.

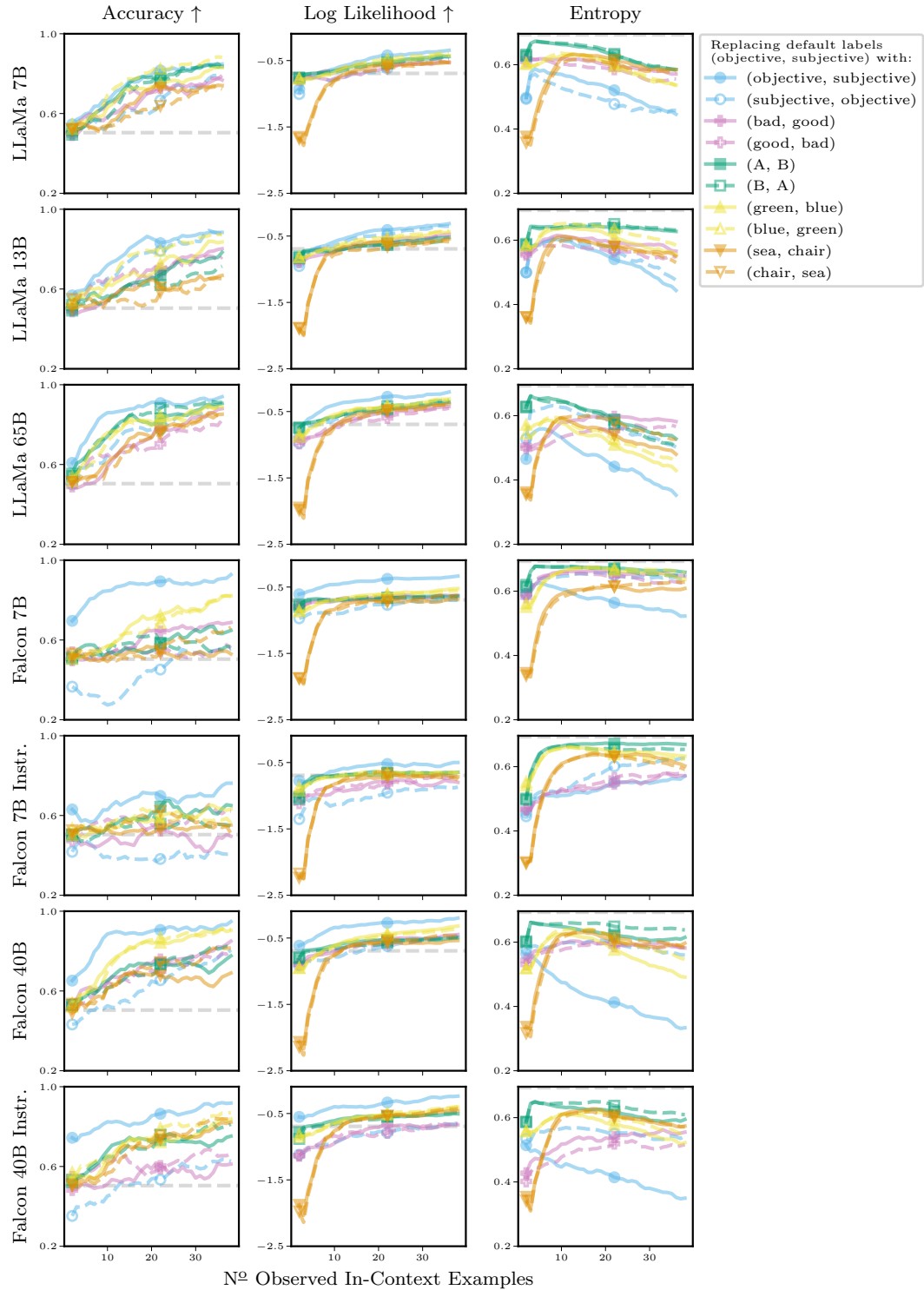

Figure F.17: Few-shot ICL with **replacement labels** on **Subjectivity** for LLaMa and Falcon models. In addition to the default labels, we study a variety of replacement labels (see legend). We average over 100 random subsets and then additionally apply moving averages (window size 5) for clarity.

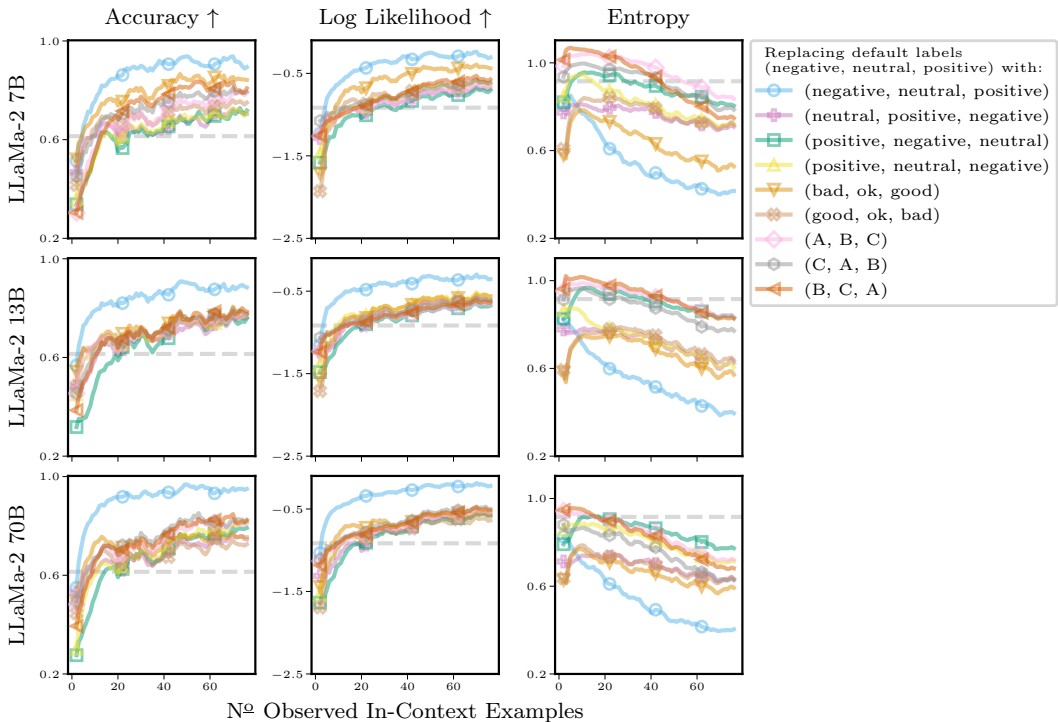

Figure F.18: Few-shot ICL with **replacement labels** on **Financial Phrasebank** for LLaMa-2 models. In addition to the default labels, we study a variety of replacement labels (see legend). We average over 100 random subsets and then additionally apply moving averages (window size 5) for clarity.

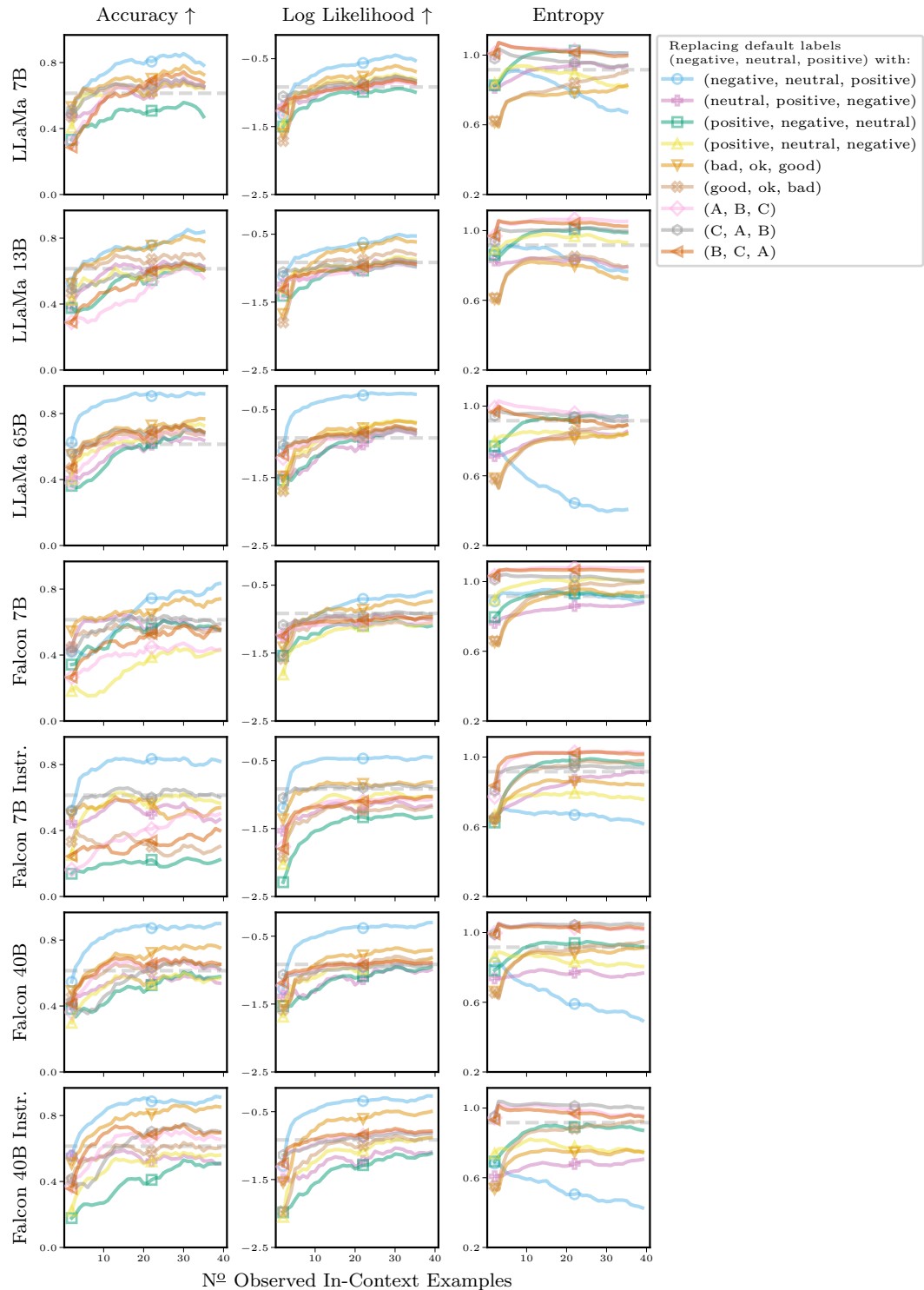

Figure F.19: Few-shot ICL with **replacement labels** on **Financial Phrasebank** for LLaMa and Falcon models. In addition to the default labels, we study a variety of replacement labels (see legend). We average over 100 random subsets and then additionally apply moving averages (window size 5) for clarity.

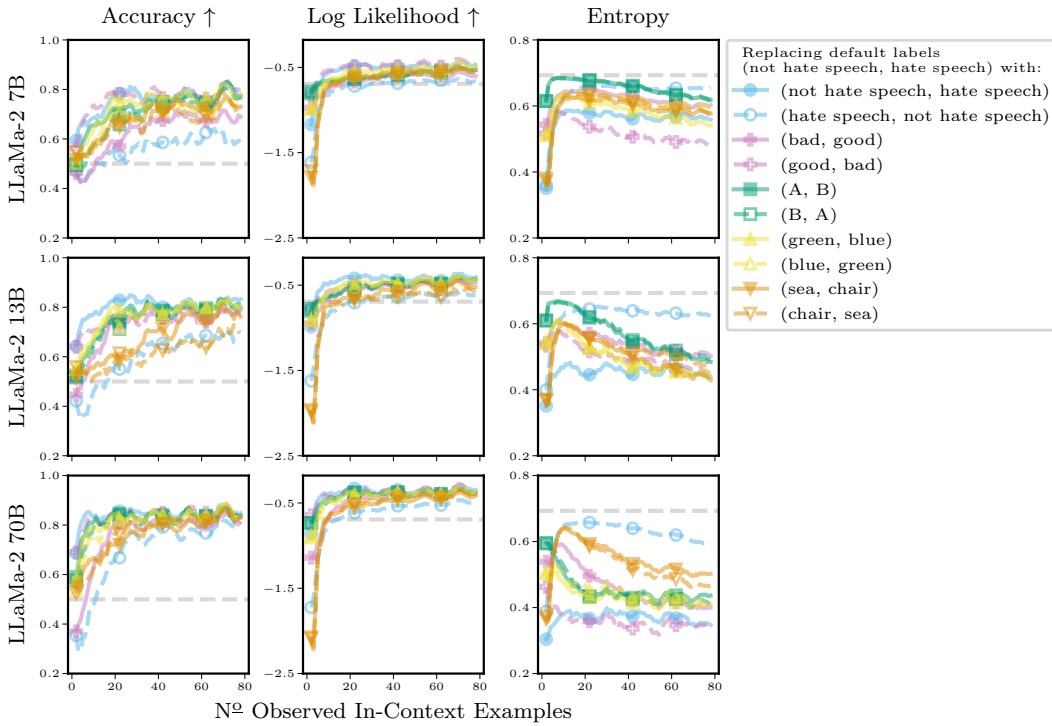

Figure F.20: Few-shot ICL with **replacement labels** on **Hate Speech** for LLaMa-2 models. In addition to the default labels, we study a variety of replacement labels (see legend). We average over 100 random subsets and then additionally apply moving averages (window size 5) for clarity.

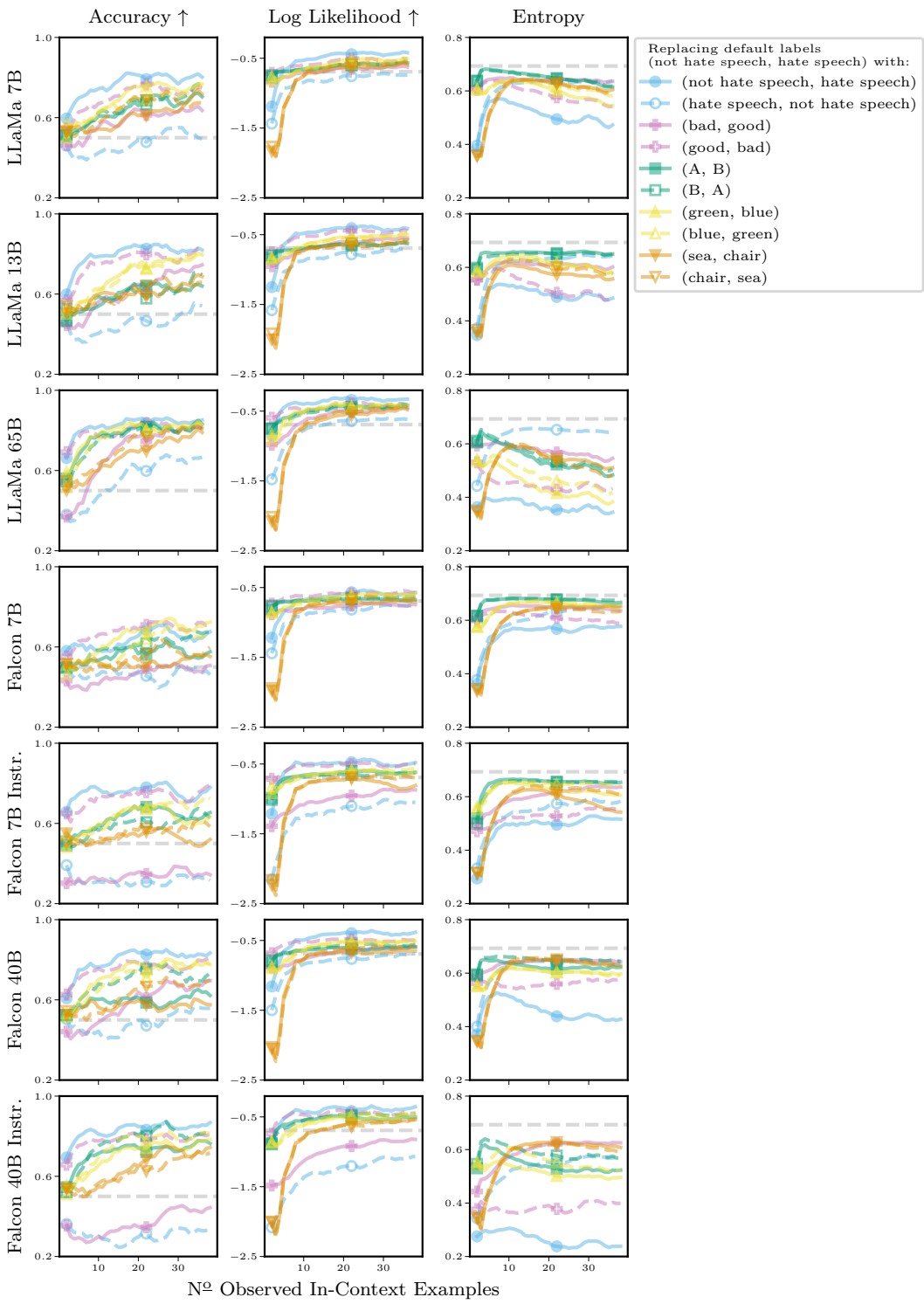

Figure F.21: Few-shot ICL with **replacement labels** on **Hate Speech** for LLaMa and Falcon models. In addition to the default labels, we study a variety of replacement labels (see legend). We average over 100 random subsets and then additionally apply moving averages (window size 5) for clarity.

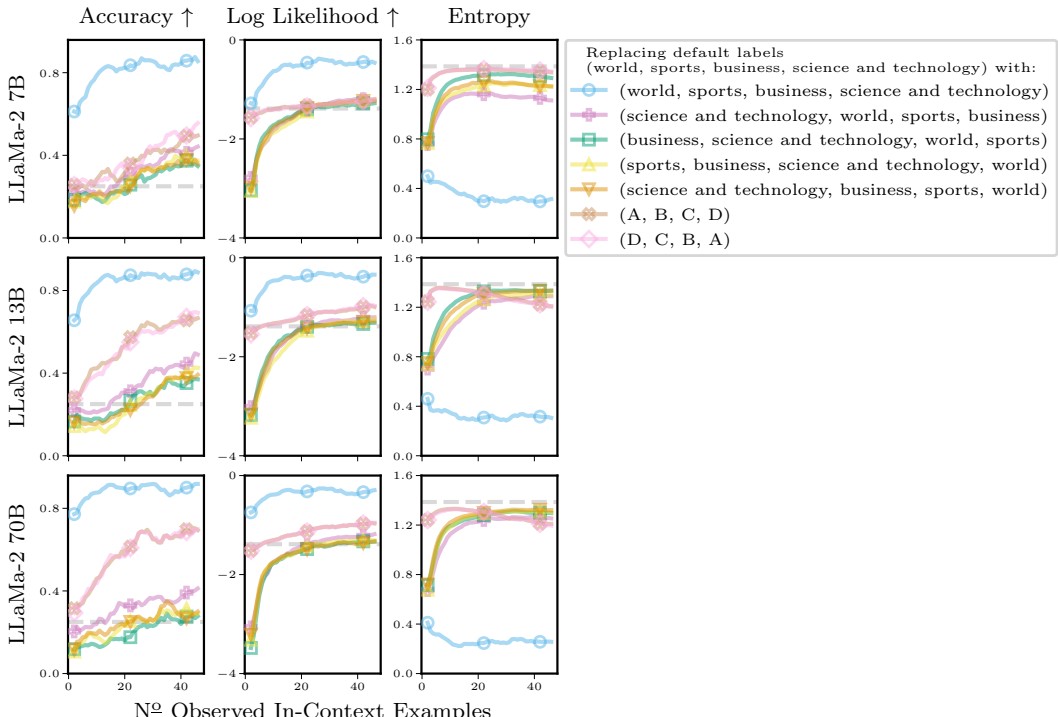

Figure F.22: Few-shot ICL with **replacement labels** on **AG News** for LLaMa-2 models. In addition to the default labels, we study a variety of replacement labels (see legend). We average over 100 random subsets and then additionally apply moving averages (window size 5) for clarity.

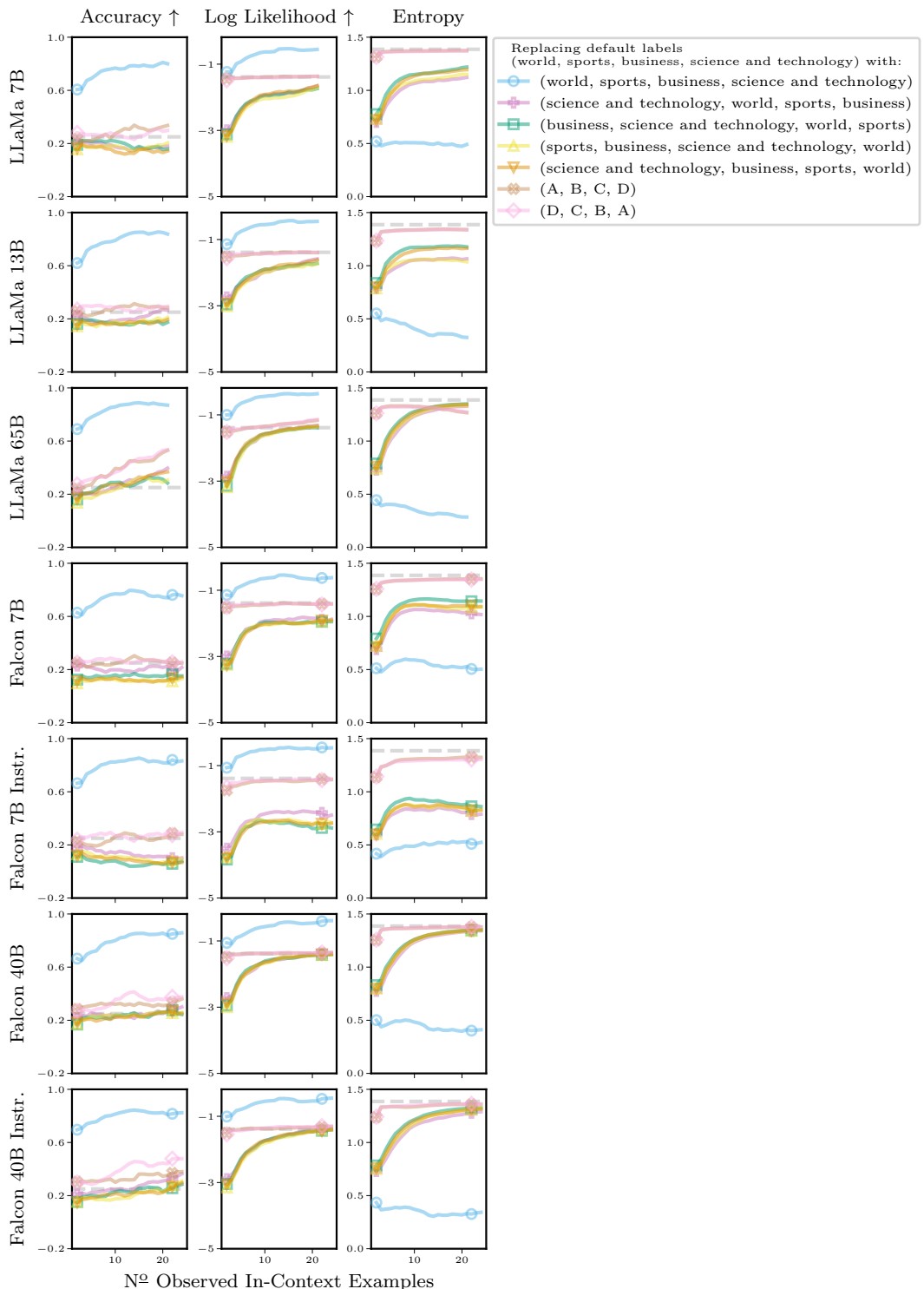

Figure F.23: Few-shot ICL with **replacement labels** on **AG News** for LLaMa and Falcon models. In addition to the default labels, we study a variety of replacement labels (see legend). We average over 100 random subsets and then additionally apply moving averages (window size 5) for clarity.

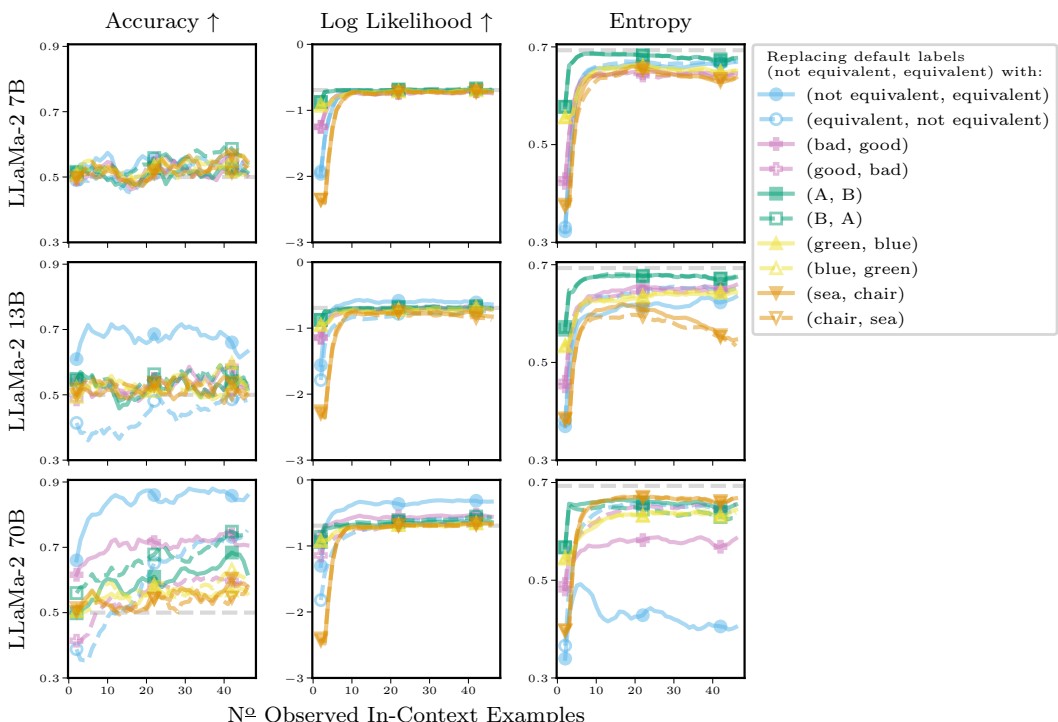

Figure F.24: Few-shot ICL with **replacement labels** on **Medical Questions Pairs** for LLaMa-2 models. In addition to the default labels, we study a variety of replacement labels (see legend). We average over 100 random subsets and then additionally apply moving averages (window size 5) for clarity.

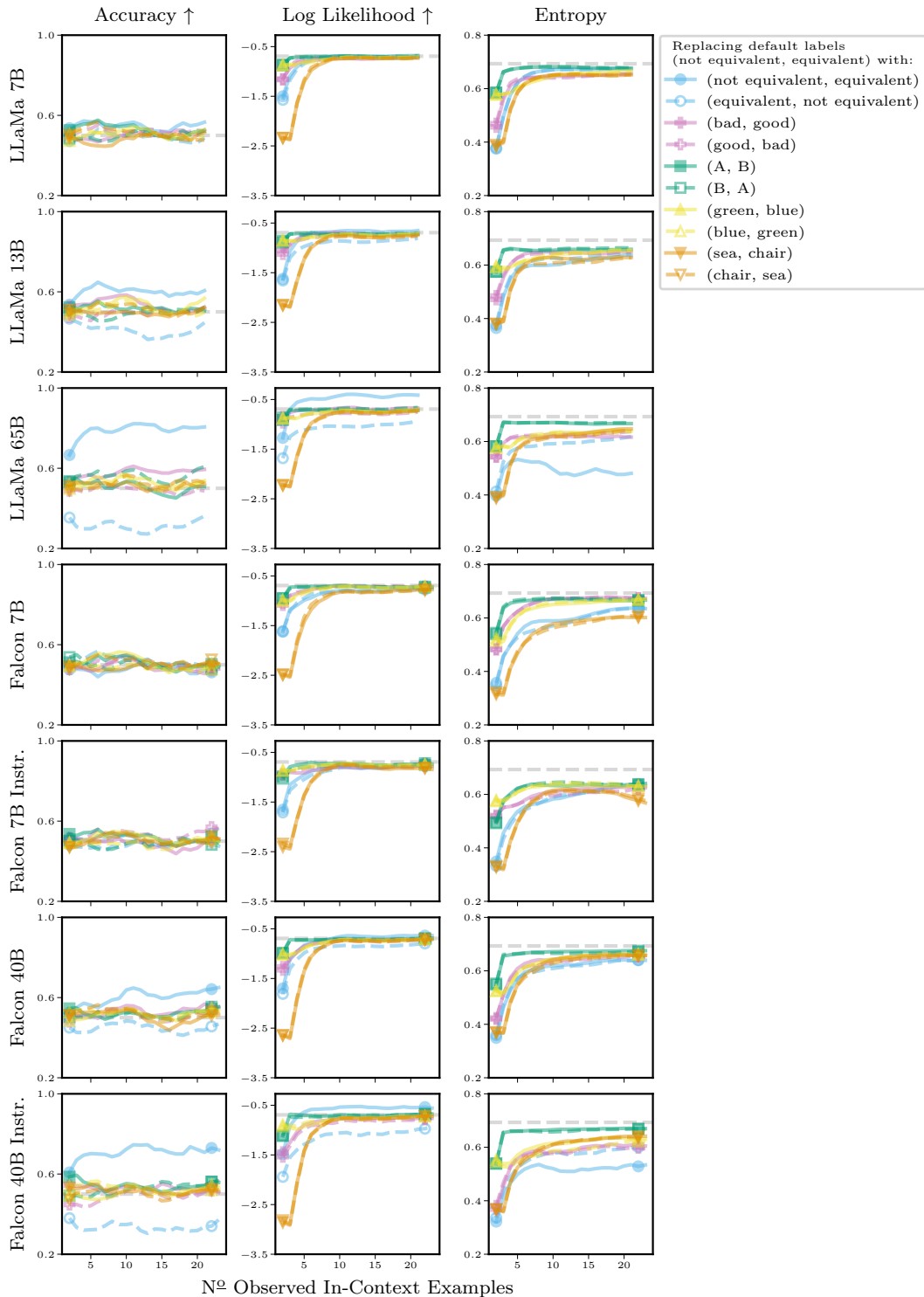

Figure F.25: Few-shot ICL with **replacement labels** on **Medical Questions Pairs** for LLaMa and Falcon models. In addition to the default labels, we study a variety of replacement labels (see legend). We average over 100 random subsets and then additionally apply moving averages (window size 5) for clarity.

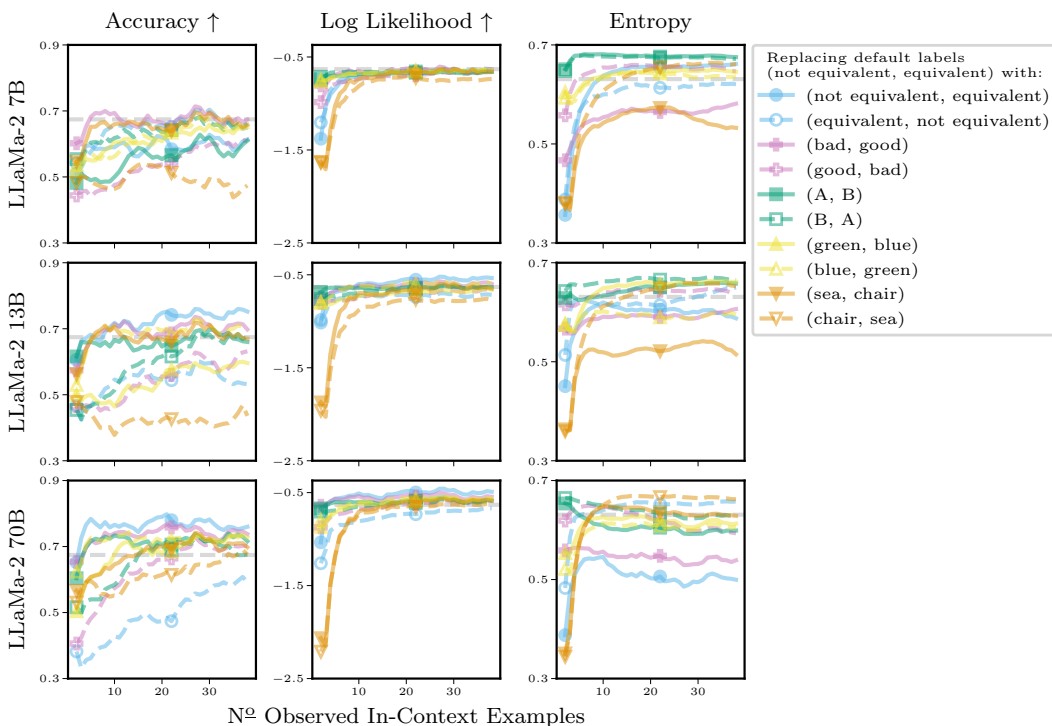

Figure F.26: Few-shot ICL with **replacement labels** on **MRPC** for LLaMa-2 models. In addition to the default labels, we study a variety of replacement labels (see legend). We average over 100 random subsets and then additionally apply moving averages (window size 5) for clarity.

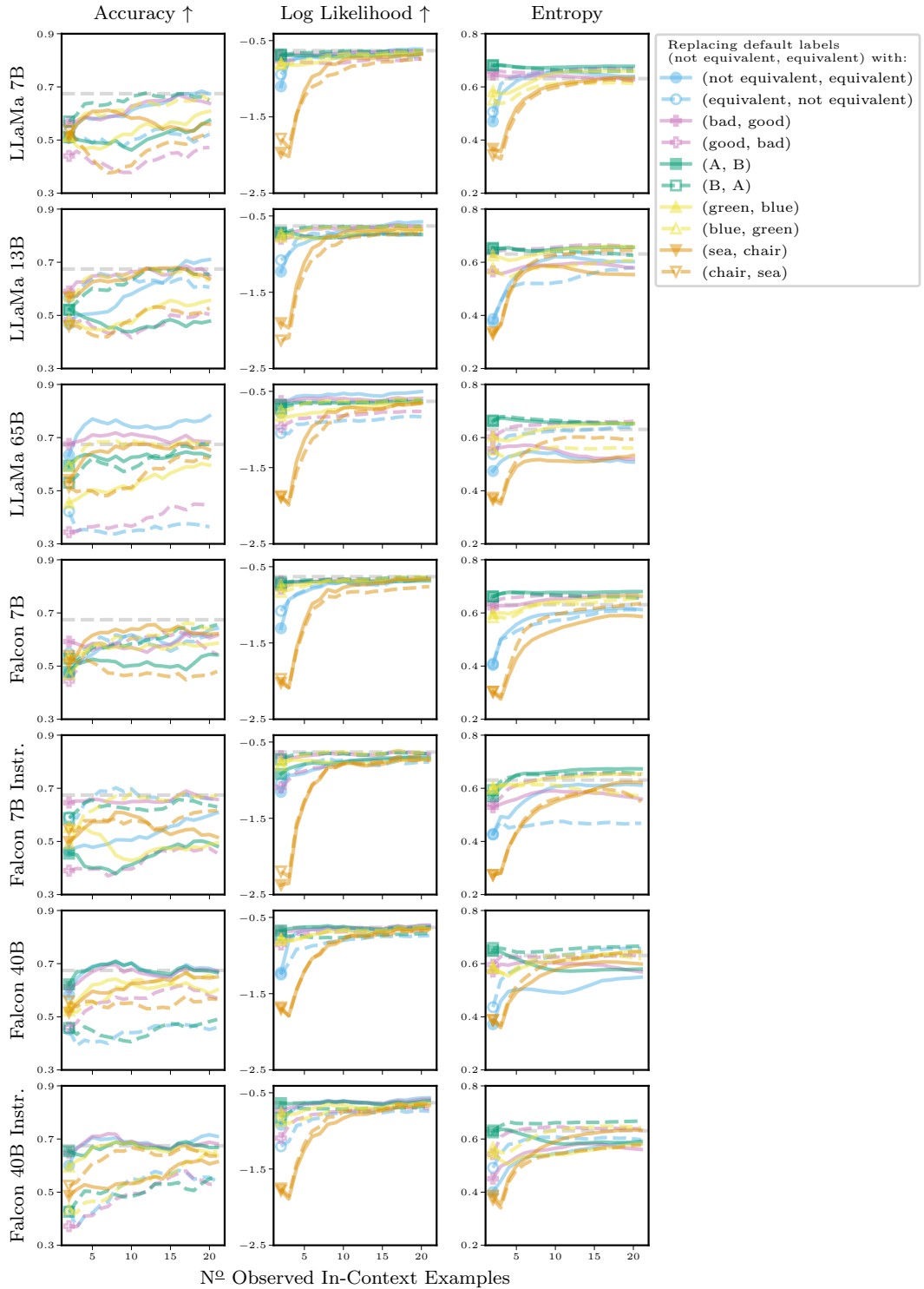

Figure F.27: Few-shot ICL with **replacement labels** on **MRPC** for LLaMa and Falcon models. In addition to the default labels, we study a variety of replacement labels (see legend). We average over 100 random subsets and then additionally apply moving averages (window size 5) for clarity.

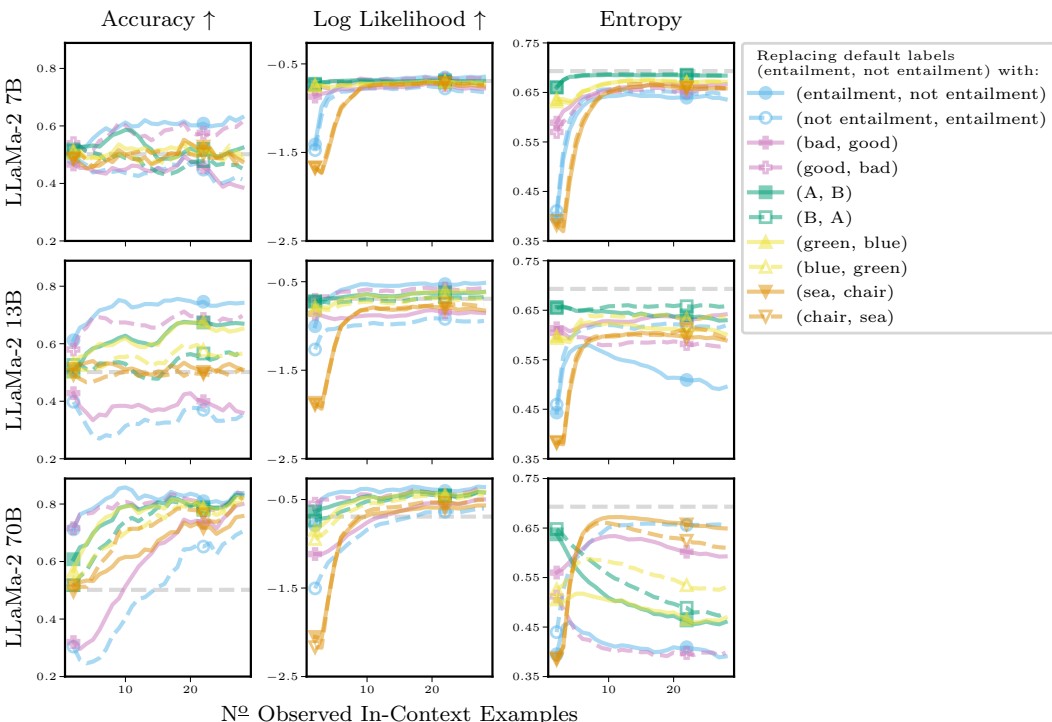

Figure F.28: Few-shot ICL with **replacement labels** on **RTE** for LLaMa-2 models. In addition to the default labels, we study a variety of replacement labels (see legend). We average over 100 random subsets and then additionally apply moving averages (window size 5) for clarity.

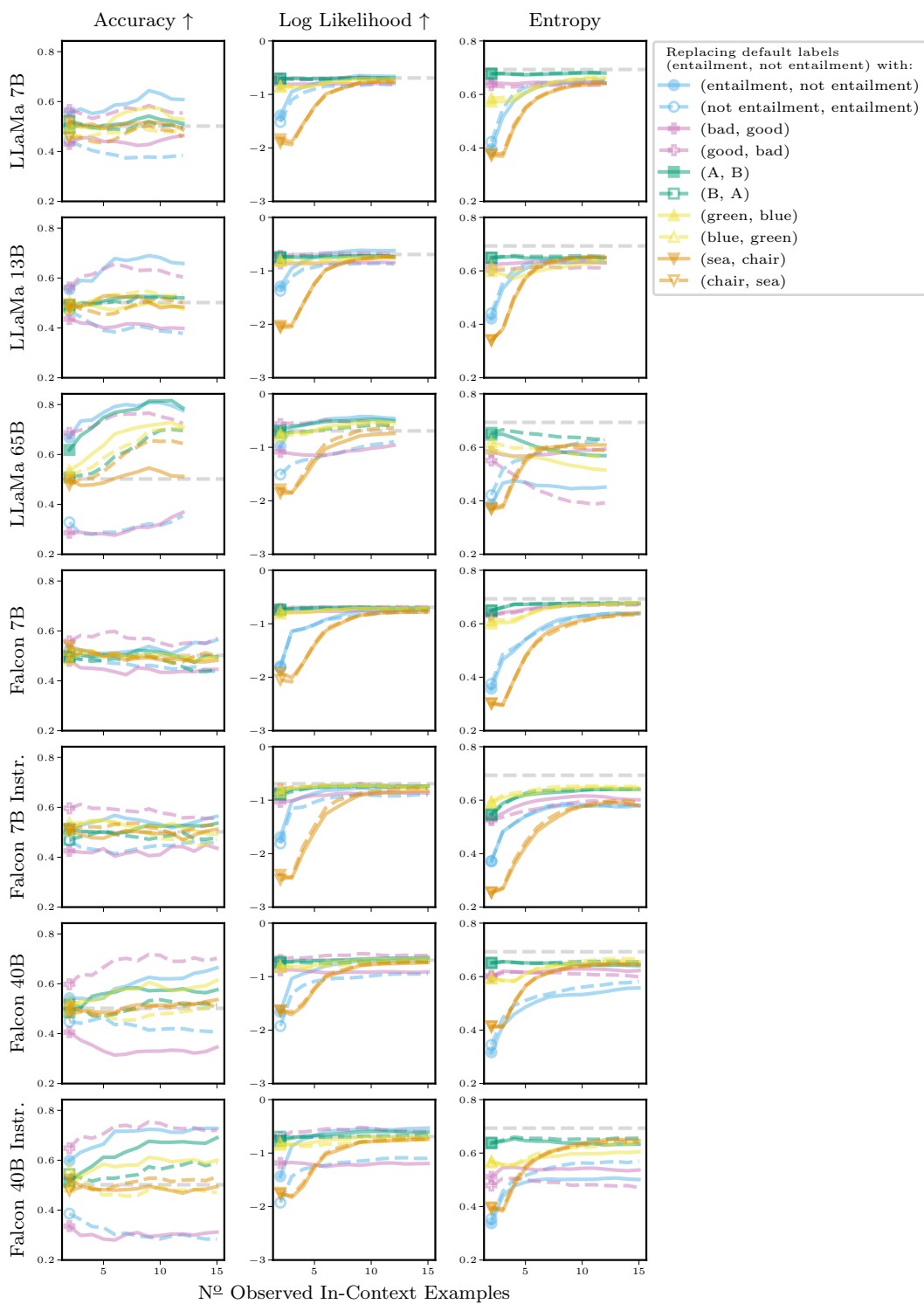

Figure F.29: Few-shot ICL with **replacement labels** on **RTE** for LLaMa and Falcon models. In addition to the default labels, we study a variety of replacement labels (see legend). We average over 100 random subsets and then additionally apply moving averages (window size 5) for clarity.

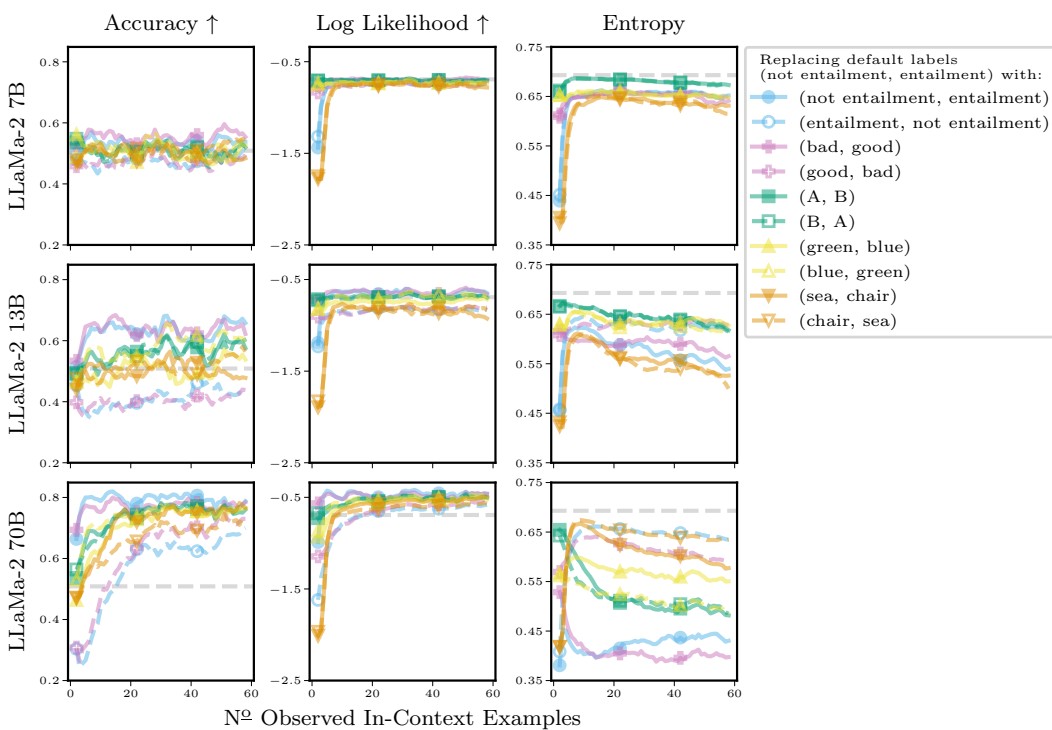

Figure F.30: Few-shot ICL with **replacement labels** on **WNLI** for LLaMa-2 models. In addition to the default labels, we study a variety of replacement labels (see legend). We average over 100 random subsets and then additionally apply moving averages (window size 5) for clarity.

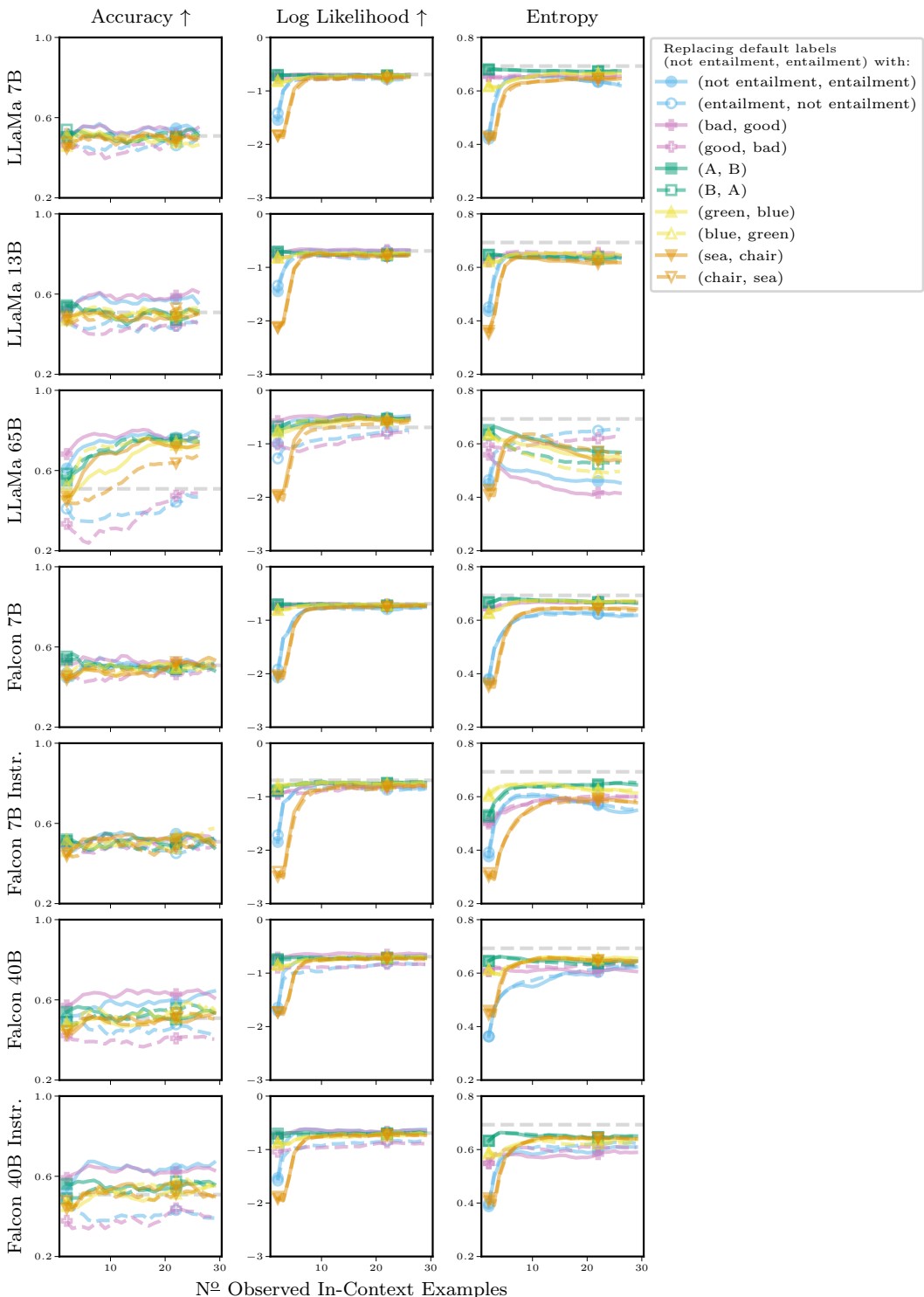

Figure F.31: Few-shot ICL with **replacement labels** on **WNLI** for LLaMa and Falcon models. In addition to the default labels, we study a variety of replacement labels (see legend). We average over 100 random subsets and then additionally apply moving averages (window size 5) for clarity.

Table F.2: Full summary statistics for the flipped label experiments of §7. We strongly encourage readers to also view the full training curves across all possible context sizes and adittional replacement labels in §F. Across all metrics, here show the average difference between the default and flipped label scenario. We compute 'Metric(Default) - Metric(Flipped)', such that positive accuracy/log likelihood differences indicate that ICL performs worse with flipped labels. We compute differences at the maximum context size for each task-model combination (cf.§E and Table E.1). We display experiments where ICL accuracies or log likelihoods on the default labels do not significantly exceed random guessing performance in lightgray. Across models, and metrics, model performance is usually significantly worse when labels are flipped (and default performance exceeds random guessing). In particular for entropies, we observe significant differences between ICL predictions. We display mean differences and standard errors over 100 runs. We bold entries for which mean differences are statistically significant.

| $\Delta$ **Log Lik.** | SST-2 | Subj | F. Phraseb. | Hate Speech | AG News | MQP | MRPC | RTE | WNLI |
|---|---|---|---|---|---|---|---|---|---|
| LLaMa-2 7B | $0.41 \pm 0.02$ | $-0.04 \pm 0.02$ | $0.36 \pm 0.03$ | $0.26 \pm 0.01$ | $0.87 \pm 0.03$ | $0.01 \pm 0.00$ | $-0.02 \pm 0.01$ | $0.22 \pm 0.01$ | $-0.00 \pm 0.01$ |
| LLaMa-2 13B | $0.31 \pm 0.02$ | $0.03 \pm 0.02$ | $0.30 \pm 0.02$ | $0.26 \pm 0.02$ | $0.90 \pm 0.03$ | $0.15 \pm 0.01$ | $0.15 \pm 0.01$ | $0.38 \pm 0.02$ | $0.20 \pm 0.02$ |
| LLaMa-2 70B | $0.08 \pm 0.02$ | $0.08 \pm 0.02$ | $0.35 \pm 0.03$ | $0.18 \pm 0.02$ | $0.90 \pm 0.02$ | $0.24 \pm 0.02$ | $0.15 \pm 0.02$ | $0.21 \pm 0.02$ | $0.08 \pm 0.02$ |
| LLaMa 7B | $0.36 \pm 0.02$ | $0.13 \pm 0.02$ | $0.53 \pm 0.02$ | $0.31 \pm 0.02$ | $1.06 \pm 0.03$ | $0.05 \pm 0.00$ | $0.06 \pm 0.01$ | $0.14 \pm 0.01$ | $-0.03 \pm 0.01$ |
| LLaMa 13B | $0.40 \pm 0.02$ | $0.09 \pm 0.02$ | $0.51 \pm 0.02$ | $0.33 \pm 0.02$ | $1.18 \pm 0.03$ | $0.16 \pm 0.01$ | $0.16 \pm 0.01$ | $0.22 \pm 0.01$ | $0.05 \pm 0.01$ |
| LLaMa 65B | $0.33 \pm 0.02$ | $0.16 \pm 0.02$ | $0.63 \pm 0.02$ | $0.19 \pm 0.02$ | $1.02 \pm 0.03$ | $0.39 \pm 0.02$ | $0.31 \pm 0.02$ | $0.41 \pm 0.02$ | $0.20 \pm 0.02$ |
| Falcon 7B | $0.48 \pm 0.02$ | $0.37 \pm 0.01$ | $0.58 \pm 0.02$ | $0.19 \pm 0.01$ | $1.38 \pm 0.04$ | $0.05 \pm 0.01$ | $0.02 \pm 0.01$ | $0.05 \pm 0.01$ | $0.12 \pm 0.01$ |
| Falcon 7B Instr. | $0.70 \pm 0.02$ | $0.48 \pm 0.01$ | $0.77 \pm 0.02$ | $0.52 \pm 0.02$ | $2.29 \pm 0.03$ | $-0.07 \pm 0.01$ | $0.01 \pm 0.02$ | $0.15 \pm 0.02$ | $0.22 \pm 0.02$ |
| Falcon 40B | $0.33 \pm 0.02$ | $0.25 \pm 0.02$ | $0.74 \pm 0.03$ | $0.27 \pm 0.02$ | $1.00 \pm 0.03$ | $0.20 \pm 0.01$ | $0.19 \pm 0.01$ | $0.39 \pm 0.01$ | $0.09 \pm 0.01$ |
| Falcon 40B Instr. | $0.45 \pm 0.02$ | $0.37 \pm 0.02$ | $0.94 \pm 0.03$ | $0.68 \pm 0.02$ | $0.89 \pm 0.03$ | $0.48 \pm 0.02$ | $0.15 \pm 0.01$ | $0.70 \pm 0.02$ | $0.19 \pm 0.01$ |

| $\Delta$ **Accuracy** | SST-2 | Subj | F. Phraseb. | Hate Speech | AG News | MQP | MRPC | RTE | WNLI |
|---|---|---|---|---|---|---|---|---|---|
| LLaMa-2 7B | $34.0 \pm 5.3$ | $-3.0 \pm 4.3$ | $15.0 \pm 4.3$ | $26.0 \pm 6.9$ | $47.0 \pm 5.6$ | $1.0 \pm 8.4$ | $-5.0 \pm 5.9$ | $32.0 \pm 7.5$ | $1.0 \pm 7.3$ |
| LLaMa-2 13B | $14.0 \pm 4.0$ | $1.0 \pm 4.0$ | $12.0 \pm 3.8$ | $18.0 \pm 5.4$ | $37.0 \pm 5.8$ | $22.1 \pm 5.3$ | $20.0 \pm 5.3$ | $36.0 \pm 7.0$ | $27.0 \pm 6.5$ |
| LLaMa-2 70B | $2.0 \pm 1.4$ | $1.0 \pm 2.2$ | $13.0 \pm 3.4$ | $0.0 \pm 3.2$ | $51.0 \pm 5.6$ | $11.0 \pm 4.7$ | $20.0 \pm 5.5$ | $7.0 \pm 5.0$ | $5.0 \pm 3.8$ |
| LLaMa 7B | $22.0 \pm 4.8$ | $3.0 \pm 4.8$ | $36.0 \pm 5.9$ | $27.0 \pm 6.6$ | $64.0 \pm 6.4$ | $6.0 \pm 6.6$ | $13.0 \pm 6.6$ | $19.0 \pm 7.0$ | $0.0 \pm 8.6$ |
| LLaMa 13B | $26.0 \pm 5.8$ | $7.0 \pm 3.5$ | $26.0 \pm 5.8$ | $36.0 \pm 5.9$ | $55.0 \pm 6.2$ | $16.0 \pm 6.7$ | $12.0 \pm 4.5$ | $28.0 \pm 7.5$ | $6.0 \pm 6.9$ |
| LLaMa 65B | $15.0 \pm 4.3$ | $8.0 \pm 3.7$ | $27.0 \pm 5.3$ | $14.0 \pm 5.8$ | $50.0 \pm 5.7$ | $29.0 \pm 7.4$ | $41.0 \pm 6.3$ | $46.0 \pm 6.7$ | $30.0 \pm 6.1$ |
| Falcon 7B | $49.0 \pm 5.9$ | $39.0 \pm 4.9$ | $34.0 \pm 6.7$ | $20.0 \pm 8.4$ | $63.0 \pm 6.7$ | $13.0 \pm 8.2$ | $7.0 \pm 4.5$ | $10.0 \pm 6.1$ | $12.0 \pm 8.3$ |
| Falcon 7B Instr. | $63.0 \pm 5.8$ | $46.0 \pm 7.0$ | $48.0 \pm 6.6$ | $42.0 \pm 7.4$ | $71.0 \pm 5.2$ | $-10.0 \pm 7.1$ | $-9.0 \pm 6.3$ | $11.0 \pm 7.6$ | $17.0 \pm 8.5$ |
| Falcon 40B | $15.0 \pm 4.3$ | $17.0 \pm 4.5$ | $36.0 \pm 5.9$ | $20.0 \pm 6.8$ | $58.0 \pm 6.5$ | $28.0 \pm 6.8$ | $32.0 \pm 6.8$ | $25.0 \pm 8.3$ | $16.0 \pm 7.8$ |
| Falcon 40B Instr. | $29.0 \pm 5.3$ | $30.0 \pm 5.2$ | $45.0 \pm 5.5$ | $53.0 \pm 6.8$ | $49.0 \pm 6.9$ | $45.0 \pm 7.0$ | $11.0 \pm 6.3$ | $48.0 \pm 7.0$ | $22.0 \pm 7.6$ |

| $\Delta$ **Entropy** | SST-2 | Subj | F. Phraseb. | Hate Speech | AG News | MQP | MRPC | RTE | WNLI |
|---|---|---|---|---|---|---|---|---|---|
| LLaMa-2 7B | $-0.519 \pm 0.022$ | $0.010 \pm 0.017$ | $-0.397 \pm 0.026$ | $-0.100 \pm 0.013$ | $-0.749 \pm 0.030$ | $-0.003 \pm 0.005$ | $0.049 \pm 0.005$ | $-0.027 \pm 0.008$ | $-0.006 \pm 0.008$ |
| LLaMa-2 13B | $-0.479 \pm 0.019$ | $-0.058 \pm 0.015$ | $-0.475 \pm 0.024$ | $-0.194 \pm 0.019$ | $-0.953 \pm 0.028$ | $-0.027 \pm 0.010$ | $-0.067 \pm 0.011$ | $-0.106 \pm 0.018$ | $-0.073 \pm 0.017$ |
| LLaMa-2 70B | $-0.167 \pm 0.019$ | $-0.100 \pm 0.015$ | $-0.398 \pm 0.029$ | $-0.255 \pm 0.018$ | $-0.995 \pm 0.024$ | $-0.243 \pm 0.019$ | $-0.150 \pm 0.015$ | $-0.258 \pm 0.019$ | $-0.209 \pm 0.017$ |
| LLaMa 7B | $-0.434 \pm 0.020$ | $-0.017 \pm 0.023$ | $-0.355 \pm 0.023$ | $-0.159 \pm 0.019$ | $-0.645 \pm 0.030$ | $0.007 \pm 0.003$ | $-0.032 \pm 0.005$ | $-0.005 \pm 0.008$ | $-0.014 \pm 0.010$ |
| LLaMa 13B | $-0.424 \pm 0.023$ | $-0.005 \pm 0.020$ | $-0.237 \pm 0.022$ | $-0.151 \pm 0.018$ | $-0.731 \pm 0.033$ | $0.008 \pm 0.009$ | $0.019 \pm 0.012$ | $-0.010 \pm 0.008$ | $-0.009 \pm 0.009$ |
| LLaMa 65B | $-0.345 \pm 0.019$ | $-0.140 \pm 0.020$ | $-0.528 \pm 0.024$ | $-0.273 \pm 0.020$ | $-1.041 \pm 0.029$ | $-0.128 \pm 0.019$ | $-0.141 \pm 0.015$ | $-0.205 \pm 0.020$ | $-0.199 \pm 0.019$ |
| Falcon 7B | $-0.276 \pm 0.019$ | $-0.118 \pm 0.014$ | $-0.022 \pm 0.018$ | $-0.062 \pm 0.013$ | $-0.518 \pm 0.037$ | $0.002 \pm 0.009$ | $0.005 \pm 0.006$ | $-0.004 \pm 0.010$ | $-0.015 \pm 0.013$ |
| Falcon 7B Instr. | $-0.367 \pm 0.020$ | $-0.068 \pm 0.012$ | $-0.327 \pm 0.021$ | $-0.052 \pm 0.016$ | $-0.220 \pm 0.028$ | $-0.024 \pm 0.009$ | $0.127 \pm 0.015$ | $0.005 \pm 0.015$ | $0.001 \pm 0.020$ |
| Falcon 40B | $-0.392 \pm 0.020$ | $-0.233 \pm 0.022$ | $-0.420 \pm 0.027$ | $-0.190 \pm 0.022$ | $-0.904 \pm 0.033$ | $-0.003 \pm 0.010$ | $-0.103 \pm 0.012$ | $-0.018 \pm 0.014$ | $-0.001 \pm 0.013$ |
| Falcon 40B Instr. | $-0.483 \pm 0.020$ | $-0.160 \pm 0.024$ | $-0.430 \pm 0.028$ | $-0.313 \pm 0.024$ | $-0.916 \pm 0.033$ | $-0.098 \pm 0.017$ | $-0.017 \pm 0.014$ | $-0.063 \pm 0.017$ | $-0.013 \pm 0.013$ |

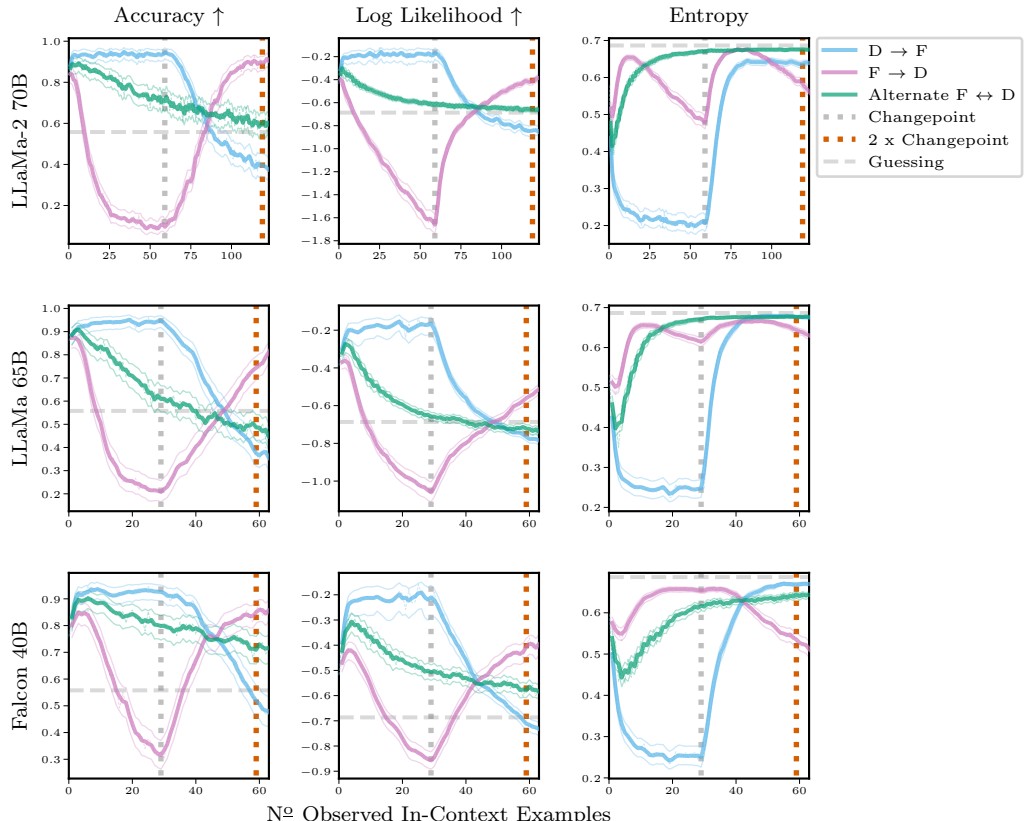

Figure F.32: Few-shot ICL on **SST-2** when the **label relationship changes throughout ICL**. For (D → F), we start with default labels and change to flipped labels at the changepoint, for (F → D) we change from flipped to the default labels at the changepoint, and for (Alternating F ↔ D) we alternate between the two label relationships after every observation. For all setups, at '2 x Changepoint', the LLMs have observed the same number of examples for both label relations. If, according to NH3, ICL treats all in-context information equally, predictions should be equal at that point—but they are not. Averages over 500 repetitions, we apply moving averages (window size 3), and thin lines are bootstrapped 99 % confidence intervals, thin dashed lines are mean results without moving average smoothing.

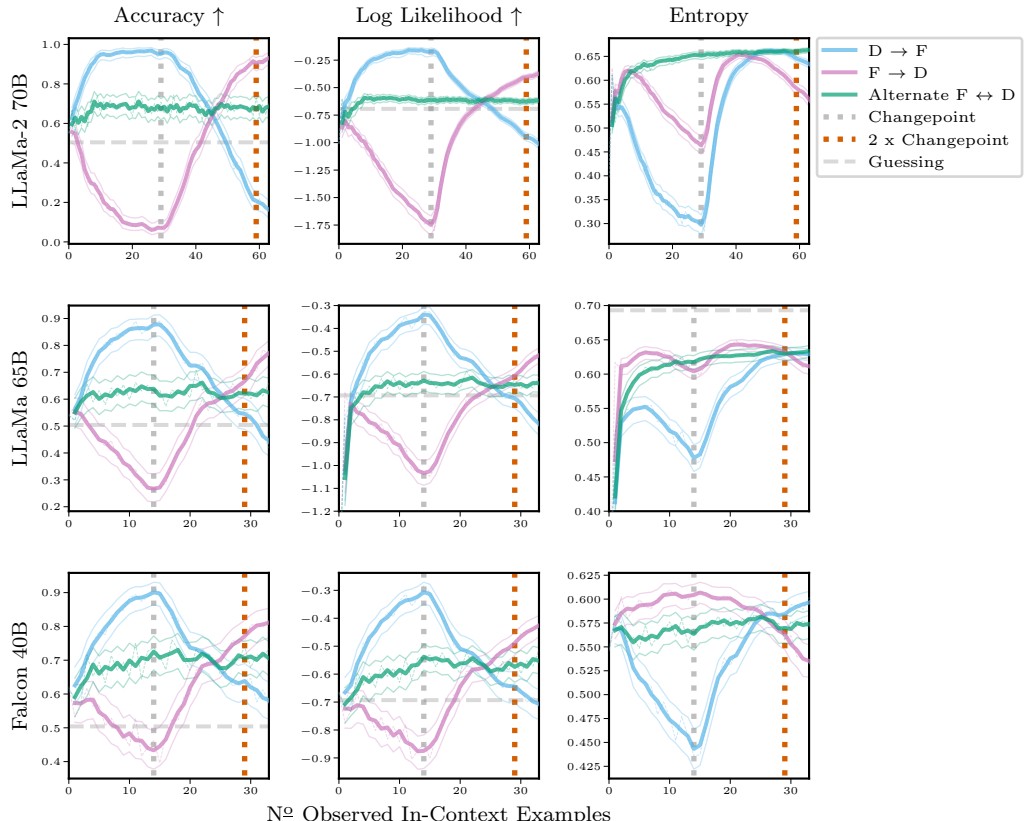

Figure F.33: Few-shot ICL on **Subjectivity** when the **label relationship changes throughout ICL**. For (D → F), we start with default labels and change to flipped labels at the changepoint, for (F → D) we change from flipped to the default labels at the changepoint, and for (Alternating F ↔ D) we alternate between the two label relationships after every observation. For all setups, at '2 x Changepoint', the LLMs have observed the same number of examples for both label relations. If, according to NH3, ICL treats all in-context information equally, predictions should be equal at that point—but they are not. Averages over 500 repetitions, we apply moving averages (window size 3), and thin lines are bootstrapped 99 % confidence intervals, thin dashed lines are mean results without moving average smoothing.

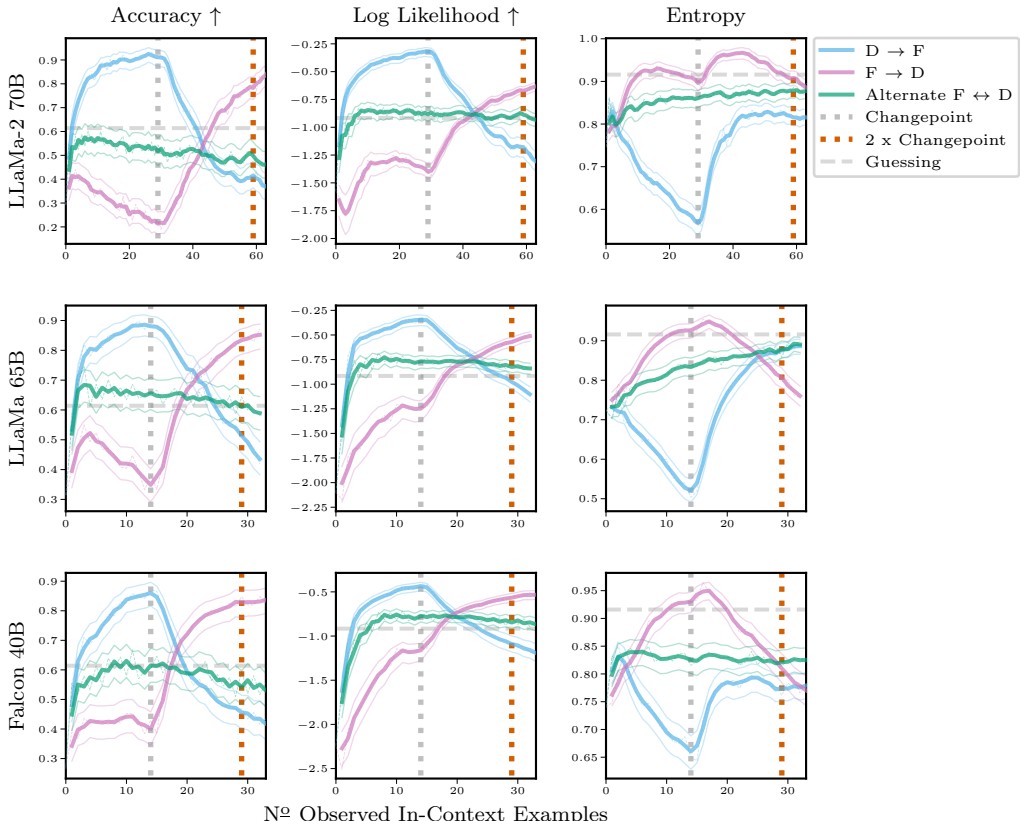

Figure F.34: Few-shot ICL on **Financial Phrasebank** when the **label relationship changes throughout ICL**. For (D → F), we start with default labels and change to flipped labels at the changepoint, for (F → D) we change from flipped to the default labels at the changepoint, and for (Alternating F ↔ D) we alternate between the two label relationships after every observation. For all setups, at '2 x Changepoint', the LLMs have observed the same number of examples for both label relations. If, according to NH3, ICL treats all in-context information equally, predictions should be equal at that point—but they are not. Averages over 500 repetitions, we apply moving averages (window size 3), and thin lines are bootstrapped 99 % confidence intervals, thin dashed lines are mean results without moving average smoothing.

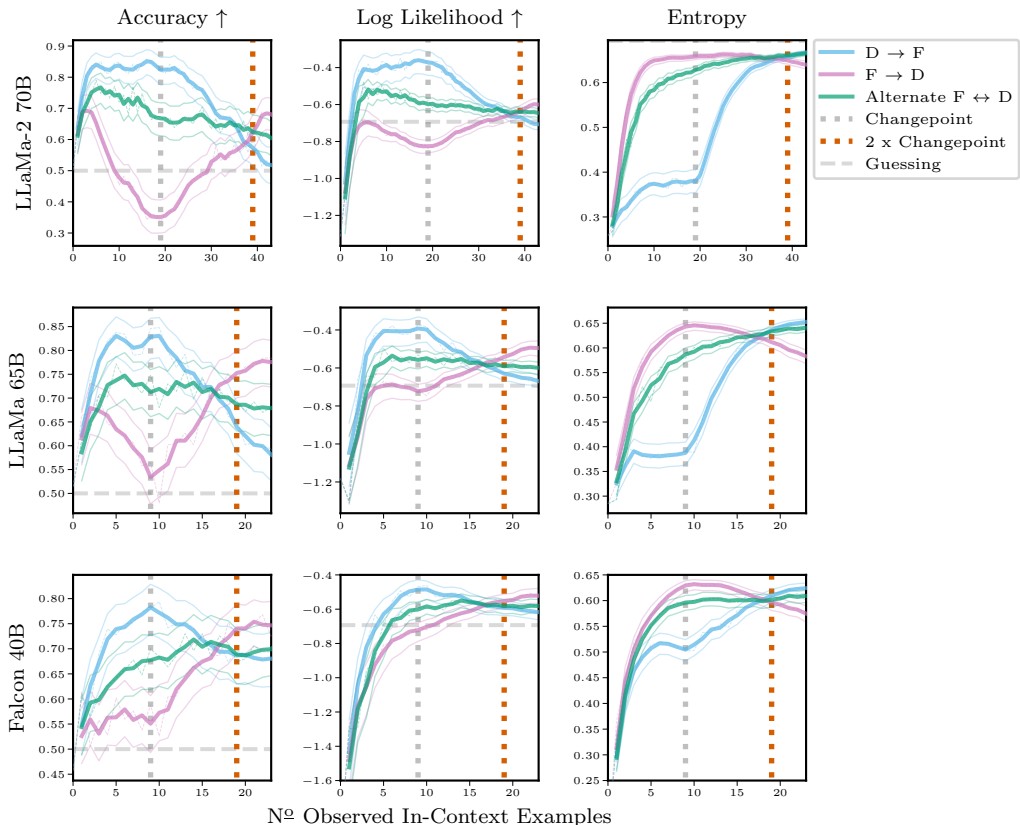

Figure F.35: Few-shot ICL on **Hate Speech** when the **label relationship changes throughout ICL**. For (D → F), we start with default labels and change to flipped labels at the changepoint, for (F → D) we change from flipped to the default labels at the changepoint, and for (Alternating F ↔ D) we alternate between the two label relationships after every observation. For all setups, at '2 x Changepoint', the LLMs have observed the same number of examples for both label relations. If, according to NH3, ICL treats all in-context information equally, predictions should be equal at that point—but they are not. Averages over 500 repetitions, we apply moving averages (window size 3), and thin lines are bootstrapped 99 % confidence intervals, thin dashed lines are mean results without moving average smoothing.

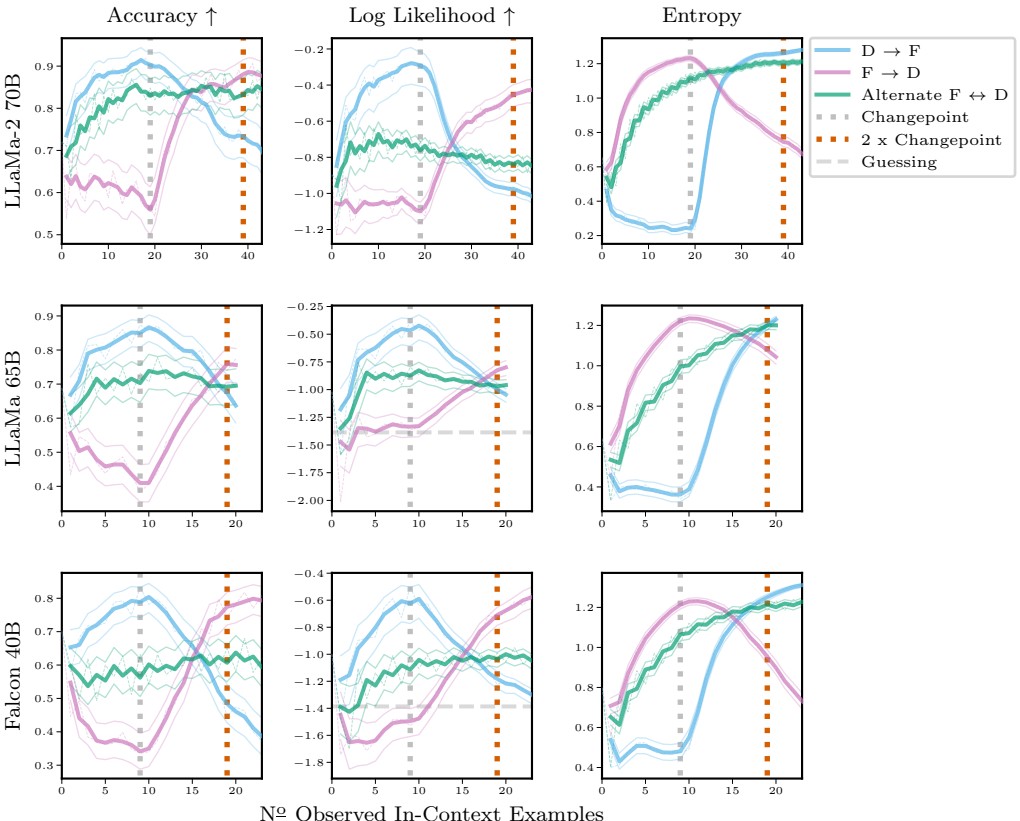

Figure F.36: Few-shot ICL on **AG News** when the **label relationship changes throughout ICL**. For (D → F), we start with default labels and change to flipped labels at the changepoint, for (F → D) we change from flipped to the default labels at the changepoint, and for (Alternating F ↔ D) we alternate between the two label relationships after every observation. For all setups, at '2 x Changepoint', the LLMs have observed the same number of examples for both label relations. If, according to NH3, ICL treats all in-context information equally, predictions should be equal at that point—but they are not. Averages over 500 repetitions, we apply moving averages (window size 3), and thin lines are bootstrapped 99 % confidence intervals, thin dashed lines are mean results without moving average smoothing.

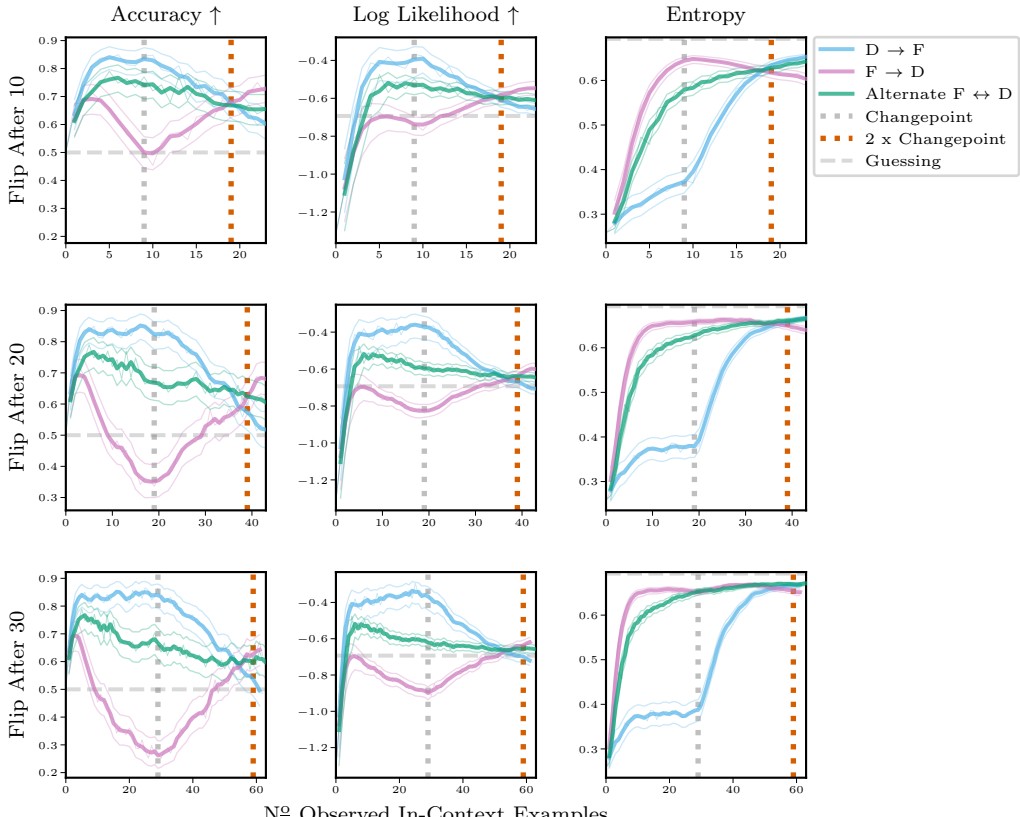

Figure F.37: Few-shot ICL on **Hate Speech** with **LLaMa-2-70B** when the **label relationship changes throughout ICL**. We here investigate a **three different changepoints**, flipping labels after 10, 20, or 30 in-context observations. For (D → F), we start with default labels and change to flipped labels at the changepoint, for (F → D) we change from flipped to the default labels at the changepoint, and for (Alternating F ↔ D) we alternate between the two label relationships after every observation. For all setups, at '2 x Changepoint', the LLMs have observed the same number of examples for both label relations. If, according to NH3, ICL treats all in-context information equally, predictions should be equal at that point, regardless of whether the changepoint is after 10, 20, or 30 observations. However, this is not the case. In particular, predictions are significantly different when the changepoint is 30. Averages over 500 repetitions, we apply moving averages (window size 3), thin lines are bootstrapped 99 % confidence intervals, thin dashed lines are mean results without moving average smoothing.

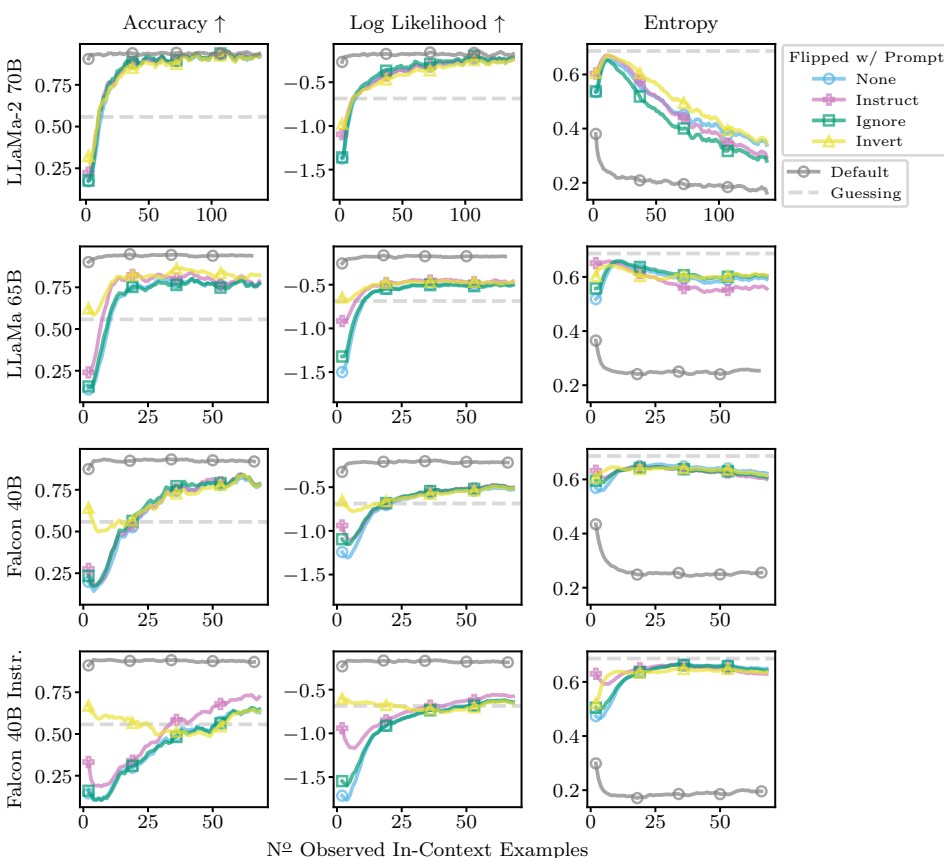

Figure F.38: Prompted few-shot ICL with flipped labels on **SST-2**. Some prompts are able to improve ICL on flipped labels compared to not using a prompt as before (label none in the figure). We average over 100 random subsets and then additionally apply moving averages (window size 5) for clarity.

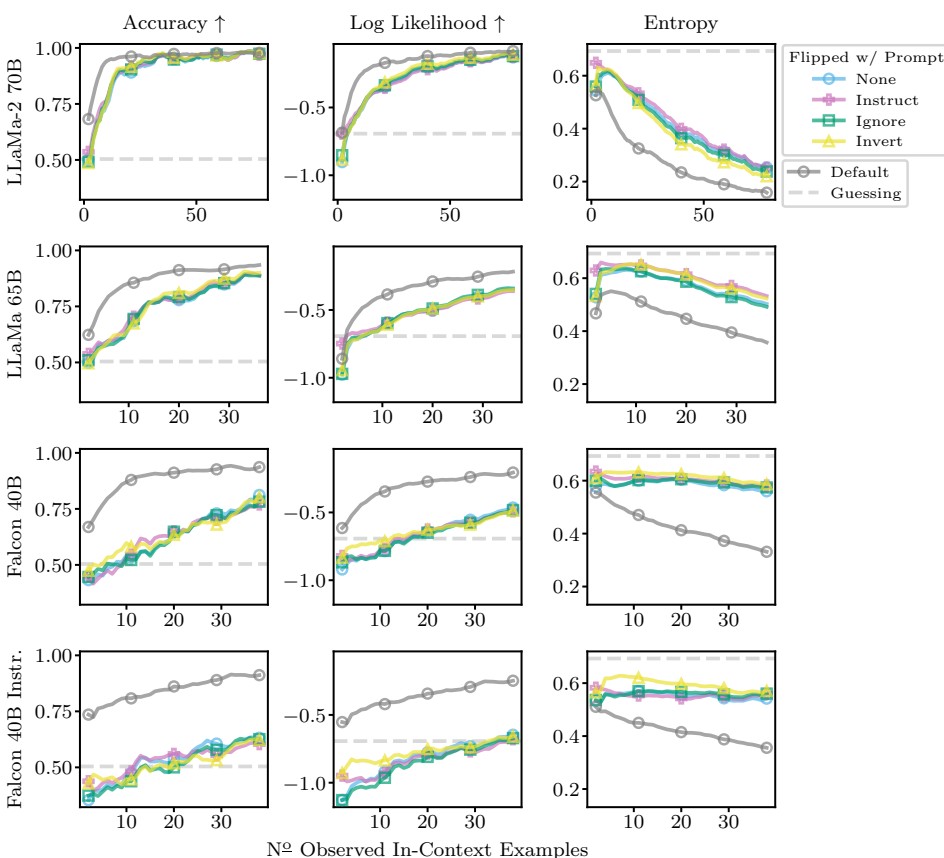

Figure F.39: Prompted few-shot ICL with flipped labels on **Subjectivity**. Some prompts are able to improve ICL on flipped labels compared to not using a prompt as before (label none in the figure). We average over 100 random subsets and then additionally apply moving averages (window size 5) for clarity.

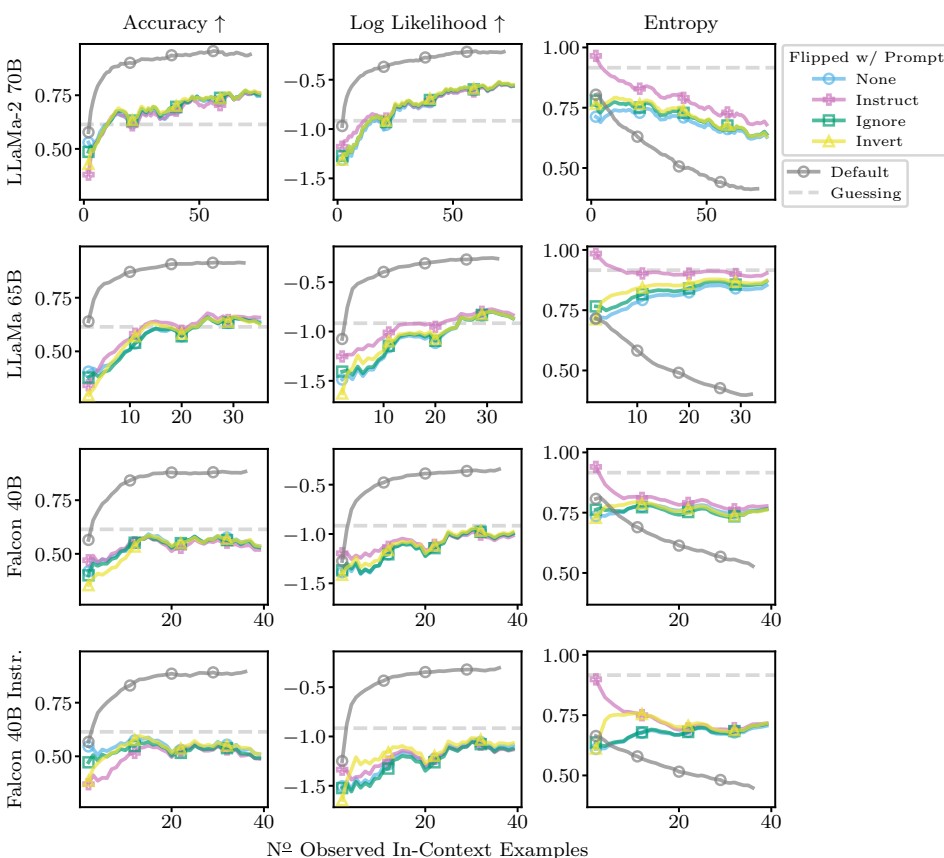

Figure F.40: Prompted few-shot ICL with flipped labels on **Financial Phrasebank**. Some prompts are able to improve ICL on flipped labels compared to not using a prompt as before (label none in the figure). We average over 100 random subsets and then additionally apply moving averages (window size 5) for clarity.

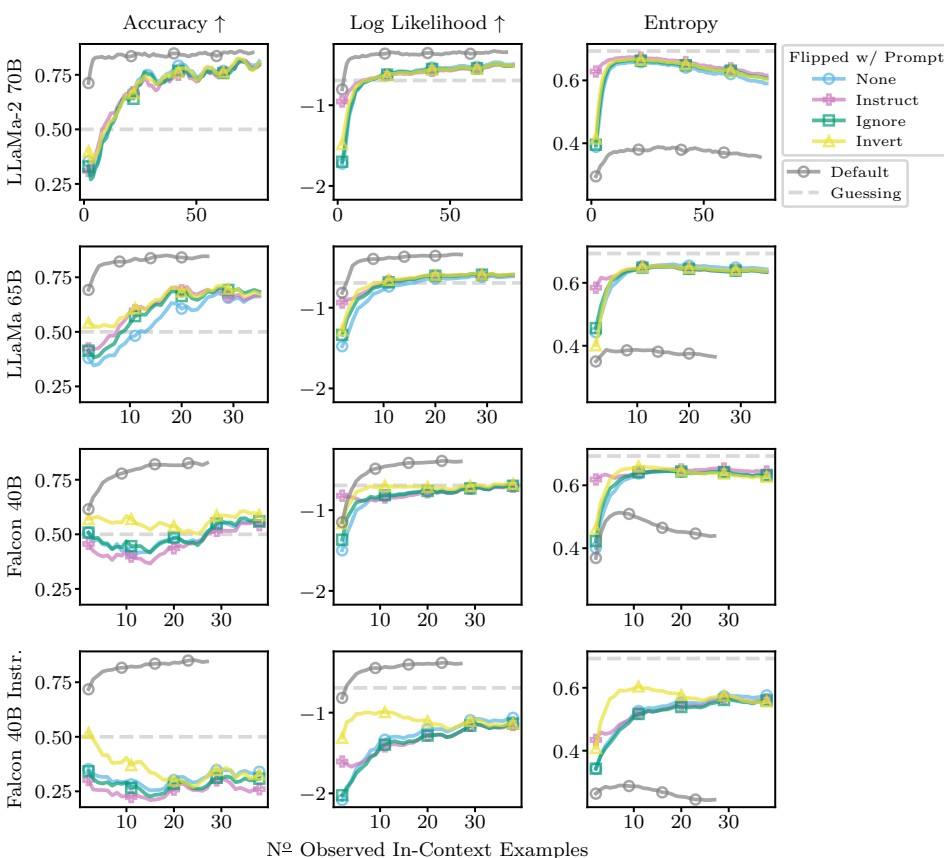

Figure F.41: Prompted few-shot ICL with flipped labels on **Hate Speech**. Some prompts are able to improve ICL on flipped labels compared to not using a prompt as before (label none in the figure). We average over 100 random subsets and then additionally apply moving averages (window size 5) for clarity.

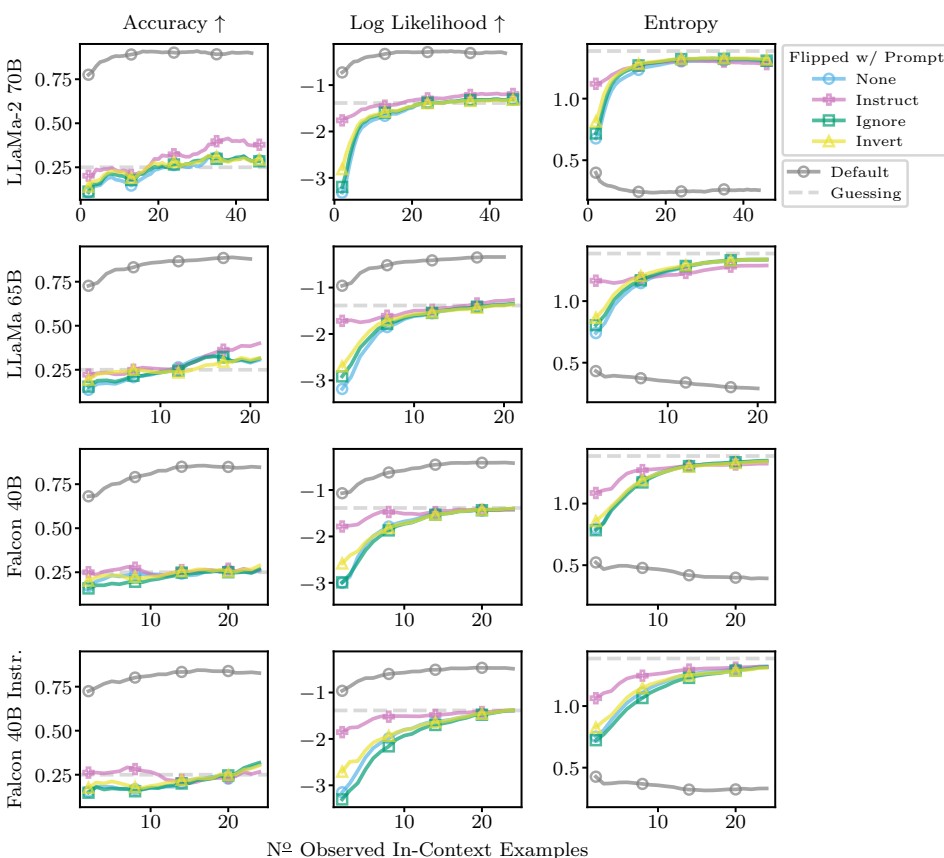

Figure F.42: Prompted few-shot ICL with flipped labels on **AG News**. Some prompts are able to improve ICL on flipped labels compared to not using a prompt as before (label none in the figure). We average over 100 random subsets and then additionally apply moving averages (window size 5) for clarity.

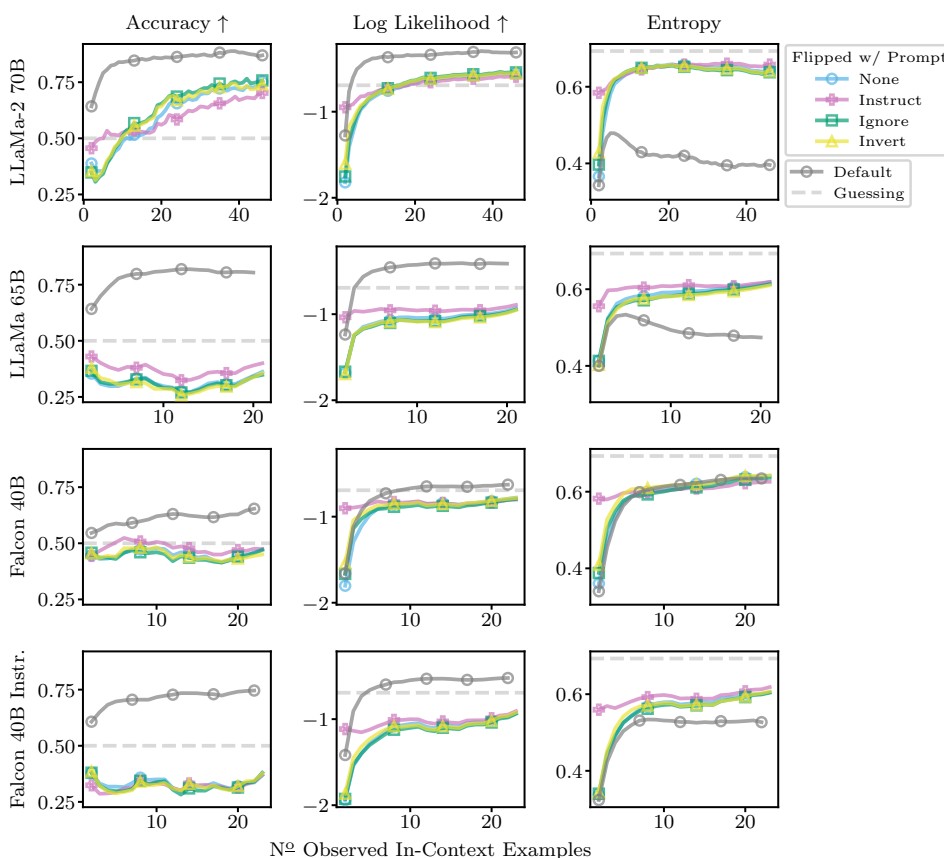

Figure F.43: Prompted few-shot ICL with flipped labels on **Medical Questions Pairs**. Some prompts are able to improve ICL on flipped labels compared to not using a prompt as before (label none in the figure). We average over 100 random subsets and then additionally apply moving averages (window size 5) for clarity.

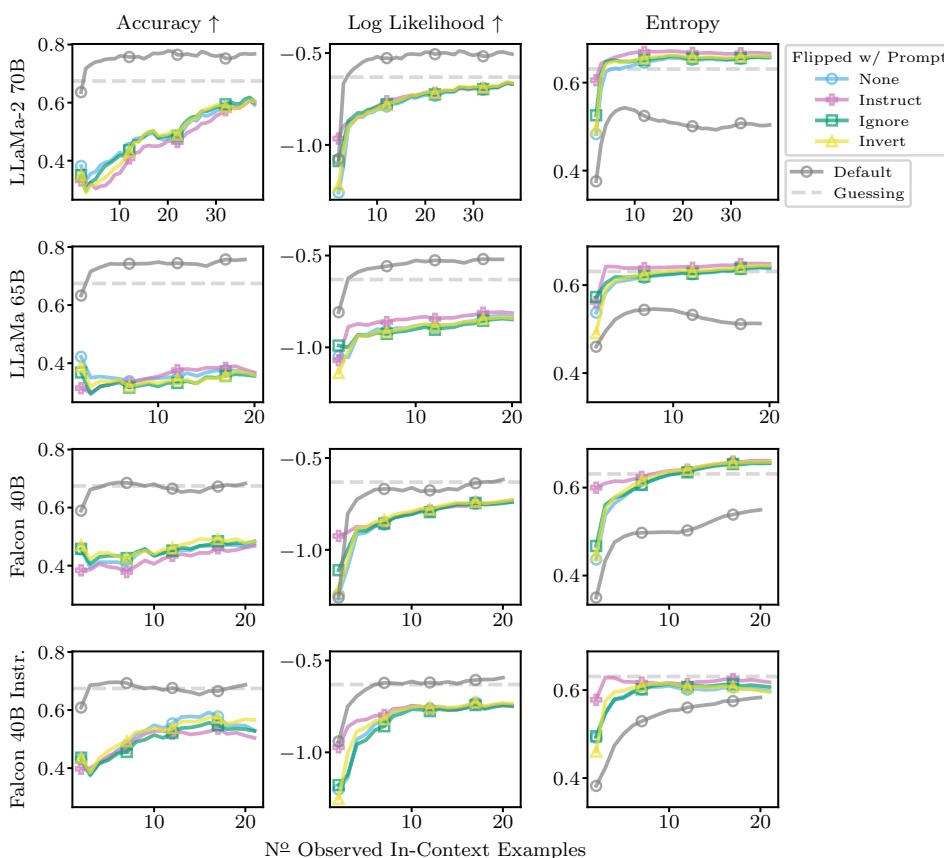

Figure F.44: Prompted few-shot ICL with flipped labels on **MRPC**. Some prompts are able to improve ICL on flipped labels compared to not using a prompt as before (label none in the figure). We average over 100 random subsets and then additionally apply moving averages (window size 5) for clarity.

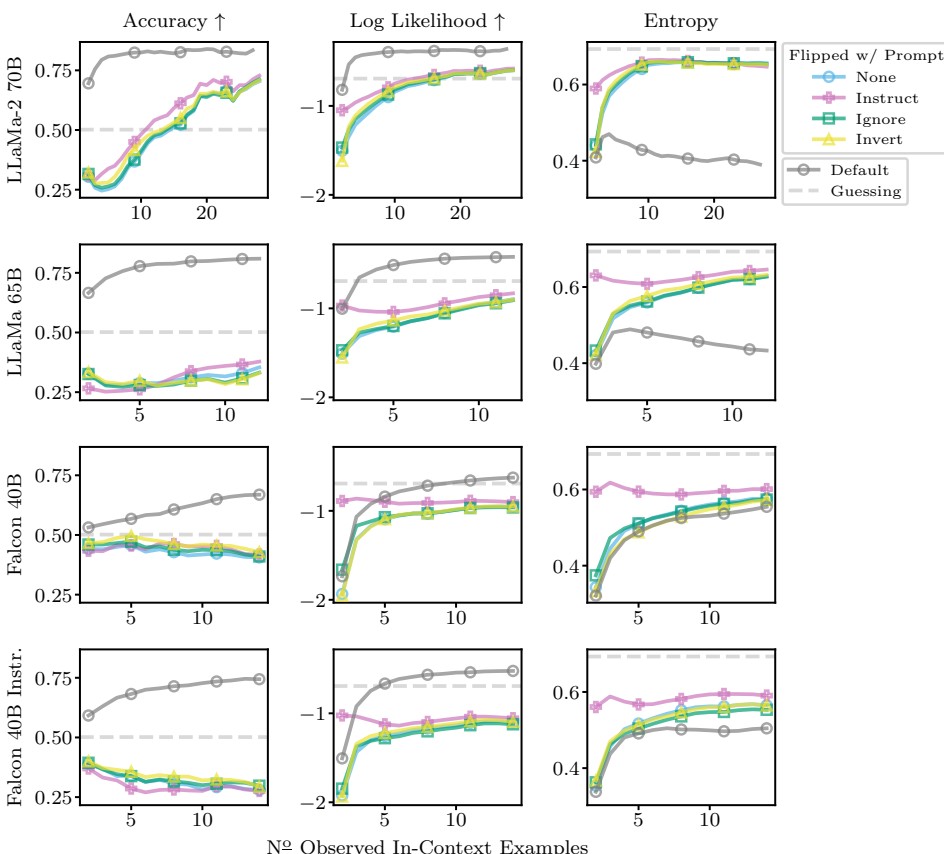

Figure F.45: Prompted few-shot ICL with flipped labels on **RTE**. Some prompts are able to improve ICL on flipped labels compared to not using a prompt as before (label none in the figure). We average over 100 random subsets and then additionally apply moving averages (window size 5) for clarity.

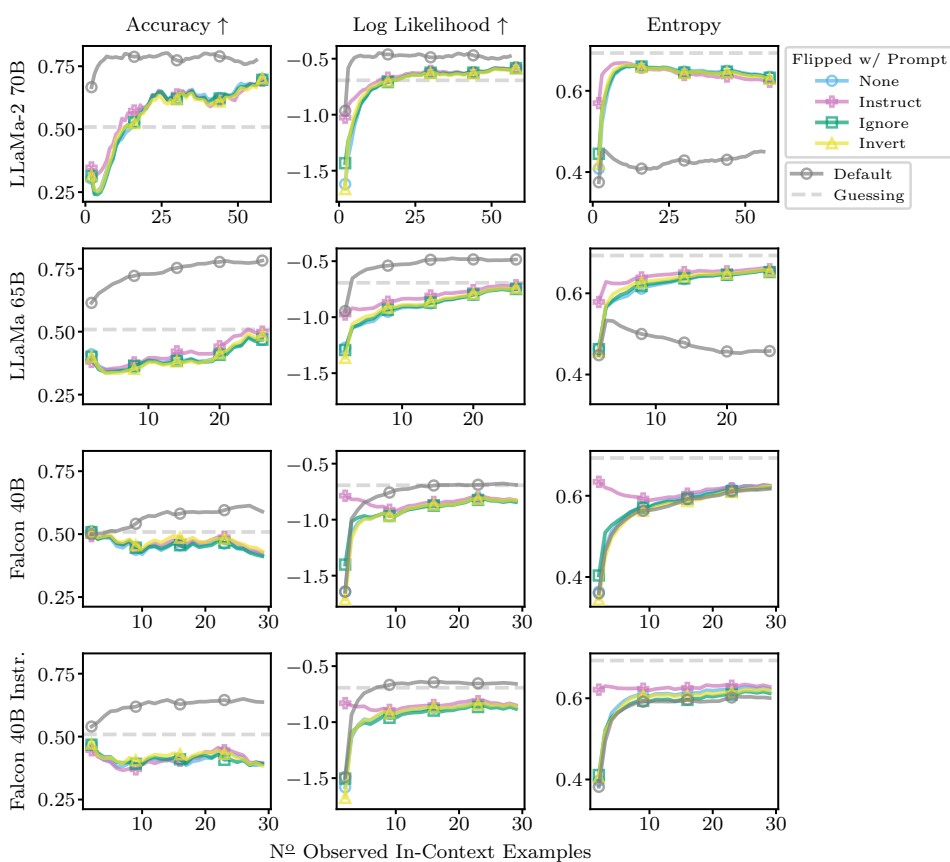

Figure F.46: Prompted few-shot ICL with flipped labels on **WNLI**. Some prompts are able to improve ICL on flipped labels compared to not using a prompt as before (label none in the figure). We average over 100 random subsets and then additionally apply moving averages (window size 5) for clarity.

## G   ADDITIONAL EXPERIMENTS

In this section, we provide additional experiments and insights into label learning in ICL.

### G.1   PROGRESSIVE RANDOMIZATION

Previously, we have observed that ICL performance suffers when *all* labels in the context are randomized. In this section, we study the changes to ICL predictions when *gradually* increasing the proportion of in-context examples with random labels. This is of practical relevance as noisy labels are a common concern in many applications.

Figure G.1 gives the performance of ICL for LLaMa-2-70B on SST-2 at different noise levels. We observe that probabilistic log likelihood immediately and clearly degrades, even at our smallest proportion of random labels of 0.2. This makes sense intuitively, as we would expect model predictions to become less certain in the face of noisy label observations. In contrast, accuracy stays relatively constant until the noise level reaches a proportion of 0.6, i.e. more than half the in-context points are noisy.

### G.2   CALIBRATED LABEL FLIPPING

We here repeat our label flipping experiments from §7 regarding NH2 with probabilities calibrated according to the approach of Zhao et al. (2021) to see if *calibrated* predictions can overcome the model's pre-training preference.

Figure G.2 is a reproduction of Fig. 5 with calibrated probabilities. We observe that differences between the calibrated and uncalibrated versions are generally small: we find almost no differences for SST-2/Falcon-40B and small improvements for all labels for Hate Speech/LLaMa-65B and MQP/LLaMa-2-70B. Most importantly, we still observe large differences in performance between the different label setups at maximum context size, such that, here, NH2 remains rejected for ICL with calibrated predictions.

We do not necessarily find these results surprising. Zhao et al. (2021) focus on scenarios with small in-context dataset sizes (up to 16). They show that this is where calibration is most important. Already for 16 in-context demonstrations, the gains from calibration are insignificant, see, e.g. their Figure 1. In contrast, we study ICL at much larger numbers of in-context demonstrations. Thus, we are not surprised that our conclusions do not change with calibration.

### G.3   LABEL FLIPPING FOR AUTHOR IDENTIFICATION

In Fig. G.3, we present results for our label flipping experiments (NH2, §7) with LLaMa-2 models on our novel author identification task (§6). Generally, we observe that accuracy is very similar across different label scenarios. However, small differences are visible for the probabilistic metrics (most clearly for entropy). Interestingly, these differences are much smaller than in most of our previous experiments. This provides additional support for the intuitions underlying §7: we would expect the

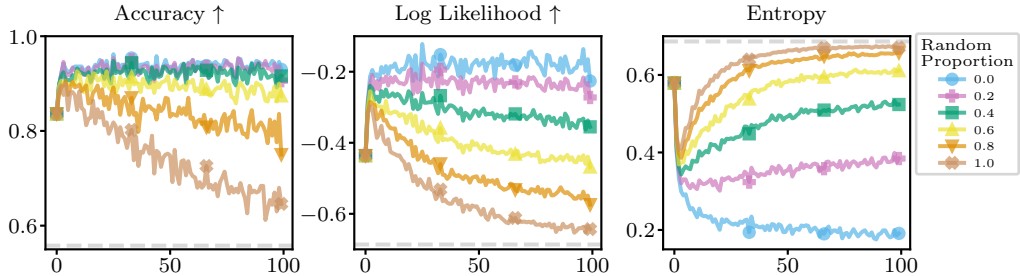

Figure G.1: Few-shot ICL with **different proportions of randomized labels** for SST-2 and LLaMa-2-70B. As the proportions of random labels increases, predictions become less certain. Accuracies stay relatively constant, until label noise increases above 50 %. We average over 500 random in-context datasets.

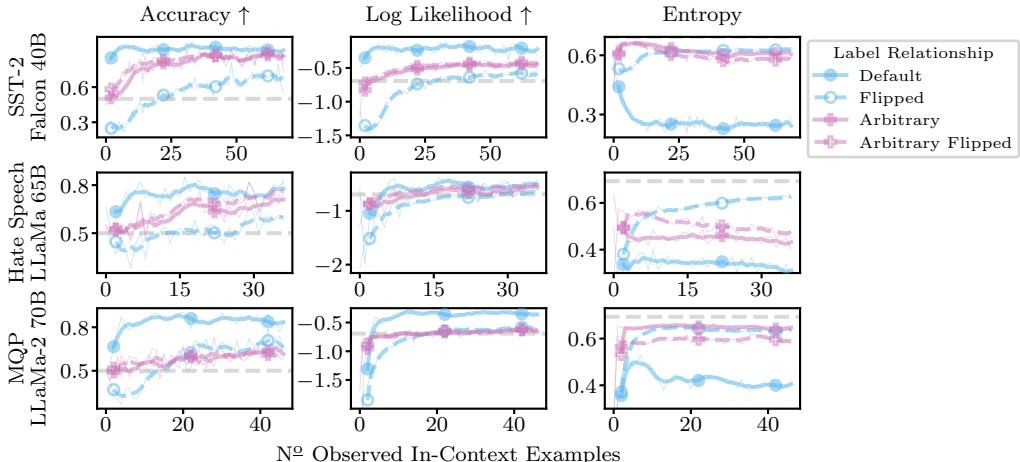

Figure G.2: Reproduction of Fig. 5 with **calibrated probabilities according to Zhao et al. (2021)**. Few-shot ICL with replacement labels for Falcon-40B on SST-2, LLaMa-2-65B on Hate Speech, and LLaMa-2-70B on MQP. Results are largely similar to the original figure Fig. 5 and NH2 remains rejected. Averages over 100 runs and thick lines with moving average (window size 5).

model to only suffer significantly from flipped or replaced in-context labels if the pre-training data actually suggests to prefer a particular label setup over another. For the author identification task, which is guaranteed to not be part of the training data (cf. §6), it makes sense that ICL would have a much weaker preference for any particular label setup.

We can also discuss the small differences that we do observe between label setups: (1) Especially initially, the performance for the arbitrary/arbitrary flipped labels, i.e. A/B or B/A, can be lower than the performance for the default/flipped labels, which are the authors' first names. It seems the model prefers assigning the sentences to 'names' rather than 'symbols'. It makes sense that this is a bias that would emerge from the pre-training data. (2) Further, we observe that the larger LLaMa-2 models can slightly prefer the default label direction. We believe that a possible explanation for this is that, sometimes, the sentences in our dataset contain the authors name, e.g. 'Hi Author 1' in a sentence from Author 2. This could confuse the model for the flipped label direction, in particular, if the model has some general understanding of author identification tasks. Lastly, we note that, to reject NH2, it is sufficient to find *any* dataset on which it does not hold, such that our conclusions in §7 remain valid regardless.

## G.4 ANSWER IN CONTEXT

To better understand how ICL leverages in-context label information, in this section, we study ICL predictions when one (or multiple) of the in-context examples exactly match the test query. That is, the model can always achieve perfect accuracy on the test query by looking up the label of the exact match in the in-context training set. We examine this across different label setups to investigate how pre-training preferences affect prediction behavior of the model here.

Figure G.4 shows results for SST-2 at 10 in-context examples averaged across our selection of models. We find the following behavior: (1) A single repetition of the test query (in the training set) increases performance for arbitrary and flipped labels but not for default labels, where performance is high already. Further, absolute performance remains highest for the default labels. We do not observe perfect accuracy, i.e. perfect copying behavior, for any of the label setups. This means ICL does not implement a nearest neighbor predictor, and single observations have but a limited influence on the decision function. This behavior seems generally sensible, in particular, if the model expects that single examples can be mislabelled. (2) For multiple (here, four) repetitions of the test query we observe perfect accuracy for all label setups. Variance across the different models is negligible. This behavior is reasonable, as we would expect that multiple observations of the same input with consistent labeling should lead to a confident prediction of that label for that input.

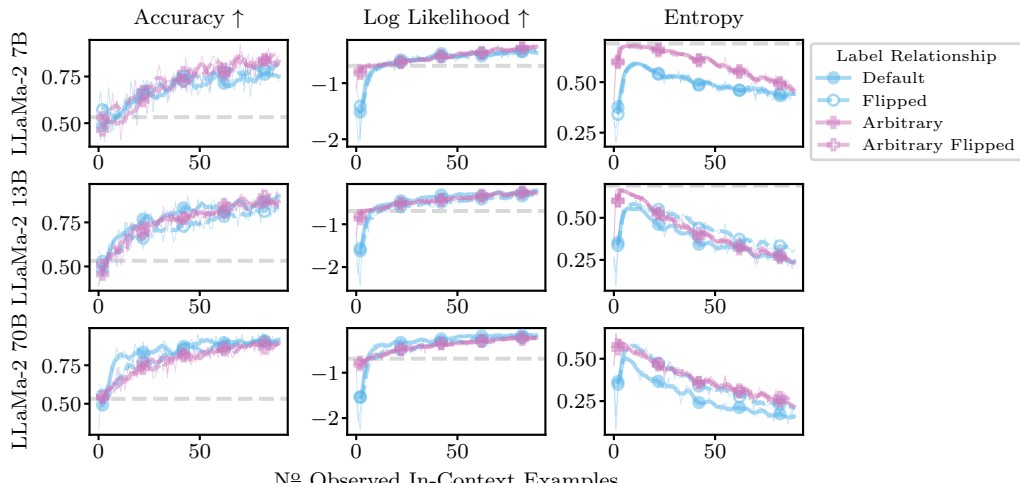

Figure G.3: Few-shot ICL with **replacement labels** for **LLaMa-2** at different sizes on our novel **author identification** task. We find that differences between different label setups are much smaller than in most previous experiments. This conforms to our expectations: the author identification task is purposefully chosen such that it is novel, and as such, we would not expect the LLMs to have a strong preference towards any particular label relationship. Averages over 100 runs and thick lines with moving average (window size 5).

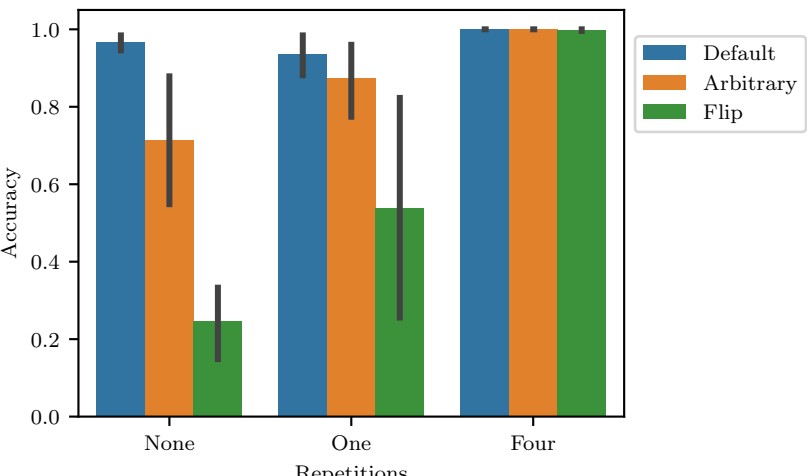

Figure G.4: Few-Shot ICL for **answer in context** experiment for **SST-2** averaged across our usual selection of models and for three different label setups. For each input, we add (No, One, Four) 'repetitions' of the exact test query to the in-context examples. For one or more repetitions, ICL could achieve perfect accuracy by looking up the exact match and predicting its label. We find that one repetition improves performance for arbitrary or flipped replacement labels, but that robust copying behavior, i.e. perfect accuracy, emerges only for multiple repetitions of the test query.

