# OpenReview forum: "In-Context Learning Learns Label Relationships but Is Not Conventional Learning"
_ICLR.cc/2024/Conference — ICLR 2024 poster_

### Official Review · Reviewer_Sjxd · 2023-10-27

**Soundness:** 3 good
**Presentation:** 4 excellent
**Contribution:** 2 fair
**Rating:** 6
**Confidence:** 4

**Summary:**

This paper explores how label information is used within ICL for LLMs. Experiments suggest that predictions given ICL almost invariably rely on the labels provided in-context. They also suggest that novel tasks can be acquired given this simple technique. In contrast, priors in the model remain hard to overcome, and statistical biases remain in the use of particular subsets of ICL components. Experiments are run on (technically) three families of LLM.

**Strengths:**

- The appendices appear very numerous, although it was not possible to read the entirety of the ~60 pages for this review and, since most of these are in the form of (at least superficially) very similar plots, the depth of their informativeness is not clear.
- A good number (perhaps moderate variety) of LLMs are considered, and a very good number of tasks.
- The inclusion of observations and discussions directly with the associated questions was appreciated, especially with regards to previous work (e.g., Min et al (2022b)).
- Section 8, regarding NH3 (which is not really treated as a null hypothesis, admittedly) is convincing.

**Weaknesses:**

- Several of the outcomes of this paper are, admittedly, already examined in some of the background work described in Sec 2 (and elsewhere), e.g., Zhao et al (2021). It may be advised to add differentiating factors with previous work directly into Sec 2.
- Additional differences exist between this work and that of Zhao et al (2021; e.g., the models are entirely different) so a direct comparison in Sec 8 of this type is somewhat disingenuous
- Although not a direct effect of this paper, the extent to which these particular results will generalize to future models is somewhat limited. On a positive note, the methods used to obtain these results may be applicable for some time.
- Some of the arguments seem to fall to the ‘hasty generalization’ fallacy. E.g., claims are made about ICL in general (at least within the context of LLMs), but really only specific examples are applicable. This is most notable on the experimentation around NH2, which is perhaps the weakest of the three so-called null-hypothesis experiments.

**Questions:**

- Can you please check your work for grammatical errors (e.g., ‘we rephrase these them as…’) and formatting errors (e.g., capitalization in references can be resolved by adding {} appropriately)?
- Would it have been more appropriate had Null Hypothesis 2 ($H^{(2)}_0$) been the reverse — that ICL cannot overcome 0-shot prediction preferences. Or is the original version of “NH2” even a null hypothesis in the statistical sense, or merely a proposition that can be proved or disproved by example? It seems like this was set up with the expected outcome in mind.
- Is it really necessary to examine _all possible_ numbers of demonstrations if the behaviours are so macroscopically clear?
- Is it really the case that the authorship identification task is truly novel? It could be confidently assumed that neither the particular messages were used in training the LLMs, nor even the particular authors, but author attribution itself is something LLMs learn implicitly, and ‘authorship embeddings’ of the authors of this paper are surely plausibly part of decoding/inference? I.e., this may be a novel dataset (which is not uncommon to see in NLP writ large), but how is this a novel _task_?
- If you couch your _research questions_ as null hypotheses, why do you not do any significance testing?

---

> ### Author Response · Authors · 2023-11-15
> **Author Response to Reviewer Sjxd (Part 1/3)**
>
> Thank you for your hard work and helpful feedback! We hope that our responses below and the accompanying paper updates alleviate your concerns.
>
>
> > Several of the outcomes of this paper are, admittedly, already examined in some of the background work described in Sec 2 (and elsewhere), e.g., Zhao et al (2021). It may be advised to add differentiating factors with previous work directly into Sec 2.
>
>
> Thanks for this suggestion! We will happily highlight differences between our submission and prior work directly in Section 2. In particular, we think there are a variety of important differences between our experiments and those of prior work, e.g. those of Zhao et al. (2021) (see below).
>
>
> > Additional differences exist between this work and that of Zhao et al (2021; e.g., the models are entirely different) so a direct comparison in Sec 8 of this type is somewhat disingenuous
>
>
> Apologies, but we believe this might be a misunderstanding. Our results in Section 8 are not a direct comparison between our results and those of Zhao et al (2021).
> In fact, we believe that Zhao et al. look at different and complementary ways in which the distance of examples to the test query, i.e the ‘recency’, affects ICL predictions.
> We already discuss this a bit at the end of Section 8 but are happy to elaborate here and in the next version of the draft.
>
>
> Zhao et al. (2021) observe a ‘recency bias’ in the predictions of ICL, i.e. they find that ICL predicts those labels more frequently that appear more often close to the query.
> For example, if the input is  $[..., [x_N-4, y_N-4=1], [x_N-3, y_N-3=0], [x_N-2, y_N-2=1], [x_N-1, y_N-1=1], x_N]$ the model predicts $y_N=1$ more often, regardless of $x_N$. And if the input is $[..., [x_N-4, y_N-4=0], [x_N-3, y_N-3=0], [x_N-2, y_N-2=1], [x_N-1, y_N-1=0], x_N$ the model predicts $y_N=0$ more often, regardless of $x_N$.
> Crucially here, all datapoints are drawn from the same fixed distribution $(X, Y)\sim p(X, Y)$, with a single fixed label relationship. Only the order of the datapoints is changed.
>
>
> In contrast, we study how ICL behaves if _multiple_ label relations are present in the input, e.g. we present the model with $[[x0, y0], …, [x_N/2, y_N/2], [x_N/2+1, y_N/2+1], …[x_N, y_N]]$, where $(x_0, y_0), \dots, (x_{N/2},  y_{N/2})\sim p_1(X, Y)$ and $(x_{N/2+1}, y_{N/2+1}), \dots, (x_{N}, y_N) \sim p_2(X, Y)$.
> Imagine that some of your in-context examples actually follow a different label relationship (e.g. due to a label distribution shift or noisy labeling): does it matter where these examples are in the context? This is what we study in NH3 by presenting the model with examples from two different label relations in the in-context demonstrations.
> Intuitively speaking, while Zhao et al. (2021) study if ICL has a recency bias for predicting a _particular_ label, we study if there is a recency bias for learning label _relationships_!
> These are two distinct biases: in our experiments, we avoid the recency bias from Zhao et al. (2021) by averaging over the order of our examples and labels.
> Lastly, we note that one cannot follow from the Zhao et al. (2021) results that a recency bias for learning label relationships exists, and the mechanisms underlying the two biases are likely different: while Zhao et al. (2021) seem to suggest that ICL uses a simple heuristic to predict (looking indiscriminately at the last few labels), this does not explain our label learning bias.
>
>
> > Although not a direct effect of this paper, the extent to which these particular results will generalize to future models is somewhat limited. On a positive note, the methods used to obtain these results may be applicable for some time.
>
>
> Thanks, we agree this could be seen as a limitation and are happy to make this more explicit in the paper. However, we also think this is not really something that can be solved, and so, we believe, should not count too strongly against our submission.
> The best we can probably do is compare performance across model sizes, where we do observe some trends (such as larger models being more susceptible to random labels and more willing to flip labels) which _might_ give insight into what behavior to expect in the future.
> Regardless, given their widespread use, we do think it is important to publish work on the behavior of these particular models.
>
>
> (Please read part two of our response next.)

---

> ### Author Response · Authors · 2023-11-15
> **Author Response to Reviewer Sjxd (Part 2/3)**
>
> (Please read part 1 of our response first.)
>
> > Some of the arguments seem to fall to the ‘hasty generalization’ fallacy. E.g., claims are made about ICL in general (at least within the context of LLMs), but really only specific examples are applicable. This is most notable on the experimentation around NH2 [...]
>
>
> Thanks for pointing bringing this to our attention. While we believe that our experiments cover a wide variety of currently relevant ICL scenarios, it is not at all in our interests to make inappropriate claims about their generalization. We have generally tried to be quite careful about this, e.g. we preface our rejection of NH2 on page 7 with ‘For the models we study’, as, particularly here, we felt results could change with future models.
> However, if you have any particular cases in mind where you felt our phrasing is too strong, we would be glad to improve these. We could also add the above discussion to our “Limitations” section, if you think this would be helpful.
>
>
> > Can you please check your work for grammatical errors
>
>
> Thanks for spotting these! We will fix these errors and will make sure to thoroughly check for any other errors before the final submission.
>
>
> > Would it have been more appropriate had Null Hypothesis 2 been the reverse— that ICL cannot overcome 0-shot prediction preferences.
>
>
> Good question!  We believe that our framing of NH2, that ICL _can_ overcome the zero-shot prediction preferences of the pretrained model, is more appropriate. That is, because, if NH2 is true, then there should eventually be _no_ difference in the predictive distributions between the flipped label and default label predictions. If NH2 is false, we will observe differences in predictions, and can thus reject the null hypothesis. That is, if we set up NH2 in this direction, it corresponds to the standard scenario where the null hypothesis assumes two distributions are the same, and we can reject it if they are not.
>
>
> > Is it really necessary to examine all possible numbers of demonstrations if the behaviours are so macroscopically clear?
>
> Apologies, but could you clarify which behaviors exactly ‘are so macroscopically clear’?
>
>
> More generally, we think it _would_ be sufficient to replace our figures with ones that have only half (or a third/fourth/fifth) of the resolution of the x-axis, i.e. only half (third/…) the numbers of in-context demonstration sizes.
>
>
> However, common practice is to only choose _very few_, e.g. one to three different numbers of few-shot demonstrations. For example, the Min et al. (2021) paper considers only the k=16-shot scenario. It is this practice that can be insufficient to draw general conclusions about ICL behavior. Considering multiple different numbers of demonstrations will often reveal differences in ICL behavior, with the effects of labels typically being larger at larger numbers of demonstrations. In particular, prior work often does not consider large numbers of in-context observations, and thus underestimates the effects labels can have on ICL. Lastly, note that we incur no extra cost from obtaining ICL behavior at all numbers of demonstrations due to our novel evaluation strategy that we describe briefly in Section 4 and more extensively in Appendix B.
>
>
> (Please read part three of our response next.)

---

> ### Author Response · Authors · 2023-11-15
> **Author Response to Reviewer Sjxd (Part 3/3)**
>
> (Please read part two of our response first.)
>
> > Is it really the case that the authorship identification task is truly novel? It could be confidently assumed that neither the particular messages were used in training the LLMs, nor even the particular authors, but author attribution itself is something LLMs learn implicitly, and ‘authorship embeddings’ of the authors of this paper are surely plausibly part of decoding/inference? I.e., this may be a novel dataset (which is not uncommon to see in NLP writ large), but how is this a novel task?
>
>
> Thanks for your comment! You are touching on an important topic here that we would be glad to discuss more in an updated version of the draft. We agree that there are nuances to how ‘novel’ a task is, and, clearly, there are limits to the tasks that ICL can learn in-context. We already mention this briefly in the second paragraph of Section 6 (‘while the LLM could have some general notion of authorship identification tasks, the specific input-label relationship is definitely novel’) but we are more than happy to elaborate here.
>
>
> We agree with you that the LLM might have already learned how to solve general author attribution tasks. And, if this were the case, this would help the LLM _learn_ to predict on our specific author identification task. Whether or not you call this a ‘novel task’, or perhaps just a ‘novel task instance’, comes down to semantics. For our part, we define two tasks as different if the distribution of their inputs and outputs are different. In that sense, the author identification task _is_ a novel task, as the label distribution Y|X that ICL has to learn to succeed is novel and unlike other previous distributions the model may have implicitly picked upon during pre-training. For example, if we assume the model has been trained on the Harry Potter books, it might already have implicitly learned to solve the author attribution task Y|X for Y in [Harry, Hermione] and X being sentences of the Harry Potter books.
> However, this different to the label distribution to Y|X for Y in [Author 1, Author 2] of this paper and X being private direct messages.
>
>
> It is important to recall that our experiments in Section 6 are in the context of NH1 and prior work, which state that ICL does not depend on the labels of in-context examples at all. This, clearly, is refuted by our experiments in Section 6. For our purposes, we are content to show they can learn _any_ novel task, i.e. learn a novel label relationship from context. We agree there are nuances to ‘how novel a task’ is, and clearly, there are limits to the tasks ICL can learn in-context, and we are more than happy to see add some of these nuances to the revised version of the paper.
>
>
> > If you couch your research questions as null hypotheses, why do you not do any significance testing?
>
>
> We _do_ actually perform significance testing: in Tables 1 and 2, we highlight results that are statistically significant with bold numbers. In the appendix, we give extended versions of these tables (Tables F.1 and F.2) that give the standard errors, and we also detail how we compute statistical significance in Appendix E.
>
>
> Again, thank you very much for your review. We hope that our reply has addressed any remaining concerns. Please let us know if there are any further changes you would like to see or if there is anything else that we can clarify.

---

> > ### Comment · Reviewer_Sjxd · 2023-11-15
> > **Raise of score, but not to '8'**
> >
> > The authors responded with great depth and accuracy to my comments and to my questions. I would increase my score from 6 to 7, but there is no 7 available.

---

> > > ### Author Response · Authors · 2023-11-15
> > > **Author Response**
> > >
> > > Thanks for your lightning-fast reply. We are glad to see you liked our rebuttal!
> > >
> > > > I would increase my score from 6 to 7, but there is no 7 available
> > >
> > > Thanks, we appreciate that!
> > >
> > > We would just like to point out that the rating system seems to be a little different than usual.
> > > Normally, a 7 would be 'normal accept' with an 8 being reserved for papers that should be 'highlighted'.
> > > But in this instance, it looks like 8 is the first real accept score, and the 10 is reserved for papers that should be 'highlighted'.
> > >
> > > If you feel like our paper should be accepted, we would appreciate it if you would consider raising your score to an 8 given the unusual score meanings.

---

### Official Review · Reviewer_tkcu · 2023-10-30

**Soundness:** 2 fair
**Presentation:** 3 good
**Contribution:** 2 fair
**Rating:** 6
**Confidence:** 5

**Summary:**

This paper offers an empirical study into characteristics of ICL compared to in-context learning. The authors start with 3 null hypotheses about how ICL works, and empirically address those hypotheses.
H1. ICL does not learn p(yIx). This Hypothesis is rejected through an experiment with label randomization.
H2. ICL can overcome model priors that gained through pre-training about label semantics. This hypothesis is rejected because changing label words in ICL examples to have neutral or opposite meanings does affect the performance of the model.
H3. ICL example order does not matter. This hypothesis is rejected through an experiment where labels of some ICL examples are corrupted in different positions of the input sequence and the performance is shown to depend on which ICL examples were corrupted.

The paper concludes that ICL is in some ways similar to conventional learning algorithms (H1) but different in other ways (H2 and H3).

**Strengths:**

This is clearly written paper that is easy to follow, with plots that tell the story well. The question of how ICL really works is an open question in the field that has not been addressed sufficiently yet. The authors go beyond using only accuracy and use several metrics to measure the model performance before and after ICL (log likelihood and entropy). The results are reported on models with various sizes.

**Weaknesses:**

The paper is framed as a comparison between ICL and conventional learning and claims that hypotheses 2 and 3 reveal differences between ICL and conventional learning. I do not agree with this for the following reasons:
H2. The empirical observations corresponding to this hypothesis are not novel and not limited to ICL. For example, in "Making Pre-trained Language Models Better Few-shot Learners", Gao et al show that even fine-tuning the models with flipped or neutral labels degrades their performance, so even conventional learning is not able to overcome the pre-training priors.
H3. Sample order does matter in conventional learning too, that is why we shuffle the training data. In addition, these experiments seem to be a re-discovery of the recency bias in LLMs, for example studied in "Calibrate Before Use: Improving Few-Shot Performance of Language Models", Zhao. et al

Lastly, one major claim of the paper is that although flipping/randomizing labels of ICL examples sometimes does not affect accuracy of the model much, but probabilistic metrics such as log likelihood or entropy reveal larger gaps in performance. But this claim is not quantified well, because accuracy has a clear and interpretable range but the same cannot be said about loglikelihood.




Minor writing suggestions:
- "expresses the sentiment" --> the word sentiment not appropriate for describing a scientific paper's conclusion
- "This is important when the context contains diverging information about a label relationship" --> unclear

**Questions:**

A hopefully minor concern I have about this work is I see a lot of traditional NLP benchmark tasks e.g. GLUE tasks being used against modern models such as LLAMA. Do we know if GLUE benchmark was present in LLAMA's training data? Could the paper's conclusions be different if newer harder benchmarks were used?

Did you try the novel task of author identification in the label flipping experiments as well? If so, can we see some examples of the author-id prompts as well as results regarding all hypotheses? I am interested in the results using novel tasks because as the authors mentioned in the paper, in that case there is less chance of contamination in the models' pretraining data.

---

> ### Author Response · Authors · 2023-11-15
> **Author Response to Reviewer tkcu (Part 1/3)**
>
> Thank you for your hard work and helpful feedback! We hope that our responses below and the accompanying paper updates alleviate your concerns.
>
>
> > The paper is framed as a comparison between ICL and conventional learning and claims that hypotheses 2 and 3 reveal differences between ICL and conventional learning. I do not agree with this.
>
>
> Apologies, but we believe your concerns here are (1) based on a misunderstanding about what we mean by ‘conventional’ learning algorithms and (2) tangential to our actual key contribution, which is to contribute to a better understanding of label learning in ICL.
>
>
> (1) You point out that finetuning of LLMs also does not conform to one of our null hypotheses. We agree this is the case and interesting related work (see next reply). However, we would also not consider this an instance of ‘conventional learning’. Our concept of a conventional learner expresses how we would _want_ a learning algorithm to behave based on classical intuitions about learning, e.g. we would want the learning algorithm to eventually predict according to the training data and overcome any ‘priors’. We could also call them ‘idealized learning algorithms’ if you think this is more clear, and are happy to clarify this in the next version of the draft.
>
>
> (2) Ultimately, however, we believe your concerns here are tangential to our actual key contribution, which is to address the need for a better understanding of label learning in ICL through rigorous experiments. The shortcomings of other learning algorithms, such as finetuning of LLMs, are tangential here and do not negate the need to better understand ICL. This is the case because the mechanism of ICL is different from other established learning algorithms: e.g. observations from gradient-based finetuning do not necessarily transfer to ICL. Our results are important, in particular, because various previous works have been quick to reach questionable conclusions about the behavior of ICL.
>
>
> > For example, in "Making Pre-trained Language Models Better Few-shot Learners", Gao et al show that even fine-tuning the models with flipped or neutral labels degrades their performance, so even conventional learning is not able to overcome the pre-training priors.
>
>
> Thanks for suggesting this interesting reference. As previously explained, we believe the behavior of other particular learning algorithms, such as finetuning of LLMs, is largely tangential to our goal of learning about the behavior of ICL, or to our definition of conventional learners.  However, analogies between ICL and finetuning of LLMs are generally interesting at a higher level, and we will be glad to add this to our discussion of related work.
>
>
> >  Sample order does matter in conventional learning too, that is why we shuffle the training data.
>
>
> We think this actually demonstrates our previous point well. A fundamental assumption in machine learning is that the data is independent and identically distributed (IID). Thus, we _want_ our learning algorithms to be independent to the order of the samples. Again, the fact that some learning algorithms fail to achieve this in practice, is independent to our investigations into ICL, which deserves to be studied on its own.
>
>
> (Although, technically, we note that in the example that you mention, the shuffling of the data (over multiple epochs of gradient-based optimization) actually does lead to approximate invariance to example order. As the optimization routine is part of the learning algorithm, we would thus actually say that many, if not most, practical instances of learning are approximately invariant to example order.)
>
>
> We hope our remarks here clarify this point and we will update the draft to include the discussion.
>
>
>
> (Please see part two of our response next.)

---

> ### Author Response · Authors · 2023-11-15
> **Author Response to Reviewer tkcu (Part 2/3)**
>
> (Please read part one of our response first.)
>
>
> > these experiments seem to be a re-discovery of the recency bias in LLMs, for example studied in "Calibrate Before Use: Improving Few-Shot Performance of Language Models", Zhao. et al
>
>
> We disagree that our experiments are a re-discovery of the results from Zhao et al. (2021), although we agree this is an important concern to clarify.
> We are aware of the experiments by Zhao et al. and briefly discuss them in the context of our paper at the end of Section 8.
>
>
> We believe that both the (1) experiments and (2) findings from NH3 in our paper are distinct from and complementary to those of Zhao et al. (2021).
> (1) While Zhao et al. (2021) manipulate the _order_ of in-context examples, we manipulate the _label relationship_ of the examples.
> (2) Zhao et al. (2021) conclude that ICL predicts those labels more frequently that appear more often close to the query – for a single fixed label relationship across all in-context examples. In contrast, our results show that ICL prefers to learn the _label relationship_ of those examples that appear closer to the query. Importantly, the bias from Zhao et al. (2021) is complimentary to the bias we find: both biases can _co-exist_. We avoid the bias from Zhao et al. (2021) in our experiments by averaging over different radom sets of in-context examples.
>
>
> Our experiments in NH3 study the scenario where some of the in-context examples follow a different label relationship (e.g. due to a label distribution shift): _how_ does the order of these examples affect predictions? The results from Zhao et al. (2021) are not sufficient to answer this question. Lastly, we note that the mechanisms underlying the two biases are likely different: while Zhao et al. (2021) seem to suggest that ICL uses a simple heuristic to predict (looking indiscriminately at the last few labels), this does not explain our label learning bias.
>
>
> If you think this is helpful, we would be more than happy to add the above discussion to an updated version of the draft.
>
>
> > Lastly, one major claim of the paper is that although flipping/randomizing labels of ICL examples sometimes does not affect accuracy of the model much, but probabilistic metrics such as log likelihood or entropy reveal larger gaps in performance. But this claim is not quantified well, because accuracy has a clear and interpretable range but the same cannot be said about loglikelihood.
>
>
> A key take-away from our work is that a focus on accuracy is one reason why previous work has reached undependable conclusions about the behavior of ICL. Our work shows that probabilistic quantities, such as log likelihood and entropy, can reveal significant additional behavior compared to accuracy, and thus should be part of any thorough analysis. For example, they can reveal statistically significant differences between predictions when accuracy cannot. This is independent from any potential concerns about their interpretability.
>
>
> For example, in Section 5, we want to know if predictions change when labels are randomized. The average absolute difference in metrics _relative to the standard error_ determines statistical significance. This ratio is smaller for accuracy than for log likelihood and entropy. When we compute the average value of this ratio for the experiments of Section 5 (see Table F.1 for individual means and standard errors), we find it is 6.2 for accuracy, 9.8 for log likelihoods, and 23.3 for entropy. Hence, accuracy is worse at finding differences in predictions because it changes less relative to its variance. Table F.1 also shows a variety of examples where differences are only significant when looking at probabilistic metrics. Lastly, we also refer you to Figure F.10 for an intuitive example of the higher variance of accuracy.
>
>
> We hope this clarifies your concern, and we will clarify our exposition of the advantages of probabilistic metrics in the draft to avoid any future misunderstandings.
>
>
> > Minor writing suggestions:
>
>
> Thanks for spotting these! We have improved our wording in these instances in the updated version of the draft.
>
>
>
> (Please read part three of our response next.)

---

> ### Author Response · Authors · 2023-11-15
> **Author Response to Reviewer tkcu (Part 3/3)**
>
> (Please read part two of our response first.)
>
>
> > A hopefully minor concern I have about this work is I see a lot of traditional NLP benchmark tasks e.g. GLUE tasks being used against modern models such as LLAMA. Do we know if GLUE benchmark was present in LLAMA's training data?  Could the paper's conclusions be different if newer harder benchmarks were used?
>
>
> Thanks, this is a great question. We agree that potential overlaps between the training data and our tasks are an interesting aspect to consider. However, we are not aware of any positive findings in prior work for the tasks we consider.
> Those previous works that do study contamination, e.g. Brown et al. (2020, https://arxiv.org/abs/2005.14165) and Touvron et al. (2023, https://arxiv.org/abs/2307.09288), conclude that the effects of data contamination are likely negligible.
> Nevertheless, we agree that dataset contamination is a potential concern, which is why we designed the authorship-identification dataset (also see below), where we can be certain that the model has not seen the data before.
>
>
> Note that we have selected tasks that are representative of previous studies on ICL, e.g. there is high overlap between our tasks and those of Min et al. (2021). If you have any suggestions for other tasks you would like to see us study a particular experiment on, we would be more than happy to do so.
>
>
> Lastly, we note that some of our tasks are arguably quite ‘hard’ for ICL already. On MQP, MRPC, RTE, and WNLI, smaller models struggle to achieve better than random guessing performance. Larger models can eventually achieve better than random performance, but only with sufficiently many few-shot demonstrations. (It is thus unlikely, these tasks were memorized by the models.)
>
>
> > Did you try the novel task of author identification in the label flipping experiments as well?
>
>
> Thanks for this interesting suggestion! Following your feedback we have performed the label flipping experiments (NH2) with the author-id dataset across all LLaMa-2 models and added them to the revised version of the draft in Appendix G.3.
>
>
> In summary, we do still find that ICL treats different label setups differently, although differences are much less pronounced for the author-id task.
> This provide additional support for our intuitions behind Section 7: we would expect the model to only suffer significantly from flipped or replaced in-context labels if the pre-training data actually suggest to prefer a particular label relation. For the author-id task, which is _guaranteed_ to not be part of the training data, it seems likely the model should not have a strong preference for a particular label relationship.
> We refer you to Appendix G.3 for further discussion of these results.
>
>
> Thanks again for suggesting we perform these experiments, which we believe strengthen our submission!
>
> > If so, can we see some examples of the author-id prompts
>
>
> Thanks for this suggestion. We have added some examples from our novel author identification task to Appendix C.
>
>
>
> Thank you very much for your review. We hope that our reply has addressed your concerns. Please let us know if there are any further changes you would like to see or if there is anything else that we can clarify.

---

> > ### Author Response · Authors · 2023-11-20
> > **Discussion Period Ends Soon**
> >
> > Dear Reviewer tkcu,
> >
> > The discussion period ends in two days. We have made significant efforts to address your concerns with our extensive reply and additional experiments. We would greatly appreciate it, if you could engage with us and our rebuttal during the last remaining hours of the discussion period.
> >
> > In particular, we hope that our reply will be able to clarify your major concerns, which, we believe, stem from (1) a misunderstanding about what we mean by 'conventional learning algorithm' and (2) from a mischaracterization of our results in the context of prior work by Zhao et al.
> >
> >
> > Many thanks in advance
> >
> > The Authors

---

> ### Comment · Reviewer_tkcu · 2023-11-22
>
> I appreciate the authors' detailed response. I re-read some parts of the paper after reading your comments. In response to your two enumerated points regarding my main concern:
>
> 1- In the paper's introduction section, "conventional learning" is described as " conventional, general-purpose learning algorithms such as Bayesian inference or gradient descent". Additionally, Brown et al is cited as "[highlighting]
>  similarities between the behavior of ICL and conventional learning algorithms, such as improvements with model size and number of examples", which is another pointer to fine-tuning as an example of "conventional learning" since that is what Brown et. al compare few-shot and zero-shot settings to. So, the authors' response that they do not consider fine-tuning as an example of conventional learning is not consistent with the language of the paper in the introduction. Maybe a concrete scientific definition of what you mean by "conventional learning" is needed since it is such a key concept in the premise of the paper.
>
> 2- I completely agree with your point here. The problem is that the paper as it is currently written does not simply claim offering a better understanding of ICL, but rather claims discovering similarities and differences between ICL and "conventional learning". Again, I think this is a framing issue that can be resolved easily.
>
> Given the experiments and clarifications you have added I would be happy to raise my score to 6 if the framing could be fixed or conventional learning could be more precisely defined in the paper.

---

> ### Author Response · Authors · 2023-11-22
> **Author Response**
>
> We are pleased to see you appreciate our rebuttal and thank you for engaging with it.
>
>
> We agree the framing of conventional learning algorithms can be improved and we have gladly updated the paper to do this.  Namely, we have just uploaded a new version that implements the following improvements, highlighted in red in the paper for your convenience:
>
>
> * In Section 1, Paragraph 2, we have removed our early use of ‘conventional’ learning to avoid the potential confusions with our use of the term later on. Thanks for highlighting this!
> * In Section 1, Paragraph 3, we now make clear right away that the conventional learning algorithm is ‘idealized’, a ‘concept’, and that it ‘encodes our beliefs’ about how learning algorithms ‘should’ behave.
> * In Section 3, we have significantly rephrased our introduction of the ‘conventional learning algorithm’ in light of the above discussion.
> * In Section 10, Paragraph 1, we have added a citation of the Gao et a. (2021) paper that you mention.
>
> We hope these changes address your remaining concerns and are sufficient for you to be able to back acceptance as suggested. If there are any other improvements you would like us to implement, please let us know – we would be glad to do so.

---

### Official Review · Reviewer_WYZX · 2023-10-31

**Soundness:** 3 good
**Presentation:** 3 good
**Contribution:** 3 good
**Rating:** 8
**Confidence:** 4

**Summary:**

In-context learning is an important property of LLMs and is often utilized to improve performance. However, there is disagreement on how in-context learning works, with some papers drawing analogy to SGD, others to kernel regression, Bayesian inference, etc. To understand how ICL works, this paper identifies several assumptions of expected behavior of "conventional learning algorithms", e.g., models that learn P(y | x) from x_train, y_train ~ P and tests if ICL follows such behaviors. There are several questions proposed for the classification setting: do model predictions depend on ICL example labels? Can ICL work on tasks that are not seen in the pre-training corpus? Can ICL ignore bias from pre-training data? And does ICL treat all in-context information equally?

Through a systematic evaluation of several models and classification tasks, the paper concludes that 1) ICL depends on the in-context labels 2) ICL can work on tasks not seen in pre-training corpus 3) the pre-training preference matters, and 4) not all information is regarded equally. This refutes the findings of a previous paper (Min et. al.) and has implications for alignment and robustness of LLMs. More generally, it imposes some conditions on the loose analogy of ICL as a conventional learning algorithm.

**Strengths:**

Originality:
- Breaking down the understanding of ICL into parallels with conventional learning algorithms and experiments on label relationships is quite novel to me. It reminds me a bit of Zhang et. al (2016)'s famous paper on rethinking generalization.

Quality:
- Thorough experimental results, well-designed experiments.

Clarity:
- The paper was very clear and I enjoyed reading it overall.

Significance:
- The conclusions are pretty valuable and suggest that we should reevaluate our expectations of alignment and robustness of LLMs via ICL.

**Weaknesses:**

Originality:
- Some of the results on their own are not surprising, such as the model paying more attention to the last in-context example. However, I think the study overall is quite novel and rigorous.

Quality:
- For section 5, it would be nice to show how the model performance degrades as randomization increases. Similarly in section 7, you could try gradually mixing in the flipped labels or the arbitrary labels. I think this could help us study the robustness of ICL (in contrast to deep learning models or conventional learning algorithms), as in-context information can often be noisy in general. For instance, I am thinking of an example where the human prompter is giving examples of their preferences to align a model, and fails to accurately articulate one or two examples.
- Another experiment I'd be interested in seeing: can you study the Lipschitzness/label consistency of ICL? For example, you can come up with a setting like "x1 y1 x2 y2 .... xtest ?" where xtest is a perturbation of one of the x_i's (e.g. a slight rewording, or very close by in embedding space). Will the model predict ytest to be equal to y_i? And will this happen even if we scramble the labels? How different from x_i does x_test need to be in order for it to no longer be significantly influenced by a flipped y_i? Overall I think extending your study to consider the relationships among the x_i's and the x_test w.r.t. the label is interesting, since this is a property of conventional learning algorithms (inputs with similar feature vectors have similar labels).

Clarity:
- Minor nit: I'd prefer some more signposting about how a conventional learning algorithm is defined, or some notation of (x_ICL, y_ICL) versus conventional (x_train, y_train). Is a deep learning model considered a conventional learner if it can fit to arbitrary noise?
- In section 5 I'd appreciate an example of what randomizing the labels exactly means. I had it confused with the experiment in section 7, whose arbitrary (A, B) experiment could be thought of as randomizing the label space but not the label relationships.
- While I found the experiment in section 6 to be quite clever, I feel like "novel task we create" can be made a bit more explicit by specifying it's based on your private data. Without any elaboration, I was surprised about how you were so certain about the task.

**Questions:**

- This observation that larger models are more sensitive to label randomization is concerning. I would have expected that with more pre-training data and parameters, the model would be more confident in its prediction and less influenced by random noise in the in-context examples. On the other hand, I do buy the argument that for small models and hard tasks, changing the in-context examples significantly will still result in the model doing poorly, thus appearing to not be as impacted by the random labels. Do you have any thoughts on how to interpret this?

- See "Quality" section of Weaknesses. Would be curious to see what these behaviors are.

---

> ### Author Response · Authors · 2023-11-15
> **Author Response to Reviewer WYZX (Part 1/2)**
>
> Thank you for your hard work and helpful feedback! We hope that our responses below and the accompanying paper updates alleviate any remaining concerns you might have.
>
>
> > section 5, it would be nice to show how the model performance degrades as randomization increases. Similarly in section 7, you could try gradually mixing in the flipped labels or the arbitrary labels.
>
>
> Thanks for this great suggestion! We agree these are interesting experiments and now provide results for gradually increasing proportions of random labels in Appendix G.1. We observe that the degradation in model performance increases consistently with increased label noise for LLaMa-2-70B on SST-2. We would like to refer you to Appendix G.1 for further discussion of these results.
>
>
> We are currently working on providing these results for our usual selection of tasks and models. Lastly, we will also provide gradual label flipping results for the experiments in Section 7 as soon as possible. Given the compute requirements of our experiments and timeline of the discussion period, we hope for your understanding in this matter.
>
>
> > can you study the Lipschitzness/label consistency of ICL? For example, you can come up with a setting like "x1 y1 x2 y2 .... xtest ?" where xtest is a perturbation of one of the x_i's (e.g. a slight rewording, or very close by in embedding space). Will the model predict ytest to be equal to y_i? And will this happen even if we scramble the labels?
>
>
> Thanks for this suggestion! We agree this is an interesting experiment relevant to understanding how ICL learns label relationships. As a first step, in Appendix G.4, we have implemented an initial version where xtest is directly equal to one (or multiple) of the xi for the SST-2 dataset. In other words, we investigate to what extent the LLMs copy labels if there are exact matches between the in-context examples and the test query.
> We perform these experiments across three label setups: default labels, flipped labels, and arbitrary replacement labels.
> We find that one repetition improves performance for arbitrary or flipped replacement labels, but that robust copying behavior, i.e.~perfect accuracy, emerges only for multiple repetitions of the test query.
> Again, we would like to refer you to Appendix G.4 for an extended discussion of these results.
> We are currently working on extending these experiments to slight rewordings of the x_i and to our usual selection of tasks.
>
>
> > Minor nit: I'd prefer some more signposting about how a conventional learning algorithm is defined, or some notation of (x_ICL, y_ICL) versus conventional (x_train, y_train).
>
>
> Thanks for bringing this to our attention! We well gladly add more detail and examples to our introduction of conventional learning algorithms.
>
>
> > Is a deep learning model considered a conventional learner if it can fit to arbitrary noise?
>
>
> Thanks, this is a good question. For the purposes of this draft, we would consider a deep learning model with appropriate optimization a conventional learning algorithm, even if it can fit to arbitrary noise. This is because we would expect this learner to conform to all the expectations we express in the null hypotheses (learns the label distribution, overcomes prior, considers all datapoints (approximately) equally).
>
>
> > In section 5 I'd appreciate an example of what randomizing the labels exactly means. [...]
>
> > While I found the experiment in section 6 to be quite clever, I feel like "novel task we create" can be made a bit more explicit by specifying it's based on your private data [...]
>
> Thanks for highlighting these. We will improve the draft here to make sure this is clear!
>
>
> (Please also see part 2 of our response.)

---

> ### Author Response · Authors · 2023-11-15
> **Author Response to Reviewer WYZX (Part 2/2)**
>
> (Please read part 1 of our response first.)
>
> > This observation that larger models are more sensitive to label randomization is concerning. I would have expected that with more pre-training data and parameters, the model would be more confident in its prediction and less influenced by random noise in the in-context examples. I would have expected that with more pre-training data and parameters, the model would be more confident in its prediction and less influenced by random noise in the in-context examples.
>
>
>
> Thanks for your thoughts here. We agree it is potentially concerning that larger models are more sensitive to randomized labels. It seems that, in general, larger models are more sensitive to the context, for better or for worse.
>
>
> We agree with the intuition that more relevant pre-training data should make a model more confident in its predictions, and thus less susceptible to random in-context noise. However, the LLaMa models are actually all trained on similar amounts of tokens (1T for the small models, 1.4 T for the large models). Further, it is unclear how ‘relevant’ this (additional) pre-training data is towards informing ICL predictions on our selection of tasks.
>
>
> However, our paper does actually contain some results that support the intuition that more relevant data leads to more confident/less flexible predictions. For the Falcon models, we report results for both the base and instruction-tuned variants. We find that instruction-tuned Falcon is somewhat more robust against label randomization, e.g. compare SST-2 for Falcon-40B in Figure F.11 and SST-2 for Falcon-40B Instruct in Figure F.12. One could speculate that the instruction tuning, which exposes the model to many tasks with non-randomized labels, reinforces the models belief to predict default label relationships confidently.
>
>
> Again, thank you very much for your review. We hope that our reply has addressed any remaining concerns. Please let us know if there are any further changes you would like to see or if there is anything else that we can clarify.

---

> > ### Comment · Reviewer_WYZX · 2023-11-22
> >
> > Thank you for your response and for sharing some of these new experimental results. I will keep my score and recommend acceptance.

---

> > > ### Author Response · Authors · 2023-11-22
> > > **Author Response**
> > >
> > > Thank you for your response! We are glad to see you appreciate our rebuttal and the new experiment results, and that you continue to recommend acceptance.

---

### Official Review · Reviewer_LQGJ · 2023-10-31

**Soundness:** 3 good
**Presentation:** 3 good
**Contribution:** 3 good
**Rating:** 6
**Confidence:** 3

**Summary:**

This paper investigates different aspect of ICL, including does ICL rely on label information, does ICL view different orders of examples the same, can ICL overcome the pre-training preference. By using extensive study, the authors show that ICL do rely on label information, ICL treat different position differently, and ICL cannot overcome the pre-training preference. Thus reaching the conclusion that ICL is not conventional learning.

**Strengths:**

I really appreciate the number of experiments done in this paper. The authors try lots of parameters and spend lots of time to do the experiments (including ablation studies). These experiments make the paper sound and solid. Besides, the presentation is generally good. The argument that: ICL cannot overcome the pretraining preference is very interesting!

**Weaknesses:**

My main concern is that: although some of the claims in this paper are interesting, the depth of the explanation/experiment design is not enough. For example, in Min et al, even in their experiment results (just focusing on the accuracy), the random label ICL performance and the true label ICL performance are not the same (I remember in Min et al, claim that many tasks’ acc difference is less than 5%, but it is not a small number of classification tasks), and on some tasks, they are different very different. Thus, one can already reach the conclusion that ICL needs to use the label information **to some extent**. However, people do not know how ICL utilizes this label information. The experiment results in this paper are not enough to tell people how ICL utilizes this label information. It only shows that under another metric (the probability), using true labels and random labels is very different. It seems more like an ablation study of Min et al. Another claim for the position importance of different ICL examples, the author claims that ICL does not treat each in-context example equally. From my side, it seems to be a well-known thing, since people change the order of the ICL examples and can observe huge differences in the prediction accuracy [1]. However, people don’t know how ICL treats these examples. I don’t think the experiment in this section deepens our understanding of how/why ICL treats different in-context examples differently. It seems to use new experiments to support some known or folklore claims.

I suggest the author to emphasize some new/deeper implications from their experiments in the rebuttal session (for NH1 and NH3). NH2 seems new to me and this finding is v. interesting (but i am not sure if other people already know this).

[1] An Explanation of In-context Learning as Implicit Bayesian Inference, Xie et al

**Questions:**

Please see the weakness part, thanks! In addition, in the NH2 --- the ICL cannot overcome pre-trained preferences-- did you try the calibrated version of ICL, proposed by [1]? I am not sure if NH2 is still rejected if you calibrate the output.

[1] Calibrate Before Use: Improving Few-Shot Performance of Language Models, Zhao et al, ICML 2021.

---

> ### Author Response · Authors · 2023-11-15
> **Author Response to Reviewer LQGJ (Part 1/2)**
>
> Thank you for your hard work and helpful feedback! We hope that our responses below and the accompanying paper updates alleviate your concerns.
>
> > One can already reach the conclusion that ICL needs to use the label information to some extent
>
>
> We actually believe that there has been substantial controversy about this in the literature, with lots of papers arguing actively against it. For example, the conclusions of Min et al. are that ICL does ‘not learn new tasks at test time’ and that ‘ground truth demonstrations are in fact not required’ in many common scenarios. This is explicitly contradicted by our work. We agree that one can find _some_ support for our conclusions even in the Min et al. paper (more on this below), but the paper’s main conclusions actively and strongly contradict ours! As such, we think there is an urgent need in the literature to clear up this issue and provide more concrete evidence. And we believe that our rigorous experiments do exactly this: they very clearly show that ICL predictions do depend on labels, making this far clearer than it was previously, for example, in the Min et al. paper.
>
>
> > I don’t think the experiment in [Section 8] deepens our understanding of how/why ICL treats different in-context examples differently. It seems to use new experiments to support some known or folklore claims.
>
>
> While we agree that the results of this section are perhaps consistent with some of the current ‘folklore’, we believe that our experiments and conclusions provide concrete evidence for something that has not been demonstrated in the existing literature. In particular, our conclusions go beyond the findings in previous work that ‘ICL does not treat each in-context example equally’. Concretely, we show that ICL prefers to learn the _label relationships_ of examples closer to the query. In other words, our work provides insights into _how_ ICL depends on order that are not evident from prior work.
>
>
> > people do not know how ICL utilizes this label information.. The experiment results in this paper are not enough to tell people how ICL utilizes this label information.
>
>
> While we agree that investigating how ICL uses utilizes label information is a highly important topic of investigation, we disagree that the contributions of this paper are ‘not enough’ for publication for two reasons. Firstly, we believe that our results _do_ contribute a variety of significant novel insights that will be of significant interest to the community: we resolve the controversies around NH1 as mentioned above, we show LLMs can learn truly novel label relationships in-context, the ‘v. interesting’ findings around NH2 show ICL struggles to overcome pre-training preference, and NH3 shows ICL does not aggregate in-context label information like conventional learners. Secondly, we completely agree with you that there is far more to investigate on this topic. However, the paper is already struggling to stay within the page limit without any redundant content to cut out. We thus feel that it is perhaps a strength of this work to provide the foundations that open up new research avenues for future work to make further progress on this important problem.
>
>
> > [Additional Insights]
>
>
> We would also like to bring to your attention to two of the four new experiments that we have performed for the rebuttal, which provide even more novel insights into label learning in ICL.
>
>
> (1) The ‘Progressive Randomization’ experiment in Appendix G.1 studies how robust ICL is to more moderate levels of label noise, an important practical scenario. (2) The ‘Answer in Context’ experiment in Appendix G.4 studies the decision function of ICL by purposefully adding _test_ queries to the in-context _training_ examples. This gives us further insight into how ICL uses label information to update its predictions.
>
>
> We would like to refer you to Appendices G.2 and G.4 for a detailed discussion of these results. Please let us know if there are any other particular experiments you would like to see, and we will happily add them.
>
>
> (Please see part two of our response.)

---

> > ### Comment · Reviewer_LQGJ · 2023-11-19
> > **Reply**
> >
> > Thanks for your reply.
> >
> > As for the first part (NH1), I don't think the answer is satisfying. I still feel like people really know that label information helps, since even in Min et al, the performance difference is already there, although the authors try to say that the difference is not so big.
> >
> > I appreciate the authors adding new experiments for NH2, the calibrated exp. This makes the findings more interesting. However previously I already felt this result surprising, thus this will not lead me to raise my score.
> >
> > For NH3, the authors' explanation reminds me that I omitted some interesting contributions in the first round: the LLM will utilize more label information of the examples presented later. This will lead me to raise my score.
> >
> > In conclusion, I would raise my score from 5 to 6.

---

> ### Author Response · Authors · 2023-11-15
> **Author Response to Reviewer LQGJ (Part 2/2)**
>
> (Please read part one of our response first.)
>
> > It seems more like an ablation study of Min et al
>
>
> Even when just focusing on Section 5 (ignoring our contributions in Sections 6-9) and when disregarding our earlier point that Min et al draw the _opposite_ conclusions from us, we still believe that our results here are of significant interest to the community for two reasons:
>
>
> 1) We demonstrate the importance of using probabilistic metrics and studying a large variety of different context sizes when investigating ICL. Prior work almost always neglects both, which has, in some cases, lead to them drawing questionable conclusions. We would hope that our work inspires future work to be more thorough when studying ICL behavior.
>
>
> 2) The absolute drop in accuracies for label randomization is _much_ larger for us than for Min et al. While they find a ‘drop in the range of 0–5% absolute’, we find drops in the range of 2.8–_49.4%_ absolute across all models and tasks with better than random performance. Further, the average performance drop we find is 20.1% +- 11.9% (mean +- standard deviation), cf. Table F.1 in the paper for individual values. While you are right that one can ‘to some extent’ see that accuracies drop with randomized labels even for the results of Min et al., the _mean_ performance drop we find is _four times_ larger than the _maximum_ drop reported by Min et al. (We discuss reasons for this in Section 5.)  We believe the community will benefit from correcting the misconception that ‘ground truth demonstrations are in fact not required’. If you think this is helpful, we are happy to add this discussion to the next version of the draft.
>
>
> > In addition, in the NH2 --- the ICL cannot overcome pre-trained preferences-- did you try the calibrated version of ICL, proposed by [1]? I am not sure if NH2 is still rejected if you calibrate the output.
>
>
> Thanks, this is an excellent suggestion. Our appendix already contains some results for calibration according to Zhao et al. (2021) with default and randomized labels (cf. Figure E.2), where we find that calibration does not significantly improve ICL across tasks. However, given your explicit request, we are happy to also provide results with calibration for the experiments in Figure 5 regarding NH2. You can find these new experiments in Appendix G.2. We again find, that the effects of calibration are negligible and do not change our conclusions.
>
>
> We do not necessarily find these results surprising.  Zhao et al. (2021) focus on scenarios with small in-context dataset sizes (up to 16). They show that this is where calibration is most important. Already for 16 in-context demonstrations, the gains from calibration are insignificant, see, e.g. their Figure 1. We study ICL at large numbers of in-context demonstrations (20-150 examples). Thus, we are not surprised that calibration does not change our conclusions.
>
>
> Again, thank you very much for your review. We hope that our reply has addressed your concerns. Please let us know if there are any further changes you would like to see or if there is anything else that we can clarify.

---

> ### Author Response · Authors · 2023-11-19
> **Author Response to Reviewer**
>
> Thanks for engaging with our response! We are very glad to see that you are satisfied with our explanation regarding NH3 and that you find NH2 even more interesting now. Thank you for increasing your score.
>
>
> However, we are disappointed to see that you still think our results regarding NH1 are not sufficiently impactful, and we would _really_ appreciate it if you would continue to engage with us on this remaining point of discussion.
>
>
> You say that you 'feel like people really know that label information helps'.
> However, this is _not at all_ the feeling one gets when looking at citations of Min et al. in the literature. The following are quotations related to Min et al. from randomly selected prior work:
>
> * "One particularly noteworthy finding is that Min et al. (2022) show that models learn just as well with incorrect labels as opposed to correct labels in priming" (https://arxiv.org/abs/2109.01247)
> * "They found that randomly replacing labels in demonstration barely hurts performance on classification tasks" (https://arxiv.org/abs/2301.07069)
> * "Min et al. (2022) similarly find that shuffling labels in in-context learning demonstrations has a minimal impact on few-shot accuracy" (https://arxiv.org/abs/2202.07206)
> * "A recent work [Min et al., 2022b] even questioned the necessity of ground-truth input-output mapping: using incorrect labels in the examples only marginally lowers the performance." (https://arxiv.org/abs/2210.03493)
> * "the role of demonstration examples is more in specifying the task rather than informing the input distribution (Min et al., 2022)" (https://arxiv.org/abs/2210.09150)
> * "However, it is unclear to what extent these models are able to learn new tasks from in-context examples alone [...] (see Min et al. [...]" (https://proceedings.neurips.cc/paper_files/paper/2022/hash/c529dba08a146ea8d6cf715ae8930cbe-Abstract-Conference.html)
> * "Recently, Min et al. (2022) showed that the model does not rely on the ground truth input-label mapping provided in the demonstrations as much as previously thought." (https://arxiv.org/abs/2112.08633)
> * "it has been observed that replacing the inputs or labels of demonstrations with random ones sampled from the input or label space does not seriously hurt the performance of LLMs, indicating that LLMs mainly recognize the target task from demonstrations instead of learning from them [370, 385]." (https://arxiv.org/abs/2303.18223)
> * "few-shot performance can be largely attributed to the model’s zeroshot learning capacity (Min et al., 2022)" (https://arxiv.org/abs/2211.04486)
> * "Min et al. (2022) show that [...] the model does not rely on these examples to generate the final output." (https://arxiv.org/abs/2212.02437)
> * "recent work raised questions as to the degree to which correct labels are necessary (Min et al., 2022c)" (https://arxiv.org/abs/2209.01975)
>
>
> As you can see, these papers repeat the claims of Min et al., mostly uncritically. Further, we believe, these works do not support the feeling that 'people really know that label information helps'. If our results here support _your intuition_ this is great, however, support for the ideas of Min et al. might be more popular than you think. And for the sake of scientific progress, we believe it is extremely important to _publish_ work that refutes the claims of Min et al. that ICL 'do[es] not learn new tasks' and that 'that ground truth demonstrations are in fact not required'.
>
>
> The performance drops we find are often _much bigger_ than the 'marginal' differences Min et al. report. **We find many instances where accuracy drops to _random guessing levels_ (!!) with random labels, see e.g. Figure F.5.** We also wish to highlight our novel experiments with the author identification task in Section 6 which show very clearly that ICL _can_ in fact learn novel tasks. These are novel results that directly contradict some of Min et al. claims and that, we believe, are of considerable interest to the community.
>
> Thanks again and we hope we can discuss this with you as much as possible in the time remaining!

---

### Author Response · Authors · 2023-11-15
**Author Response to All Reviewer**

We thank all the reviewers for their insightful comments and helpful suggestions. We are glad that the paper was generally well received and hope that our responses and paper updates alleviate the concerns that were raised.


We are pleased that you felt our paper studies an ‘open question in the field’ (tkcu) in a ‘novel and rigorous’ (WYZX) and ‘sound and solid’ (LQGJ) way, and that we draw ‘valuable’ (WYZX) conclusions. In particular, we are glad to see LQGJ highlighting NH2 as ‘new’ and ‘interesting’ while Sjxd appreciates NH3 as ‘convincing’ and WYZX finds our novel author attribution task experiments ‘quite clever’. We appreciate you perceive our experimental study as ‘extensive’ (LQGJ), ‘thorough’/’well-designed’ (WYZX) and as considering a ‘very good number of tasks’ (Sjxd). Lastly, we are grateful you felt our submission is a ‘clearly written paper that is easy to follow, with plots that tell the story well’ (tkcu), ‘very clear’ (WYZX), and with an ‘excellent’ score for presentation (Sjxd).


Following your feedback, we have updated the draft with _four new experiments_ in Appendix G at the end of the paper:

* **(G.1) Progressive Randomization**: Following feedback from reviewer WYZX, we complement our experiments on label randomization (Section 5, NH1) by studying ICL predictions when only a fraction of the labels is randomized. We find that, as the proportion of random labels increases, ICL probabilities sensibly react with increased uncertainty.
* **(G.2) Calibrated Label Flipping**: Following feedback from reviewer LQGJ, we study if the calibration technique of Zhao et al. (2021) helps ICL overcome its pre-training preference in the label flipping experiments of Section 7 (NH2). We find this is not the case and our conclusions remain unchanged.
* **(G.3) Label Flipping for Author Identification**: Following feedback from reviewer tkcu, we repeat the label flipping experiments of Section 7 (NH2) for our author identification task (Section 6). As the author id task is guaranteed to be novel for the LLM, this allows us to study how ICL behaves in a scenario where pre-training preference is likely weak. Consistent with our expectations, we find that differences between label scenarios are much smaller than for most of our previous experiments.
* **(G.4) Answer In Context**: Following feedback from reviewer WYZX, to better understand how ICL uses label information, we study ICL when some of the in-context examples exactly match the test query. That is, the model can achieve perfect accuracy by looking up the label of the exact match from the context. We find that, across label setups, a single repetition of the query is insufficient to elicit copying behavior, but that this emerges robustly for multiple repetitions.


We refer to Appendix G for an extended discussion of these results.
As there is no separate phase for us to prepare a rebuttal before the discussion period, we wanted to get back to you as soon as possible.
However, this means that, in some instances, we are still working on extending our additional experiments to our usual selections of task and models.
Further, note that we have only collected all experiments in Appendix G for your convenience during the discussion period, and we will will distribute them throughout the paper as appropriate for the final version.


Again, thank you for your hard work. We believe your input has already helped improve the paper and look forward to engaging with you further during the discussion period. Please see our individual replies to each of you below.

---

### Meta-Review · Area_Chair_2nTB · 2023-12-10

**Metareview:**

The authors present a hypothesis-driven analysis of in-context learning in LLMs, testing 3 questions: 1) does ICL depend on accurate label information, 2) can ICL overcome priors learned through pretraining, and 3) does ICL consider all information given in context equally?

The overall consensus seems to be that the paper is well-written and clear, and addresses important gaps in our current understanding of ICL. The experiments appear to be well thought out and offer useful new insights. The main criticisms centered around whether the findings were novel (vs eg Min et al and Zhao et al), general clarity issues, and also a point about the exact definition of “conventional learning”. The reviews were detailed and thoughtful, suggesting several experiments that authors were able to add to the paper, further strengthening it.

My impression is that this paper contributes some very valuable insights into how LLMs perform in-context learning, insights which are sorely needed. I found their rebuttal responses convincing, and that their findings are indeed novel and contributed to my understanding. During the course of discussions, the reviewers were able to achieve consensus to accept. Therefore I'm happy to recommend this paper for acceptance.

**Justification For Why Not Higher Score:**

I'm undecided between a poster and a spotlight. In the end I think the findings are not so ground-breaking that they need to be highlighted with a talk, but I'm happy to bump this up.

**Justification For Why Not Lower Score:**

This paper contributes useful insights, is well-written, and contains a thorough set of experiments. I believe it will be of great interest to the wider ICLR audience.

---

### Decision · Program_Chairs · 2024-01-16

Accept (poster)